# Southern Hemisphere westerlies as a driver of the early deglacial atmospheric CO$_2$ rise

L. Menviel [1,2], P. Spence [1], J. Yu [3], M.A. Chamberlain[4], R.J. Matear[4], K.J. Meissner [1] & M.H. England[1]

The early part of the last deglaciation is characterised by a ~40 ppm atmospheric CO$_2$ rise occurring in two abrupt phases. The underlying mechanisms driving these increases remain a subject of intense debate. Here, we successfully reproduce changes in CO$_2$, $\delta^{13}$C and $\Delta^{14}$C as recorded by paleo-records during Heinrich stadial 1 (HS1). We show that HS1 CO$_2$ increase can be explained by enhanced Southern Ocean upwelling of carbon-rich Pacific deep and intermediate waters, resulting from intensified Southern Ocean convection and Southern Hemisphere (SH) westerlies. While enhanced Antarctic Bottom Water formation leads to a millennial CO$_2$ outgassing, intensified SH westerlies induce a multi-decadal atmospheric CO$_2$ rise. A strengthening of SH westerlies in a global eddy-permitting ocean model further supports a multi-decadal CO$_2$ outgassing from the Southern Ocean. Our results highlight the crucial role of SH westerlies in the global climate and carbon cycle system with important implications for future climate projections.

---

[1] Climate Change Research Centre and ARC Centre of Excellence for Climate System Science, University of New South Wales, NSW 2052 Sydney, Australia.
[2] Department of Earth and Planetary Sciences, Macquarie University, NSW 2109 Sydney, Australia. [3] Research School of Earth Sciences, The Australian National University, ACT 0200 Canberra, Australia. [4] CSIRO Oceans and Atmosphere, ATAS 7004 Hobart, Australia. Correspondence and requests for materials should be addressed to L.M. (email: l.menviel@unsw.edu.au)

The natural climate variability of the last 800,000 years is dominated by glacial–interglacial cycles, with atmospheric $CO_2$ variations providing a major positive feedback[1]. However, the sequence of events leading to deglacial $CO_2$ rise remains poorly constrained and a combination of mechanisms has been invoked to explain the full ~90 ppm amplitude. These include reduced $CO_2$ solubility, global ocean alkalinity decrease[2], reduced iron fertilisation[3–5], increased Southern Ocean ventilation[6,7] and poleward shift of the Southern Hemisphere (SH) westerlies[8,9]. Modelling studies trying to tackle the problem of glacial–interglacial $CO_2$ changes mostly involve idealised sensitivity studies, often performed under constant pre-industrial[8,10–12] or LGM boundary conditions[3,4]. A detailed study of the deglaciation would provide a more direct link between changes in the climate and carbon cycle and would allow a direct model-data comparison, thus further constraining the processes responsible for atmospheric $CO_2$ changes.

HS1 (~17.6–14.7 ka), at the beginning of the last deglaciation, is an important period to understand as it represents a major phase of atmospheric $CO_2$ rise and the transition out of the glacial period. Paleoproxy records suggest that North Atlantic Deep Water (NADW) formation weakened significantly during HS1[13], effectively reducing the meridional heat transport to the North Atlantic and leading to cold and dry conditions over Greenland[14] and the North Atlantic[15]. In contrast, paleoproxies indicate that Antarctic surface air temperature and Southern Ocean surface waters experienced a warming of ~5 °C and ~3 °C[16,17], respectively. This warming is partly due to increased heat content in the South Atlantic, subsequent advection of warm waters through the Antarctic Circumpolar Current and the concurrent 40 ppm atmospheric $CO_2$ increase. However, the mechanisms leading to HS1 $CO_2$ rise are still poorly constrained and an overarching mechanism linking this $CO_2$ rise to the North Atlantic cooling and high southern latitude warming is still missing.

Recent high-resolution Antarctic ice core records[7,18,19] show that atmospheric $CO_2$ rose in two major phases at ~17.2 and ~16.2 ka, each associated with a ~0.2‰ decrease in the atmospheric carbon isotopic composition ($\delta^{13}CO_2$). In addition, atmospheric radiocarbon content ($\Delta^{14}C$) declined by ~112‰ between 17.6 and 15 ka[20]. Explaining the atmospheric $CO_2$ increase thus also requires attributing the concurrent $\delta^{13}CO_2$ and $\Delta^{14}C$ declines. $\delta^{13}CO_2$ integrates changes in terrestrial carbon, marine export production, oceanic circulation, and air–sea gas exchange[21], while $\Delta^{14}C$ is controlled by atmospheric $^{14}C$ production and carbon exchange between the atmosphere and abyssal ocean carbon or old terrestrial carbon.

A number of processes have been put forward to explain the early deglacial atmospheric $CO_2$ increase. Co-variations of iron flux and nutrient utilisation in the sub-Antarctic[5] suggest that iron fertilisation could exert a significant control on atmospheric $CO_2$ during HS1 through its modulation of the Southern Ocean biological pump efficiency. Modelling studies show that reduced iron fertilisation could lead to a millennial atmospheric $CO_2$ increase of ~10 ppm coupled with a 0.1 ‰ $\delta^{13}CO_2$ decrease[3,4,19,22], during the early deglaciation. However, iron fertilisation does not affect atmospheric $\Delta^{14}C$ or ocean ventilation ages, and variations in atmospheric iron deposition to the Southern Ocean are ultimately controlled by the exposure of the continental shelves, SH hydrology and winds. Indeed, a deglacial decline in South Atlantic ventilation ages has been observed[6], indicating a possible role of Southern Ocean ventilation in driving the deglacial $CO_2$ increase. This change in Southern Ocean ventilation could be modulated by the strength and position of the SH westerlies[7–9]. Idealised modelling studies performed under constant pre-industrial conditions have shown that stronger or poleward shifted SH westerlies could enhance deep ocean

ventilation thereby leading to an atmospheric $CO_2$ rise and $\delta^{13}CO_2$ decline[10,11,21]. While all numerical experiments performed show that stronger SH westerlies lead to an atmospheric $CO_2$ increase[10,11,21,23], the impact of changes in their latitudinal position is more ambiguous and could depend on their initial latitudinal position[23,24]. However, the latitudinal position of the SH westerlies at the LGM remains unclear[25,26]. In addition, a recent modelling study[12], also performed under constant pre-industrial boundary conditions, concluded that the SH westerlies did not lead to significant changes in atmospheric $CO_2$ during HS1. Instead, and even though no abrupt atmospheric $CO_2$ rise was simulated, the conclusion was that the HS1 atmospheric $CO_2$ rise was solely due to a reduced efficiency of the biological pump, resulting from a weaker Atlantic Meridional Overturning Circulation.

As the magnitude and rate associated with an oceanic carbon release to the atmosphere during the deglaciation have been questioned, a deglacial transfer of carbon from the terrestrial to atmospheric reservoir has been put forward, either as thawing of Northern Hemisphere permafrost[27] during HS1, or as a Northern Hemisphere terrestrial carbon release due to a southward shift of the Intertropical Convergence Zone (ITCZ) at 16.2 ka[19,28]. However, the global terrestrial carbon reservoir increased over the deglaciation[29,30] and the timing and magnitude of the permafrost carbon contribution remain poorly constrained.

So far, no three-dimensional transient simulation was able to reproduce the changes in atmospheric $CO_2$, its isotopic composition, as well as oceanic $\delta^{13}C$ and ventilation ages across HS1. Here, we explore the processes leading to the two-stage $CO_2$ increase during HS1, and their links to NADW weakening and Antarctic warming by performing a suite of transient experiments of HS1 with the carbon-isotopes enabled Earth System Model LOVECLIM[21]. This suite of simulations assesses the impact of Southern Ocean ventilation changes, including the potential role of buoyancy and dynamic forcing, such as meltwater and SH westerlies, in driving the rapid atmospheric $CO_2$ increase during HS1. The ocean carbon response to changes in SH westerlies is further assessed in a global eddy-permitting ocean model. We show that enhanced Southern Ocean convection and upwelling of Circumpolar Deep Water, driven by intensified SH westerlies, lead to an atmospheric $CO_2$ rise, $\delta^{13}CO_2$ and $\Delta^{14}C$ decrease in agreement with paleo-records[7,19,20].

## Results

**Simulating atmospheric $CO_2$ and $\delta^{13}CO_2$ during HS1.** The transient simulation is initialised from a Last Glacial Maximum (LGM) state constrained by oceanic $\delta^{13}C$ and ventilation age distributions[31]. The LGM ocean circulation is characterised by shallow NADW, relatively weak North Pacific Intermediate Water (NPIW) and very weak Antarctic Bottom Water (AABW), obtained by adding a meltwater flux into the Southern Ocean and weakening the SH windstress by 20%[31]. By forcing the model with changes in orbital parameters, Northern Hemisphere ice-sheet extent and albedo as well as freshwater input in the North Atlantic (Methods), the deep ocean convection in the Norwegian Sea is suppressed, thus resulting in very weak NADW formation during HS1 (Fig. 1a). Reduced moisture transport from the Atlantic to the Pacific and a deepening of the Aleutian low due to NADW cessation leads to stronger NPIW formation in agreement with paleoproxy records[32,33] (Fig. 1j).

The NADW weakening induces a southward shift of the ITCZ (Fig. 2, Supplementary Figs. 1 and 2) via reduced meridional heat transport to the North Atlantic. Modelling studies[34,35] have shown that this ITCZ shift could weaken the SH Hadley cell and strengthen the subtropical jet, which in turn would shift the

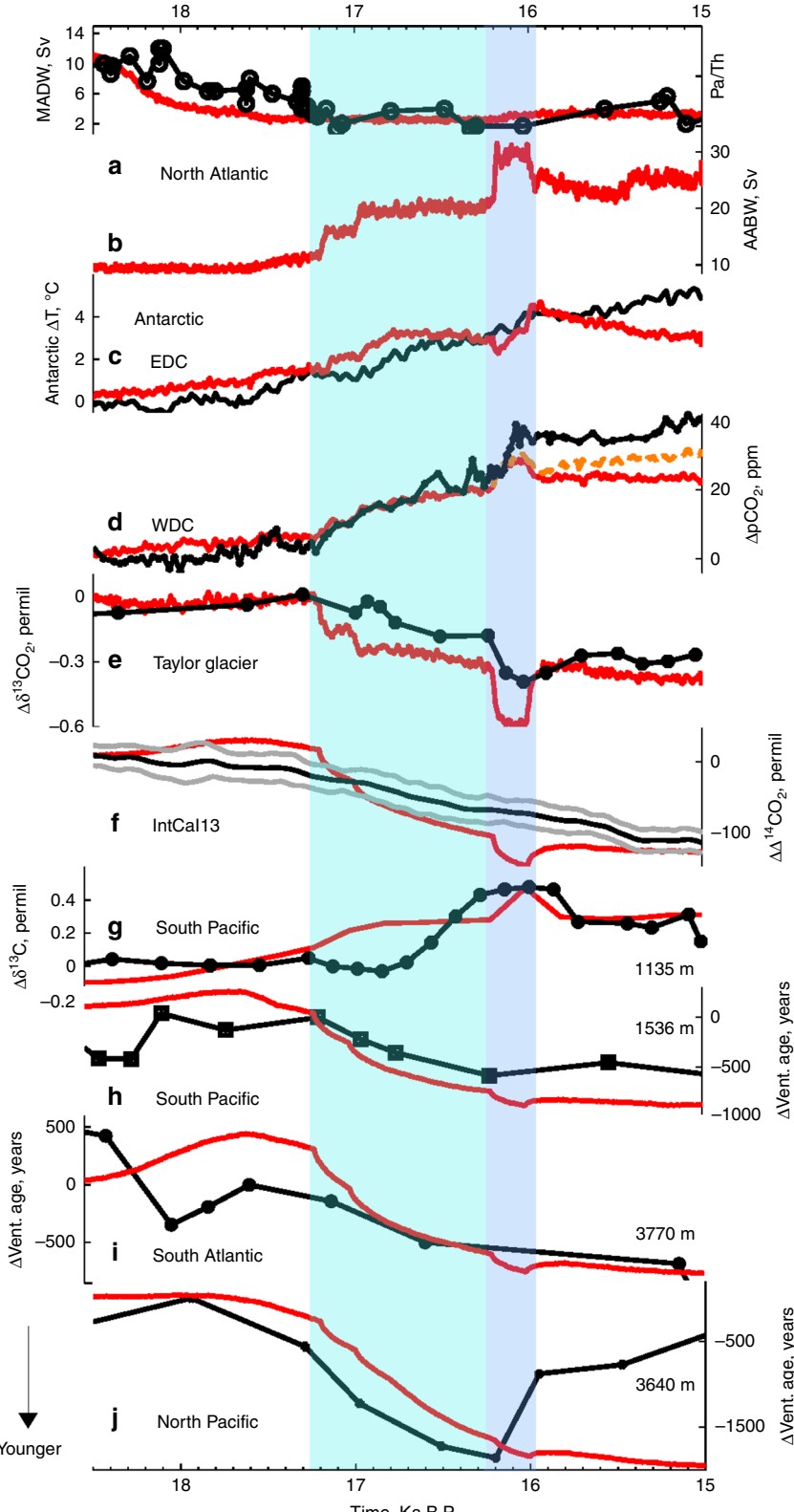

**Fig. 1** Model-paleodata comparison across HS1. Time evolution of selected paleo records across HS1 (black) compared to the results of a transient simulation (red, LH1–SO–SHW). **a** North Atlantic Pa/Th[81] and simulated NADW transport; **b** AABW transport; **c** Antarctic air temperature[16]; **d** Atmospheric $pCO_2$[7] on the WD2014 chronology[82]; **e** $\delta^{13}CO_2$[19] on the WD2014 chronology[82]; **f** $\Delta^{14}CO_2$[20]; **g** $\delta^{13}C$ anomalies averaged over the intermediate South Pacific (170–180°E, 40–30°S, 1007–1443 m) and compared to a benthic $\delta^{13}C$ record[55]; Ventilation age anomalies from **h** the intermediate South Pacific 75°W–50°W, 44°S–48°S, 1443–1992 m[38], **i** the deep South Atlantic 14°W–8°W, 40°S–48°S, 3300–4020 m[6] and **j** the deep North Pacific 152°W–147°W, 51°N–57°N, 3300–4020 m[33]. EDC and WDC, respectively, stand for EPICA Dome C and West Antarctic Ice Sheet Divide ice cores. 5–21 years moving average are shown for all the simulated variables except for $\Delta^{14}CO_2$ and the oceanic ventilation ages to filter the high-frequency variability. The dashed orange line in **d** includes a global ocean alkalinity decrease of $-6\,\mu mol\,L^{-1}$ per 1000 years

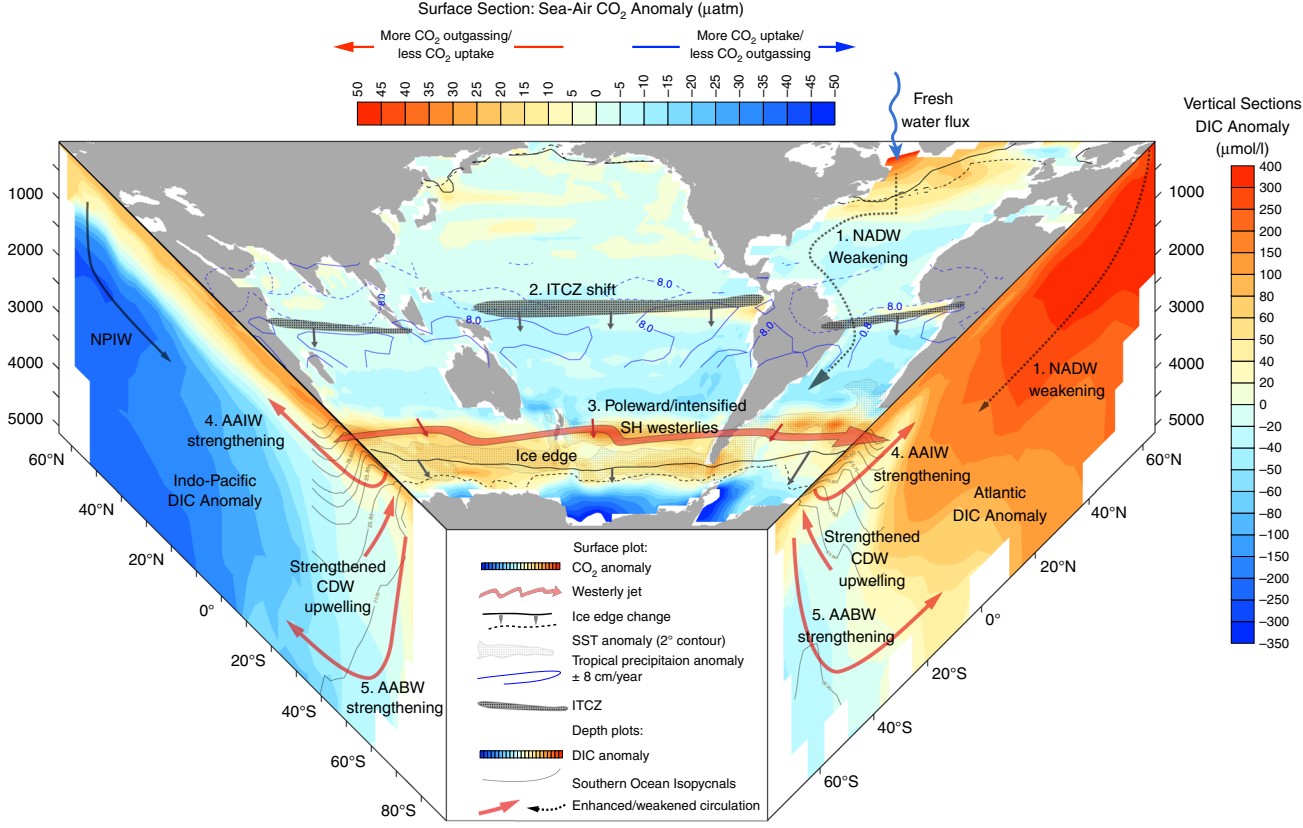

**Fig. 2** Sequence of events leading to a $pCO_2$ rise and initiation of the deglaciation. Results of experiment LH1–SO–SHW (as shown in Fig. 1) at 16 ka and compared to 19 ka. (Top centre panel) Sea-air $pCO_2$ anomalies (µatm), with positive values indicating a potential $CO_2$ flux out of the ocean (Southern Ocean) or reduced $CO_2$ uptake (North Atlantic). Overlaid are tropical precipitation anomalies (blue contour lines, ±8 cm yr$^{-1}$), Southern Ocean SST anomalies (grey dots indicate an area with ΔSST ≥ 2 °C), and the 0.1 m austral summer sea-ice contour (19 ka, solid black and 16 ka, dashed black lines). (Side panels) DIC anomalies zonally averaged over the (left) Indo-Pacific and (right) Atlantic basins. Southern Ocean isopycnals at 16 ka are overlaid (grey contours, 0.05 kg m$^{-3}$). 1. NADW cessation cools the North Atlantic and warms the South Atlantic, thus 2. shifting the ITCZ southward, 3. which strengthens/shifts the SH westerlies (SHW) poleward, 4. thus enhancing the upwelling of Circumpolar Deep Water (CDW) on a decadal timescale. 5. Polar/intensified SHW enhances deep ocean convection, leading to a centennial-scale Southern Ocean $CO_2$ outgassing and mid/high southern latitude warming

eddy-driven jet poleward and strengthen the SH westerlies by ~25%. To include this atmospheric teleconnection between the tropics and the high southern latitudes, which is not well represented in our coarse resolution atmospheric model, the SH westerlies are artificially strengthened from their LGM state commencing at 17.2 ka (Methods, simulation LH1–SO–SHW). This timing of 17.2 ka corresponds, within dating uncertainties, to the Bermuda Rise $^{231}$Pa/$^{230}$Th record reaching maximum values[13], thus indicating very weak NADW transport. However, another phase of NADW weakening[36], probably associated with the disintegration of the Laurentide ice-sheet and Heinrich event 1, occurred at ~16.2 ka[37]. A further intensification of the SH westerlies and reduced Southern Ocean freshwater flux at 16.2 ka enhances Southern Ocean convection in our simulation, resulting in stronger Antarctic Intermediate Water (AAIW) and AABW formation during HS1 (Fig. 3d, e), consistent with reduced ventilation ages in the South Atlantic[6] and the Pacific[38] (Fig. 1h, i) as well as a peak in the Southern Ocean opal flux[9].

Enhanced formation of AAIW and AABW and the associated upwelling of Circumpolar Deep Water decrease the oceanic carbon content below ~2000 m depth and particularly in the deep South Pacific, leading to $CO_2$ outgassing in the Southern Ocean (Fig. 2). As a result, the simulated atmospheric $CO_2$ increases in close agreement with high-resolution Antarctic ice core records (Fig. 1d)[7], with a simulated 19 ppm $pCO_2$ increase between 17.2

and 16.2 ka and an abrupt 9 ppm rise at 16.2 ka. Over the course of the experiment the global ocean carbon content decreases by ~100 GtC, while the terrestrial carbon content increases by ~50 GtC (Fig. 3g red line, Supplementary Table 1). This terrestrial carbon increase is mostly due to enhanced carbon storage in vegetation, roots and soils of the southern tropics due to a southward shift of the ITCZ, and the fertilisation effect linked to the atmospheric $CO_2$ increase (Supplementary Fig. 1).

In line with ice core records[19], each phase of atmospheric $CO_2$ rise is associated with ~0.25‰ $\delta^{13}CO_2$ decrease (Fig. 1e). Each of these drops in $\delta^{13}CO_2$ is however ~0.05‰ higher than the ones recorded in ice-core records, thus leading to a significant overshoot at ~16 ka. These $\delta^{13}CO_2$ decreases are primarily due to enhanced ventilation of deep and intermediate waters with low $\delta^{13}C$ signatures and upwelling in the Southern Ocean (Fig. 4a, b)[21]. This reduces the respired oceanic carbon content (Supplementary Fig. 3) and results in positive $\delta^{13}C$ anomalies along the AAIW and AABW pathways, as recorded in benthic $\delta^{13}C$ (Fig. 4a, b, Supplementary Table 2). During HS1, ventilation of "old" deep waters decreases atmospheric $\Delta^{14}C$ by ~150‰, consistent with atmospheric $\Delta^{14}C$[20] and marine $\Delta^{14}C$ reconstructions[6,32,33,38] (Fig. 1f,h–j and 4c, d, Supplementary Fig. 4).

Finally, due to the concurrent atmospheric $CO_2$ rise and enhanced meridional heat transport towards high southern

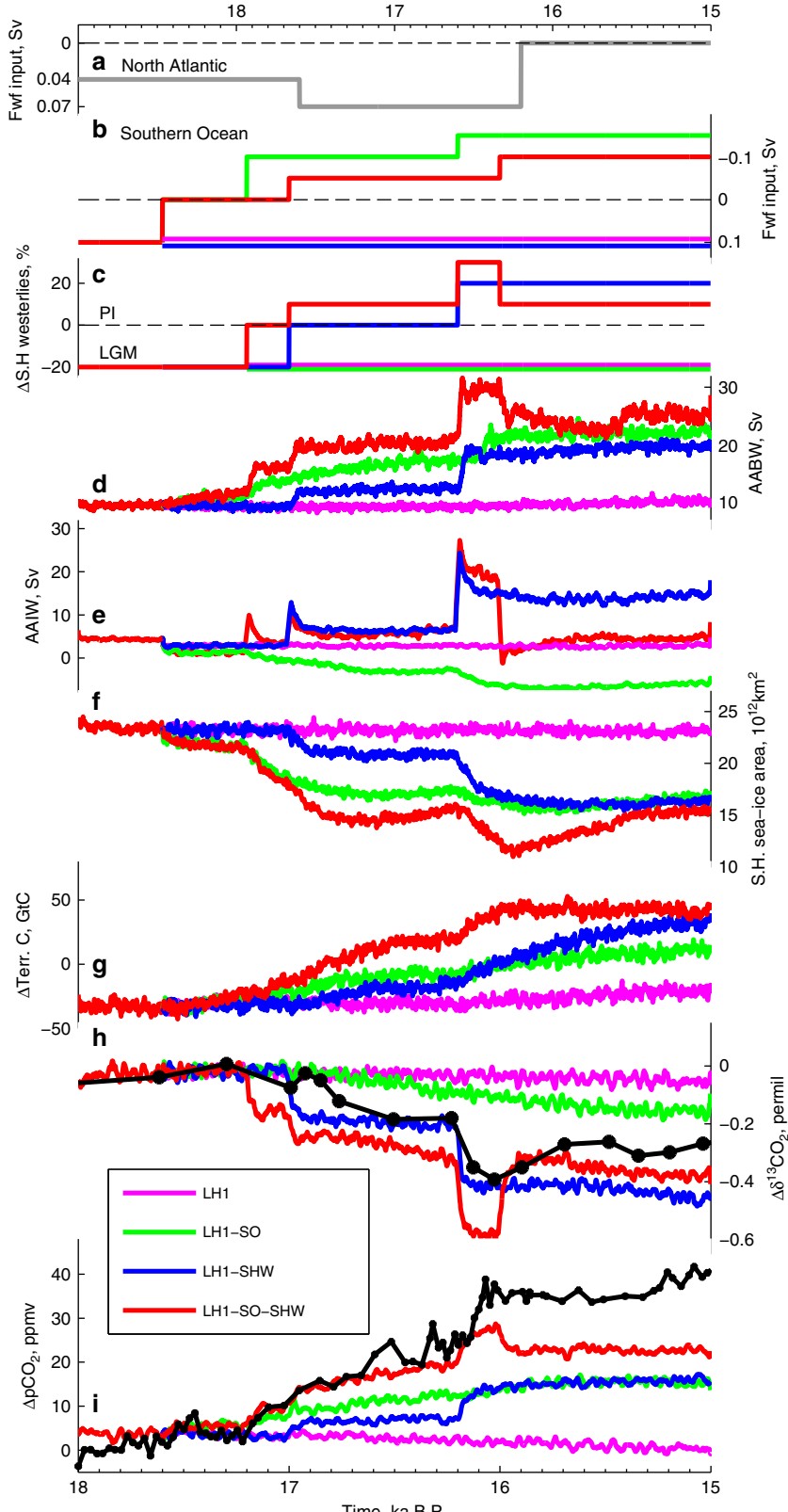

**Fig. 3** Summary of transient experiments. Timeseries of freshwater input into **a** the North Atlantic and **b** the Southern Ocean and **c** SH westerly wind forcing. Simulated **d** AABW; **e** AAIW; **f** SH sea-ice area; **g** terrestrial carbon reservoir anomalies with respect to 19 ka (Supplementary Table 1); **h** Δδ$^{13}$CO$_2$ and **i** ΔpCO$_2$; for all transient experiments (colour). 5–21 years moving average are shown for all the variables except for the terrestrial carbon to filter the high-frequency variability. WDC pCO$_2$[7] and Taylor Glacier δ$^{13}$CO$_2$[19] are shown in black

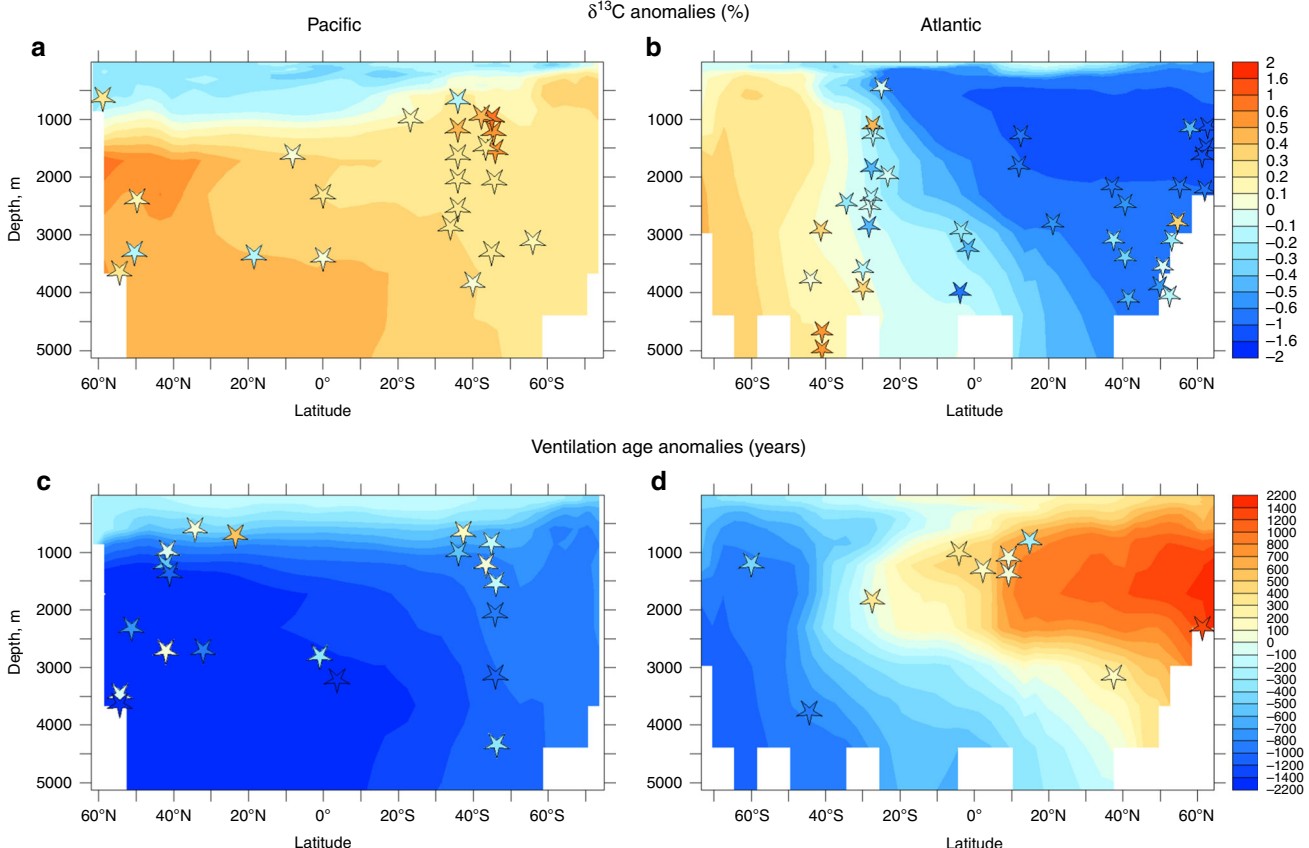

**Fig. 4** Ocean carbon isotopes. Simulated **a**, **b** $\delta^{13}C$ (‰) and **c**, **d** ventilation ages anomalies (years) at 16 ka compared to 19 ka, zonally averaged over **a**, **c** the Pacific and **b**, **d** the Atlantic in experiment LH1–SO–SHW. Stars represent paleodata estimates: **a**, **b** benthic $\delta^{13}C$ anomalies ($R = 0.77$, $p = 0.01$, Supplementary Table 2) and **c**, **d** ventilation ages anomalies ($R = 0.52$, $p = 0.01$, Supplementary Table 3)

latitudes, Southern Ocean sea surface temperature (SST) and Antarctic air temperature increase by up to 3 °C and 4 °C, respectively, between 17.6 ka and 15 ka, and the Southern Ocean sea-ice cover is significantly reduced (Figs. 1c, 2 and 3f red line). This is in agreement with idealised experiments performed with a global eddy-permitting coupled ocean sea-ice model, which shows that stronger AABW induces a Southern Ocean SST increase through enhanced poleward heat transport[39]. Enhanced AABW leads to a positive feedback between SST increase and atmospheric $CO_2$ rise through the solubility effect (Fig. 5 green and red), thus explaining a major part of the early deglacial warming at mid- and high southern latitudes.

**Processes driving atmospheric $CO_2$, $\delta^{13}CO_2$ and $\Delta^{14}C$ changes.** To investigate the mechanisms driving atmospheric $CO_2$ increase and $\delta^{13}CO_2$ decrease during HS1, three additional transient simulations (LH1, LH1–SO, LH1–SHW) are performed with LOVECLIM (Fig. 3, Methods). Similar to the simulation presented in Fig. 1 (LH1–SO–SHW, red), the experiments start from a LGM state featuring meltwater input in the Southern Ocean and weaker SH westerlies, and include a meltwater input in the North Atlantic during HS1 (Fig. 3a–c).

For LH1, Southern Ocean meltwater input and weaker SH westerlies are kept at the LGM level thus maintaining weak AAIW and AABW formation (Fig. 3, magenta). Enhanced NPIW decreases the carbon content at intermediate depths in the North Pacific (~700–2500 m depth), but NADW cessation leads to carbon accumulation in the Atlantic Ocean, mostly as respired carbon (Supplementary Fig. 3) at intermediate depths in the North Atlantic (Fig. 2, Supplementary Fig. 5). As a result, there is

little change in atmospheric $CO_2$, $\delta^{13}CO_2$ or $\Delta^{14}C$ (Fig. 3h, i, Supplementary Fig. 6 and Table 1), as ventilation of North Pacific intermediate waters is compensated by reduced Atlantic ventilation.

In LH1–SO, the meltwater input in the Southern Ocean is stopped, resulting in stronger AABW, but SH westerlies and AAIW stay weak (Fig. 3d, e, green). Enhanced AABW steepens Southern Ocean isopycnals (Supplementary Fig. 5) and strengthens deep ocean ventilation, thus decreasing the ocean carbon content below 2000 m depth, particularly in the Pacific Ocean (Supplementary Table 1)[40]. This leads to a gradual 15 ppm atmospheric $CO_2$ increase through Southern Ocean $CO_2$ outgassing (Supplementary Fig. 7). In addition, the ventilation of $^{13}C$ and $^{14}C$ depleted deep ocean causes slow atmospheric $\delta^{13}C$ and $\Delta^{14}C$ decreases of 0.016 and 130‰, respectively[21] (Fig. 3h and Supplementary Fig. 6). However, the simulated rates of change of $CO_2$ and $\delta^{13}CO_2$ are smaller than recorded in Antarctic ice cores and constant throughout HS1, despite a step-wise forcing. As ventilating the deep ocean is a multi-centennial process, even relatively rapid AABW changes cannot reproduce the fast atmospheric $CO_2$ increase at 16.2 ka.

Finally, when the SH westerly windstress is intensified but the Southern Ocean meltwater input is kept at LGM level (LH1–SHW, Fig. 3b–e, blue), both AAIW and AABW strengthen and the isopycnal slopes steepen in the Southern Ocean with deep and intermediate waters outcropping south of 50°S (Supplementary Fig. 5)[41]. This results in very abrupt $CO_2$ outgassing in the Southern Ocean (Supplementary Fig. 7), and leads to a 8 ppm atmospheric $CO_2$ rise and 0.2 ‰ $\delta^{13}CO_2$ decline within ~100 years at 16.2 ka.

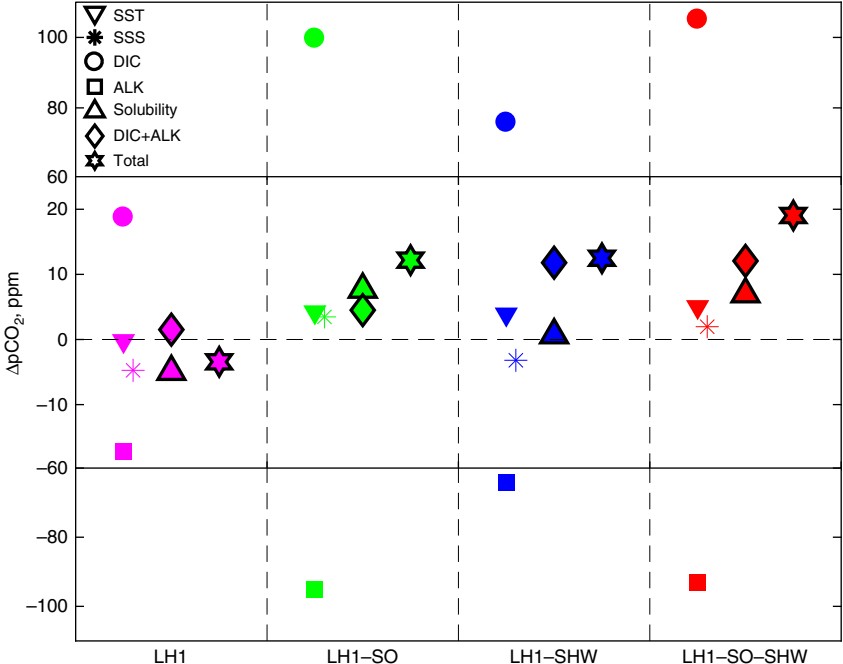

**Fig. 5** Attribution of $p$CO$_2$ changes. Deconvolution of $\Delta p$CO$_2$ into (from left to right in each column) its SST (downward-pointing triangles), SSS (six-pointed asterisks), DIC (circles) and ALK (squares); solubility (SST + SSS, upward-pointing triangles) and combined DIC and ALK components (diamonds Methods); and total $p$CO$_2$ change (six-pointed stars), for all transient experiments (LH1: weak AABW, magenta; LH1–SO: strong AABW through buoyancy forcing, green; LH1–SHW: strong AABW through SH westerlies, blue; and LH1–SO–SHW strong AABW through buoyancy forcing and wind, red)

A further weakening of the oceanic circulation from the LGM into HS1 as simulated in experiment LH1 only leads to small changes in the carbon reservoirs (Supplementary Table 1), with a slight global increase in the remineralized PO$_4$ content, and a 2% decrease in the preformed PO$_4$ content (Supplementary Fig. 3). In contrast, enhanced Southern Ocean ventilation leads to a carbon loss in the Pacific Ocean associated with a global decrease in remineralized PO$_4$ and a ~6% increase in preformed PO$_4$, thus implying a reduced efficiency of the biological pump in all experiments with enhanced AABW formation (LH1–SO, LH1–SHW and LH1–SO–SHW).

However, the processes leading to the atmospheric CO$_2$ increase also depend on the nature of the forcing: buoyancy (LH1–SO) or dynamic (LH1–SHW) forcing. Enhanced AABW via reduced surface freshwater flux leads to a global SST increase coupled to an increase in surface salinity, which induces a 8 ppm CO$_2$ rise through the solubility effect (Fig. 5, green triangle). In contrast, no notable CO$_2$ increase occurs through the solubility effect in the wind only forcing case (Fig. 5, blue triangle). While stronger deep ocean convection triggered by changes in the buoyancy forcing increases the southward baroclinic flow and thus the oceanic meridional heat transport to high southern latitudes[39], the SST response to changes in windstress is more complex. Stronger windstress over the Southern Ocean increases the Ekman transport and enhances the oceanic heat loss to the atmosphere, thus triggering an initial cooling in some areas of the Southern Ocean. However, after ~20 years, the eddy-driven poleward flow and enhanced upwelling of relatively warm Circumpolar Deep Water reverse the cooling trend into a warming trend[42]. In addition, an intensification of SH westerlies leads to an increase in the ventilation of low salinity AAIW, resulting in a global sea surface salinity (SSS) decrease.

The CO$_2$ rise in the wind forced simulation is due to an imbalance between Dissolved Inorganic Carbon (DIC) and alkalinity (ALK) increase (Fig. 5, blue diamond). As seen in Fig. 2, stronger AABW and enhanced upwelling in the Southern

Ocean lead to a DIC transfer from deep and intermediate depths of the Pacific and Southern Oceans up to the surface of the Southern Ocean. While an increase in surface DIC leads to an atmospheric CO$_2$ rise, an increase in surface ALK has an opposite effect. DIC and ALK thus tend to compensate each other as their oceanic distributions are similar (Supplementary Fig. 9) and their fractional impacts on CO$_2$ are of similar magnitude but opposite (Methods). When AABW is enhanced through reduced buoyancy forcing, this imbalance leads to a 5 ppm CO$_2$ increase, while it causes the 12 ppm CO$_2$ increase in the wind forcing case (Fig. 5, blue and green diamonds). This difference is mainly due to enhanced formation of AAIW in the simulation with stronger SH westerlies (Fig. 3e). Stronger AAIW formation entrains intermediate depth waters, characterised by low alkalinity, to the surface. In addition, this process acts on a faster timescale than deep ocean ventilation through enhanced AABW.

Both $\delta^{13}$CO$_2$ and $\Delta^{14}$C decrease are abrupt and of higher amplitude in the wind forcing experiment because of ventilation of intermediate waters and enhanced air–sea gas exchange, which is particularly important for carbon isotopes. When multi-millennial deep ocean ventilation and solubility decrease are combined with decadal scale changes in intermediate waters (Fig. 5 red, LH1–SO–SHW), the atmospheric CO$_2$ increase is in close agreement with ice core records (Fig. 1d).

The atmospheric $\Delta^{14}$C changes shown in Fig. 1f and S6 were obtained by keeping the atmospheric $^{14}$C production rate constant at LGM levels (Methods). Our results thus suggest that most, if not all, of the atmospheric $\Delta^{14}$C changes can be attributed to changes in ocean circulation and air–sea gas exchange with a smaller contribution from a varying atmospheric $^{14}$C production rate (Supplementary Fig. 4). The relatively large changes in atmospheric $\Delta^{14}$C simulated here are in line with a deglacial experiment performed with CLIMBER-2[43] showing the dominant role of reduced Southern Ocean stratification in decreasing atmospheric $\Delta^{14}$C across HS1. However, the simulated atmospheric $\Delta^{14}$C decrease is larger than the one simulated by

global carbon cycle box models[44,45], probably because of the large oceanic $\Delta^{14}C$ gradient present in the initial LGM state, itself resulting from the weak LGM oceanic circulation.

While the simulated changes in atmospheric $\delta^{13}CO_2$ and $\Delta^{14}C$ are in very good agreement with paleo-records, the $pCO_2$ increase is slightly underestimated over the last 1000 years of HS1. This could be due to an overestimation of the surface alkalinity increase in these simulations (Fig. 5 squares). It is important to note that the global ocean alkalinity inventory was kept constant during these transient simulations. However, a rising sea-level and an increase in deep ocean carbonate ion saturation during HS1[29] could have enhanced carbonate sedimentation, thus leading to a reduced global alkalinity content. Assuming a total glacial–interglacial alkalinity change of 96 μmol L$^{-1}$ (Methods), an ocean alkalinity decrease of 6 μmol L$^{-1}$ per 1000 years would induce a steady atmospheric $CO_2$ rise of ~5 ppm per 1000 years (orange line in Fig. 1d).

Recent studies[19,28] raised the possibility that the abrupt $CO_2$ increase occurring at 16.2 ka could be due to a Northern Hemisphere terrestrial carbon release resulting from a southward shift of the ITCZ. Our model-data comparison shows instead that an increase in Southern Ocean ventilation during HS1 cause an abrupt atmospheric $CO_2$ rise and $\delta^{13}CO_2$ decline in agreement with ice core and marine sediment records. In addition, our simulated terrestrial carbon content increases during this period due to a higher $CO_2$ content and warmer conditions (Fig. 3g). However, given the short duration of the 16.2 ka event and the relative low resolution of marine sediment records, a terrestrial carbon release cannot be ruled out (Supplementary Fig. 8).

Our simulations suggest that enhanced ventilation of AABW and AAIW played a crucial role in driving the atmospheric $CO_2$ increase during HS1. Stronger AABW transfers carbon from the deep ocean to the atmosphere, and concurrently leads to a warming south of 30°S on a millennial timescale. In contrast, increased AAIW formation caused by SH westerlies leads to a multi-decadal $CO_2$ rise.

**Rapid ocean carbon release in a global eddy-permitting model.** The strength of SH westerly winds exerts a strong control on the slope of Southern Ocean isopycnals and on the strength of the oceanic circulation. Mesoscale eddies are ubiquitous in the Southern Ocean and have a tendency to compensate for many of the wind-driven circulation changes that appear in non-eddying models[46]. To test whether the Southern Ocean $CO_2$ outgassing response to a strengthening of SH westerlies shown above is a robust feature, we perform an experiment under fixed modern-day forcing with a global high-resolution ocean sea–ice carbon cycle model that resolves most of the ocean mesoscale energy (Methods). The model is perturbed with a poleward intensifying Southern Ocean wind scenario based on observed and projected 21st century wind trends (Supplementary Fig. 10). The model's response is dominated by a large polynya in the Weddell Sea, that is similar in scale to an observed 1970's Weddell Sea polynya[47] and lasts for ~6 years (Supplementary Fig. 10). The polynya intensifies deep convection in the Weddell Sea, thus leading to a strengthening of AABW from 19 to 30 Sv and a $CO_2$ outgassing in the Southern Ocean (Fig. 6a, b). Open-ocean convection was likely the dominant source of AABW formation during glacial periods because grounded ice covered most of the Weddell and Ross Seas[48], where AABW is primarily formed today. Enhanced ventilation of intermediate and bottom waters, and associated increased upwelling of Circumpolar Deep Water lead to a total ocean carbon loss of 42 GtC over 50 years, corresponding to an upper estimate of ~20 ppm atmospheric $CO_2$ increase. Ventilation of Southern Ocean is rapid and leads to large

(~−60 μmol L$^{-1}$) DIC anomalies at intermediate depths (Fig. 6c). At 4300 m depth, negative DIC anomalies spread from the Southern Ocean towards the western Pacific and Atlantic basins, reaching about 20°N after 50 years (Fig. 6d). Enhanced chloro-fluorocarbon content in bottom and intermediate waters further confirms enhanced AABW and AAIW ventilation (Supplementary Fig. 11).

Similar to the coarse resolution experiment, most of the $CO_2$ flux is caused by a surface-water DIC increase, due to enhanced Ekman pumping of DIC-rich waters, with a strong compensation effect coming from the associated alkalinity increase (Supplementary Fig. 12a, b). However, warming of surface waters south of Australia and in the South Atlantic (Supplementary Fig. 12c), due to a poleward shift of the subtropical front, also significantly contributes to the $CO_2$ flux. In agreement with the simulations performed with LOVECLIM, global export production increases due to enhanced nutrient upwelling. Even though this simulation was performed under fixed modern-day forcing and the carbon cycle response to a latitudinal shift of the SH westerlies could depend on their initial position[24], this high-resolution simulation supports the significant role played by intensified SH westerlies in driving abrupt Southern Ocean $CO_2$ outgassing.

**Discussion**
Reduced NADW formation during HS1[13] weakens the meridional heat transport to the North Atlantic and induces a Southern Ocean warming, but our simulations suggest that the magnitude of this direct heat redistribution is small. In addition, reduced North Atlantic ventilation increases the carbon content in the Atlantic Ocean (Supplementary Table 1). Through oceanic and atmospheric teleconnections, NADW cessation enhances the formation of NPIW, thus ventilating the intermediate North Pacific. However, the decrease in North Pacific oceanic carbon content is compensated by the carbon increase in the North Atlantic, with negligible net effect on atmospheric $CO_2$ (Fig. 3i, Supplementary Fig. 5). Other mechanisms impacting SH climate and/or carbon cycle must have therefore been at play.

Modelling studies[34,35] have shown that the North Atlantic cooling can strengthen and shift the SH westerlies poleward via a southward shift of the ITCZ (Fig. 2). During the first phase of weak NADW transport at ~17.2 ka[13], an intensification of the SH westerlies and associated enhanced Southern Ocean deep convection could have led to a rise in atmospheric $CO_2$ concentration and initiated the deglaciation at high southern latitudes. Our simulations show that the resulting enhanced transport of AABW leads to a transfer of carbon from the deep Pacific to the surface of the Southern Ocean, thus inducing a millenial-scale atmospheric $CO_2$ increase. In addition, enhanced AAIW formation and the associated increased upwelling of Circumpolar Deep Water, lead to a multi-decadal $pCO_2$ increase. Both stronger SH westerlies and buoyancy loss near the Antarctic continent act to steepen Southern Ocean isopycnals, thus increasing the southward baroclinic flow. This results in a stronger southward meridional heat transport[39], decreasing Southern Ocean sea-ice concentration and further warming the mid- to high southern latitudes. Our simulations thus suggest that through its impact on both atmospheric $CO_2$ and temperature, enhanced Southern Ocean ventilation played a significant role during the last deglaciation.

Changes in SH westerlies, Southern Ocean circulation and the resulting impact on the hydrological cycle and terrestrial biosphere could have reduced the aeolian iron input into the Southern Ocean, thus potentially contributing to the atmospheric $CO_2$ rise during HS1[5]. However, changes in iron fertilisation alone cannot explain the observed variations in atmospheric

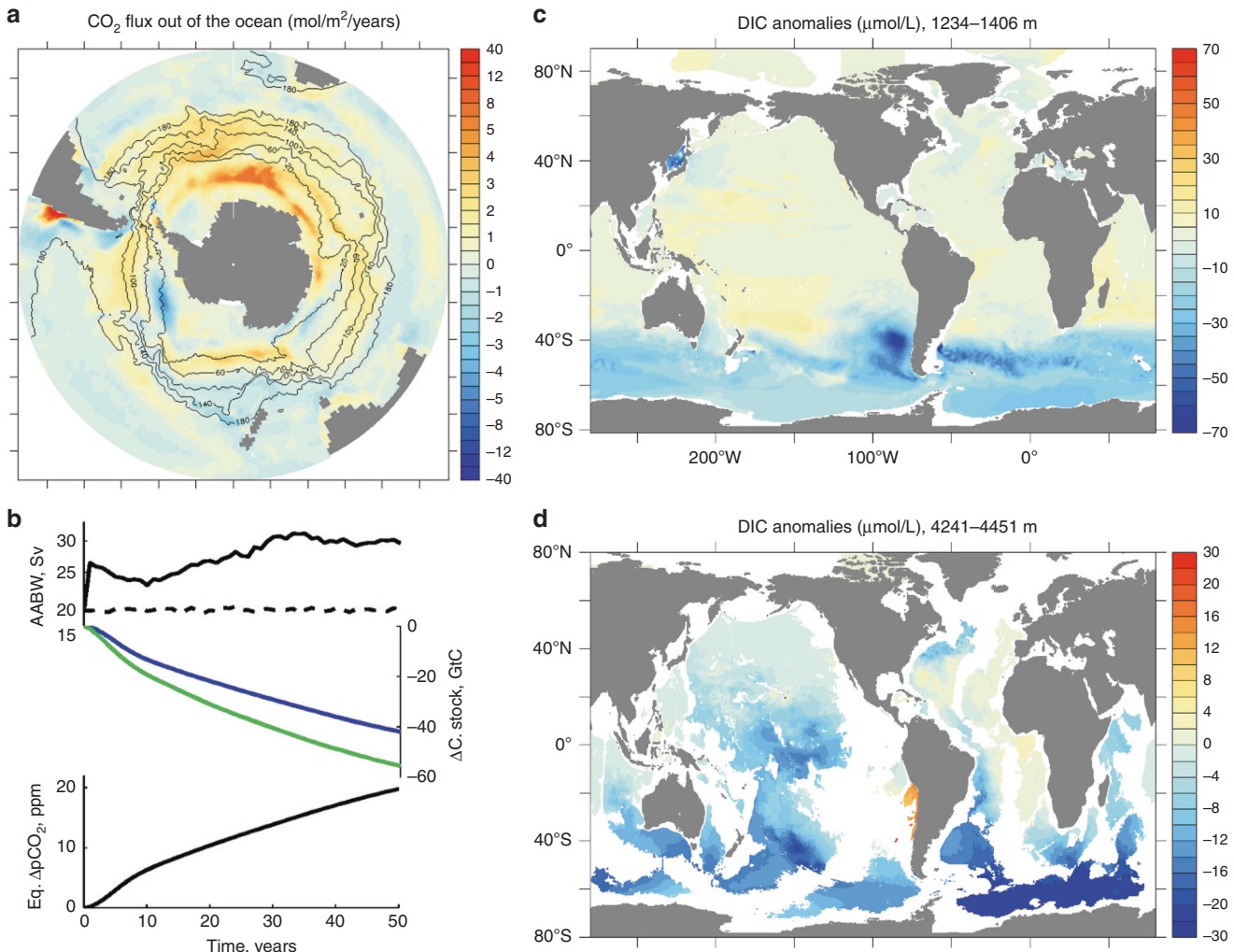

**Fig. 6** Oceanic carbon response to poleward intensified SH westerlies as simulated by a global eddy-permitting model. **a** $\Delta CO_2$ flux out of the ocean (shading, mol m$^{-2}$ yr$^{-1}$) with Antarctic Circumpolar Current stream lines (Sv) overlaid for the poleward intensified SH westerlies case. **b** Timeseries of maximum AABW transport in the perturbed (solid) and control (dashed) experiments, total (blue) and south of 36°S (green) ocean carbon change and $p$CO$_2$ equivalent ($\Delta C_{oc}$/2.12); DIC anomalies ($\mu$mol L$^{-1}$) at water depths **c** 1234–1406 m and **d** 4241–4451 m. Anomalies are at year 50 compared to year 50 of the control run

$\Delta^{14}$C, ocean ventilation and high southern latitude warming. We therefore suggest that changes in iron fertilisation mostly respond to changes in sea-level and Southern Ocean ventilation, possibly providing a positive feedback. In addition, while marine export production decreased north of the polar front[5], marine sediment cores south of the polar front display an increase in opal flux[9], thus indicating a limited geographic extent of iron fertilisation changes.

Sediment records from the Iberian margin have shown that ~16.2 ka probably corresponds to the beginning of Heinrich event 1 stricto sensu[37], and to another phase of NADW weakening[36]. North Pacific records suggest that after a period of strong NPIW formation during the first phase of HS1, NPIW weakened significantly at ~16.2 ka[33,49]. In addition, Chinese speleothems record a shift to significantly drier conditions at ~16.1 ka[50] (Supplementary Fig. 2). Both NADW and NPIW weakening at ~16.2 ka would cool the Northern Hemisphere, thus shifting the ITCZ southward, including in the Pacific sector[51] (Supplementary Fig. 1). This would further strengthen/shift the SH westerlies, enhance Southern Ocean deep convection and lead to the abrupt atmospheric CO$_2$ increase and $\delta^{13}$CO$_2$ decrease at ~16.2 ka. A

rapid CO$_2$ outgassing in the Southern Ocean due to intensified SH westerlies is confirmed by a simulation performed with a global ocean eddy-permitting model (Fig. 6).

Taking smoothing and dating uncertainties (~1 kyr) into account, the above sequence of events is in agreement with radiocarbon records, which suggest increased ventilation of the deep South Atlantic[6] and Pacific[52] and enhanced mixing between deep and intermediate waters at high southern latitudes[38,53] during HS1 (Figs. 1h, i and 4c, d). In addition, enhanced ventilation of deep and intermediate waters is consistent with positive benthic $\delta^{13}$C anomalies across HS1 south of 30°S and below 1000 m depth[38,54,55] (Figs. 1g and 4a, b) as well as high opal production in the Antarctic zone[9].

Our study highlights the crucial role of SH westerlies in driving abrupt atmospheric CO$_2$ rise and associated global climate changes. Given the projected poleward intensification of SH westerlies over the 21st Century, and the fact that the Southern Ocean has absorbed ~10% of anthropogenic CO$_2$ emissions[56,57], our results suggest a future reduction in CO$_2$ sequestration in the Southern Ocean, with significant impacts on future atmospheric CO$_2$ and climate change.

## Methods

**Carbon isotopes-enabled earth system model and LGM state**. Transient experiments were performed with the carbon isotope-enabled ($^{13}C$ and $^{14}C$) Earth system model LOVECLIM[58]. This coupled model consists of a free-surface primitive equation ocean model ($3° \times 3°$, 20 vertical levels), a dynamic–thermodynamic sea ice model, an atmospheric model based on quasi-geostrophic equations of motion (T21, three vertical levels), a land surface scheme, a dynamic global vegetation model[59] and a marine carbon cycle model[21,60].

The initial LGM state was selected amongst 28 LGM experiments based on its representation of oceanic $\delta^{13}C$ and ventilation age distributions[31]. It was obtained by equilibrating LOVECLIM under 35 ka B.P. boundary conditions, namely appropriate orbital parameters[61], Northern Hemisphere ice-sheet extent, topography and albedo[62], an atmospheric $CO_2$ content of 190 ppm, a $\delta^{13}CO_2$ of $-6.46‰$ and $\Delta^{14}C$ of 393‰. After a 10,000 years long equilibration phase, the model was run transiently until 20 ka with prognostic atmospheric $CO_2$, $\delta^{13}CO_2$ and $\Delta^{14}C$. During the equilibration phase, an atmospheric $^{14}C$ production rate of 2.05 atoms $cm^{-2} s^{-1}$ was diagnosed and then subsequently applied for all transient simulations, except for the deglacial simulations presented in Supplementary Fig. 4, in which the $^{14}C$ production rate is varied according to Hain et al.[44]. This rate is higher than Holocene and present-day $^{14}C$ production rate estimates[63,64] of 1.64 and 1.88 atoms $cm^{-2} s^{-1}$, consistent with a relatively high LGM[20] $\Delta^{14}C$.

The LGM state features weak (11.2 Sv) and shallow (~2500 m) NADW and very weak AABW (5.1 Sv) obtained through 20% weaker SH westerlies and a 0.1 Sv meltwater input into the Southern Ocean[31].

**Transient experiments of HS1**. A suite of transient experiments is performed starting from this LGM state by forcing the model with time-varying changes in orbital parameters[61] and Northern Hemispheric ice-sheet extent, topography and albedo[62]. In addition, meltwater is added to the northern North Atlantic (grey line, Fig. 3a) to obtain a nearly collapsed NADW. To explore the impact of changes in AABW on the global carbon cycle and climate, additional experiments are performed whereby AABW is enhanced by decreasing the buoyancy forcing (LH1–SO, LH1–SO–SHW) in the Southern Ocean and/or enhancing the simulated southern hemispheric westerly windstress (LH1–SHW, LH1–SO–SHW; Fig. 3b, c). The magnitudes of these changes in windstress are within estimates of probable windstress changes. Indeed some PMIP3 models display 10–20% weaker SH westerly winds at the LGM compared to pre-industrial times[65] and SH westerlies have increased by ~8% from 1990 to 2010 (equivalent to a ~15% increase in windstress)[66] and are forecast to keep on strengthening by ~15% over the coming century[67].

In all transient experiments presented here, the atmospheric $^{14}C$ production rate is kept constant and so is the total ocean alkalinity content except for the simulation shown by the orange line in Fig. 1d, where the total ocean alkalinity was decreased at a rate of 6 $\mu$mol $L^{-1}$ per 1000 years starting at 16.2 ka. The magnitude of that decrease is equivalent to a linear global alkalinity decrease of 96 $\mu$mol $L^{-1}$ over a time interval of 16,000 years. This corresponds to a first order approximation based on total alkalinity conservation and on changes in ocean volume ($V$, linked to sea-level changes): $\overline{[ALK_{PI}]} * V_{PI} = \overline{[ALK_{LGM}]} * V_{LGM}$.

The timeseries of AABW and AAIW transport refer to the maximum stream function of the zonally integrated meridional transport south of 60°S for AABW and at 1225 m between 30 and 60°S for AAIW.

**Decomposition of $pCO_2$ changes**. Changes in surface water $pCO_2$, which exert a dominant control on atmospheric $CO_2$, arise due to changes in DIC, ALK and solubility (SSS and SST)[68]. $pCO_2$ changes can thus be decomposed as follows (Fig. 5):

$$\Delta pCO_2 = \Delta pCO_{2DIC} + \Delta pCO_{2ALK} + \Delta pCO_{2SST} + \Delta pCO_{2SSS} \quad (1)$$

For DIC, ALK and SSS, $\Delta pCO_2$ can be expressed as

$$\Delta pCO_{2X} = \Delta X * \gamma_X * pCO_{2Ref} / \overline{X} \quad (2)$$

where $pCO_{2Ref}$ is the $pCO_2$ value at 19 ka; $\overline{X}$ represents the mean surface DIC, ALK or salinity value and $\gamma_{DIC}$, $\gamma_{ALK}$, $\gamma_{SSS}$ are Revelle factors equal to 10, $-9.4$ and 1, respectively[68].

The temperature contribution is derived from

$$\Delta pCO_{2SST} = e^{(\Delta SST * \gamma_{SST})} * pCO_{2Ref} - pCO_{2Ref} \quad (3)$$

where $\gamma_{SST}$ is equal to 0.0423 ([68]).

**Eddy-permitting global ocean sea-ice carbon cycle model**. Experiments are conducted with the eddy-permitting global ocean, sea-ice model MOM5, which is based on the Geophysical Fluid Dynamics Laboratory CM2.4 and CM2.5 coupled climate models[69,70]. The model has a 1/4° Mercator horizontal resolution with ~11 km grid spacing at 65°S. The MOM5 ocean model has 50 vertical levels and is coupled to the Sea Ice Simulator dynamic/thermodynamic sea-ice model. The atmospheric forcing is derived from version 2 of the Coordinated Ocean-ice

Reference Experiments Normal Year Forcing (CORE-NYF) reanalysis data[71,72]. CORE-NYF provides a modern-day climatological mean atmospheric state at 6-h intervals for 1 year and includes synoptic variability.

The model is coupled to the Whole Ocean Model with Biogeochemistry and Trophic-dynamics (WOMBAT) model, a Nutrient-Phytoplankton-Zooplankton-Detritus (NPZD) model[73,74]. WOMBAT includes DIC, alkalinity, oxygen, phosphate and iron, which are linked to the phosphate uptake and remineralisation through a constant Redfield ratio. Phytoplankton growth is limited by light, phosphate and iron, with the minimum of these three terms limiting growth. The biogeochemical parameters are slightly modified from the values used in the ACCESS-ESM simulations[75,76]. Two important changes needed for the high-resolution simulations were to increase detritus sinking rate to 20 m $d^{-1}$ and the background iron concentration was set to 0.3 $\mu$mol Fe $m^{-3}$ to reduce nutrient trapping and improve export production in the tropical East Pacific. The formation of calcium carbonate is a constant fraction of organic carbon production. The air–sea exchange of carbon dioxide is a function of wind speed[77] and sea ice concentration. Initial conditions for the biophysical fields are derived from an observation-based climatology[78].

The model was initialised with modern-day temperature and salinity distributions and equilibrated for 180 years of Normal Year Forcing (CORE-NYF). A control run and a wind perturbation experiment were then run for 50 years. The wind perturbation experiment includes a poleward intensifying wind forcing, namely a 4° southward shift and 15% increase in 10 m wind speeds between 30°S and 65°S (Supplementary Fig. 10). This wind forcing is based on projected SH wind changes in CMIP5 business as usual scenarios[67]. The model setup and experimental design are similar to the one employed in a previous study[79], except that previous simulations did not use any neutral physics ocean parameterisations. In the simulations presented here, neutral physics parameterisations are used, based on options from the ACCESS Ocean Model[80], with Redi diffusivity (600 $m^2 s^{-1}$) and Gent McWilliams skew diffusion (600 $m^2 s^{-1}$). These neutral physics options improved the simulated AABW transport and the distribution of ocean biogeochemical tracers relative to observations, for instance, the oxygen in the Southern Ocean and alkalinity of bottom waters penetrating into ocean basins (Supplementary Fig. 9).

**Data availability**. Results of the modelling experiments are available at https://doi.org/10.4225/41/5af39aae7960f and under T1c13 at http://climate-cms.unsw.wikispaces.net/ARCCSS+published+datasets.

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

## Acknowledgements

This project was supported by the Australian Research Council. L. Menviel, K. Meissner, P. Spence and J. Yu acknowledge funding from the Australian Research Council grants DE150100107, DP180100048, DE150100223, FT140100993 and DP140101393 as well as from NSFC41676026. K. Meissner acknowledges support from the UNSW Science Goldstar award. LOVECLIM experiments were performed on a computational cluster owned by the Faculty of Science of the University of New South Wales, Sydney, Australia. MOM5 experiments were performed on the Australian NCI super computer Raijin. Figure 2 was done with the help of Dr. Alex Sen Gupta.

## Author contributions

L.M. designed the study, performed and analysed LOVECLIM experiments. M.C. and R. M. coupled WOMBAT to MOM and calibrated MOM-WOMBAT. P.S. performed MOM-WOMBAT experiments with the help of M.C. P.S. and L.M. analysed MOM-WOMBAT experiments. L.M. wrote the manuscript with contributions from P.S., J.Y., M.C., R.M., K.M. and M.E.

## Additional information

**Competing interests:** The authors declare no competing interests.

