## [Peer Review File · Nature Communications]

Reviewers' Comments:

Reviewer #1:

Remarks to the Author:

This paper uses a simplified coupled climate model to study the response of the ocean, atmosphere and carbonate system to fresh water forcing. The objective is to simulate and understand the observed CO₂ increase in a couple of steps during the last deglaciation, 17-16,000 years ago. The subject is relevant and very interesting, and a coupled ocean-atmosphere model that incorporates an active carbonate system and tracks isotopic composition seems the right tool. The general idea that the Southern Ocean participated in causing glacial CO₂ changes, and that this was due to the response of the ocean biogeochemistry to changes in ocean circulation and mixing, is not new (the authors do only partial justice to previous works on the subject). This implies that the paper is to be judged to a significant degree based on its technical details, rather than as a representation of a completely new idea. Given this, I am afraid I did not find the case made by the paper to be strong due to several issues, making it difficult to recommend publication in Nature communications.

1) Adequacy of model formulation:

A) The general approach followed in the paper is to prescribe fresh water forcing in the northern hemisphere, presumably due to the deglaciation that started there for reasons that are beyond the scope of this paper, and observe the response. This response involves changes to the atmospheric inter-tropical convergence zone, leading to a response of the southern hemisphere winds, and then to a response of the Southern Ocean circulation and stratification, and finally to a response of the Southern Ocean biogeochemistry. However, the atmospheric model used here is a highly simplified 3-level quasi-geostrophic (QG) model. While simple models are extremely helpful in understanding the behavior of both the climate system and of more complex models, it seems that such a simplified model would not be able to realistically simulate the response of the atmospheric tropical convergence zones, which also puts in doubt the rest of the chain of arguments. Simulating these zones requires a reasonable representation of atmospheric convection, clouds and radiation feedbacks that a 3 level QG model seems incapable of including. In addition, the QG approximation breaks down at the equator, and it is therefore not clear how such a model can simulate the near-equatorial ITCZ atmospheric response. While the simple model used here can probably be tuned more easily to fit the observed record, it is not obvious what are the robust lessons the resulting simulation can provide.

B) Similar issues, although if less extreme, seem to exist with the ocean model, with its coarse vertical resolution (20 levels), given the need to represent different water masses at different ocean levels, and vertical convection and mixing processes that are an important part of the proposed mechanism.

C) Lines 70-71: "The LGM state is characterised by shallow NADW, relatively weak North Pacific Intermediate Water (NPIW) and very weak AABW, obtained by adding a melt water flux into the Southern Ocean and weakening the SH windstress by 20%."

- This seems to imply artificial fresh water forcing and artificial wind stress modification, both imposed in the Southern Ocean, in order to overcome what the authors considered model biases. But, of course, introducing such biases can lead to compensating model errors that can then influence the results of the forced simulation as well and thus influence the proposed mechanism. This seems reminiscent of the flux correction procedure that is long-gone in climate modeling.

D) I am confused whether the changes to the Southern Ocean winds are imposed by the authors (as suggested by Fig 3b), or are a response to other forcing (fresh water in north Atlantic) as the manuscript seemed to imply. Perhaps I simply misunderstood, but if the wind changes are imposed rather than simulated, this further weakens the case made by the paper.

2) Interpretation of the results:

A) Lines 165+: "Through its impact on both atmospheric CO₂ and temperature, enhanced Southern Ocean ventilation could have thus initiated the last deglaciation."

- It is not clear precisely how ice sheets are updated during the model run (there is a reference in the methods section to Peltier's reconstruction, but no details). Assuming that they are modified in time, and given that fresh water forcing is changed by the authors in the northern hemisphere and the Southern Ocean merely responds to these forcings, I don't see how the SO could be considered the initiator of deglaciation.

B) Lines 146-147: "Open-ocean convection was likely the dominant source of AABW formation during glacial periods, in contrast to modern climate where the frequency of large-scale Weddell Sea polynya events is uncertain."

- The logic seems problematic here: how can we be confident about the state of Antarctic water mass formation during glacial periods if there is a large uncertainty even at present?

C) Reason for DIC changes: the carbonate system response is described as involving changes to the Dissolved Inorganic Carbon concentration, but the mechanism is not clearly explained. In addition, is ocean biology (productivity) playing a role?

D) Lines 90-92: "Finally, due to the concurrent atmospheric CO₂ rise and enhanced meridional heat transport towards high southern latitudes, Southern Ocean Sea Surface Temperature (SST) and Antarctic air temperature increase by up to 3C and 4C, respectively..."

- It does not seem likely that such a small CO₂ change would play a dominant role in the significant 3 or 4C warming. This would indicate climate sensitivity beyond typically calculated values.

Also,

3) Figure 2: the fit between the observations (stars) and model results (shading) does not seem very good. It needs to be quantified and some measure of quality of fit needs to be used to make a convincing case that this is, indeed, a good fit.

4) It seems that figure 3 would best be divided into 3 different and still fairly complex figures. This, with some other comments above suggest that a different venue that allows providing more details about the study and using more figures may be a more appropriate choice.

Reviewer #2:

Remarks to the Author:

Review of "Abrupt early deglacial atmospheric CO₂ increase driven by Southern Hemisphere westerlies"

Menviel et al., submitted to Nature Communications

The manuscript by Menviel et al. provides exciting modelling results that tackle important questions regarding the drivers of glacial-interglacial CO₂. The paper tests a widely discussed but somewhat untested hypothesis for glacial-interglacial CO₂ by demonstrating a clear mechanistic link between Southern Hemisphere westerlies and atmospheric CO₂ during the last glaciation. The paper also proposes a completely new and exciting mechanism for centennial-scale changes in atmospheric CO₂ that involves ventilation of Antarctic intermediate waters. The paper is particularly timely as it is one of the first modelling studies that incorporates new ice core records in a thought-provoking method. It also builds upon a solid foundation of previous carbon cycle modelling with LOVECLIM. I believe the paper meets the requirements of Nature Communications in terms of novelty and quality (after some major revisions).

I have a few comments and suggestions that are likely addressable but I feel amount to a major revision. Most are related to providing a more detailed and clear description of the model experiments and greater discussion of previous work in the main body of the text. Given that the paper seems very short (main body = 1900 words) compared to the word limit (5000 words), I suspect this will not be problem. Overall, I thought the paper could have benefited from more in-depth discussion, so hopefully a more generous word count will help.

My comments will focus on the interpretation of the data as I am not qualified to discuss the technical details of the modelling.

Description of model forcing and the motivation behind the experiments

The biggest difficulty I had understanding the results was understanding why and exactly how the model had been forced. Just as an example, during my first casual reading of the paper I was stuck by the close between the modelled atmospheric CO₂ and data in terms of timing – most notably the 16.3 ka event. I was initially under the impression that this “event” was an unforced variation, but upon additional reading I became unsure.

In the first paragraphs (line 67 through line 97) the model description reads “By forcing the model with changes in orbital parameters, Northern Hemisphere ice-sheet extent and albedo as well as freshwater input in the North Atlantic (Methods, grey line Fig. 3a) during HS1....”. The description then goes into model predictions that are a consequence of these forcings. I took this to mean that these were the only forcings applied to the model and the following paragraphs were on simulated variables. The next paragraph continues “Via a southward shift of the Intertropical Convergence Zone (ITCZ), the North Atlantic cooling can strengthen and shift the SH westerlies poleward^{23, 24} (Fig. 2). Intensified SH westerlies and reduced Southern Ocean freshwater flux at 17.2 and 16.2 ka enhance Southern Ocean convection in our simulation, resulting in stronger AAIW and AABW formation during HS1...”

I found this language very ambiguous. Is a shift in the ITCZ observed in the model because the North Atlantic cooling or is this more general background information? Note that Figure 2 only shows some qualitative indicators of ITCZ and westerlies.

Furthermore, and this is vital to the whole paper, have the intensified westerlies and freshwater water fluxes in the SH forcings been chosen by the authors or did they emerge as a consequence of other forcings? Delving in the sensitivity experiments and Figure 3, it seems to me that the former is correct, which means that some of the agreement between the model and ice core data is artificial. If so, this needs to be stated explicitly. Also, the motivation for changing westerlies in the series of rapid steps should be outlined before the experiments are described. At the moment, there is some justification but it is only discussed very late in the paper (lines 167-172). I feel the paper would benefit from greater discussion of the motivation in the introduction (there is a narrative that describes NADW shutdown, the bipolar seasaw and ACC, but nothing about the westerlies). It would be a much stronger paper if the change in the westerlies could be derived in an objective way.

Are the authors justified in forcing the model in this way? I have one suggestion for providing this motivation that would be provide a more in-depth discussion of the previous hypotheses for the centennial-scale CO₂ variability during HS1, which I also think is necessary.

Marcott et al., 2014¹ hypothesized that the 16.3 ka event could ultimately be a consequence of Heinrich Event 1 “We propose that the abrupt rise in CO₂ at 16.3 kyr ago is directly related to an iceberg discharge event recorded in several ocean cores from the North Atlantic.... One potential cause of the CO₂ increase at this time could be a southward shift in the position of the Southern Hemisphere westerlies²⁴, leading to increased ocean upwelling and outgassing of respiration-derived CO₂”. By analyzing WAIS Divide CH₄ record Rhodes et al., 2015² found support for the

idea of a southward shifted ITCZ during Heinrich Stadials (right at 16.3 ka during the deglaciation). Bauska et al., 2016 used the carbon isotopes of CO₂ to propose that the 16.3 ka event could either be a terrestrial source (from a drying of the NH associated with the southward shifted ITCZ) or an increase in air-sea gas exchange in the Southern Ocean. "The southward migration of the ITCZ also led to drying in parts of the NH, possibly causing a reduction in organic land carbon, most notably around 16.3 ka. Alternatively, or additionally, the changing SH westerlies reached a threshold around 16.3 ka, in which wind speed over the SO increased, leading to enhanced air-sea gas exchange and possibly greater upwelling."

At the moment, these studies are used only for the data and not their ideas but to me they provide the perfect motivation for these experiments. It is also important to acknowledge that this hypothesis is already out there in the literature (even though you will prove the Bauska air-sea gas exchange hypothesis wrong with your superior model and may be able to test the terrestrial hypothesis - see more below).

Previous Modelling Studies

Some significant modelling work is currently ignored. How does this paper improve on the results of Tschumi et al., 20114 which tested the sensitivity of Bern3D model to westerly wind strength? Is the centennial-scale ventilation of AAIW present in Bern3D? Could this be a model dependent phenomenon or has it just been ignored or mischaracterized in previous work? There is also the work of Lee et al., 20115 with MEMSO that goes unmentioned, which may provide a stronger link between southward shift of the ITCZ and Southern Hemisphere westerlies (especially if the ITCZ shift is absent in the LOVECLIM model as discussed above).

It is also important to mention that an alternate hypothesis for the HS1 CO₂ was offered in Schmittner et al., 20156. These new results seem to contradict Schmittner despite using fairly similar model. For example the Schmittner abstract states "Modeled effects of AMOC-induced wind changes on the carbon and isotope cycles are small, suggesting that Southern Hemisphere westerly wind effects may have been less important for the global carbon cycle response during HS1 than previously thought." What drives this model dependent response? Which model is better suited for simulating the paleodata? Schmittner didn't spin up to a glacial-state (or at least a realistic one) so the Menviel et al., study is clearly a big improvement.

Eddy Permitting Model

The CO₂ prediction from the eddy-resolving method is really fascinating. Has the CO₂ increase been predicted with this type of model before? I am familiar with some work on CO₂ by Lauderdale et al., 20167 with MITgcm and an eddy-permitting sector model which may or may not be relevant. Also was the model forced only with windstress a no other 21st century changes (for example SST)? I think the language in the extended data is slightly ambiguous about the forcing.

Critical Evaluation of the Paleodata

I am somewhat concerned that the model predicts a large drop in 14C at the 16.3 ka event that is not present in the data. Bauska suggests that the rapid increases could be terrestrial or enhanced air-sea gas exchange. The absence of a 14C drop in data might rule out westerly hypothesis for this event (a more contemporaneous carbon source would be a better fit if it was superimposed on a slower ventilation change). But maybe IntCal13 isn't high enough resolution or accurate enough to capture this event? Please include the IntCal13 underlying data with error bars or uncertainty on the reconstruction (<http://www.radiocarbon.org/IntCal13.htm>). The same goes for the ice core data, which are currently plotted as lines which implies continuous rather than discrete data.

Moreover, some of the radiocarbon record is related to production. Are you able to include variable production in your model? One could use the results of Hain et al., 20148.

The model also fails to simulate the full increase in atmospheric CO₂ (missing about 20 ppm at 15ka) but seems capture the range in 13C and 14C. The missing CO₂ is significant at 50% of the

total rise. The authors should discuss this shortcoming. The best source to explain CO₂ and δ¹³C would be from rising SST or CaCO₃ feedbacks. Perhaps the authors should check their globally-averaged SST against Shakun et al., 2012? Changes in iron fertilization have been widely-discussed as important in interval but are completely ignored in this study. This effect wouldn't help the authors as it would cause δ¹³C to decrease even further. Therefore the results may limit the role of iron fertilization. However, I feel it is important to at least mention that this is possible source that is not treated in the model.

I am surprised to not see a close comparison with the δ¹⁴C coral data from Burke et al., 2012 and Chen et al. 2015. If the resolution is there, I would think these data provide a stronger test to the AAIW ventilation hypothesis than the deep sites currently used. At the moment the data model comparison only revisits an idea about deep ventilation in the comparison which is well-established. Also, I see that they are included in the Extended Data Table 3 and there is one data point at ~56S and 2000 meters depth Extended Figure 1, but the Drake Passage Coral data are mostly from 1000 meters and as far south as 60S. Is some data missing?

Trigger for the deglaciation?

"Through its impact on both atmospheric CO₂ and temperature, enhanced Southern Ocean ventilation could have thus initiated the last deglaciation."

This is minor conclusion that the authors have not oversold, but I am wonder if it is fully justified. To show that these changes are capable of sustaining atmospheric CO₂ and global temperature (or at the very least Antarctic temperature), I would have liked to see the model simulate the CO₂ plateau during the Bolling-Allerod and Antarctic Climate Reversal. At the moment the model run ends without a resumption of AMOC (stopping about 15ka). If the run was continued and CO₂ started to fall back to glacial levels it would fail to support this conclusion. The same critique could be made of Schittner et al., and many other results and (in my opinion) is a major problem for the models. If, on the other hand, the model was successful in sustaining CO₂ during the ACR, then it appears to have made a major breakthrough.

Terrestrial carbon

I could only see from one table in the Extended Data (Table 1) that the model was run with a terrestrial biosphere model. As far I as can tell this is not mentioned in the main text nor is the table referenced anywhere. Please convey this information to the reader and, if only just briefly, discuss how the terrestrial component acts as sink. How confident are the authors in the accuracy of the terrestrial model? A timeseries of the terrestrial and oceanic inventories would also be useful as seems standard for these type of modelling papers. The paper also opens with questions about the possibility of a terrestrial dominated source for the early deglacial CO₂ rise (Crichton et al. 12) but then doesn't address this question directly. This study appears to say completely the opposite of Crichton. Why is this the case? It also doesn't support the terrestrial hypothesis for the 16.3ka event from Bauska et al., which should also be mentioned. Can the model rule out these hypotheses or are the available experiments incapable of testing this (e.g. is there permafrost carbon in LOVECLIM to test Crichton or are there physical realistic changes in the precipitation during Heinrich Events to test Bauska?)

Line-by-line comments

Title: I feel that the title may be too strong but I am willing to be convinced otherwise. It seems like the westerly wind hypothesis will remain a hypothesis (albeit a much stronger one after this study). Maybe something like "SH westerlies as a driver of early deglacial...". Or something about the rapid ventilation of AAIW which is the more novel component of the study.

Abstract: I would suggest using a term other than "jumps" to describe the rapid increases in atmospheric CO₂ (despite the fact that I see the term used once in Marcott et al. 2014). Also, lumping the CO₂ rise from ~17.5 to 16.5 with the truly rapid rise at 16.3 ka with the same term

"jump" goes against the previous characterization.

Line 48. "So far, no 3-dimensional transient simulation..." The Schmittner et al., 2015 study requires this statement to be a bit more nuanced. If you provide a description and reference to that paper then you can easily make your case.

Lines 59-66. This reiterates some of the information in the abstract whilst also jumping to some conclusions. I am not sure it is totally necessary and might be a better place for some motivation for the study.

Line 133. Revelle factors should be explained in a more generic form in the main body. Also, the section on Revelle factors in the methods doesn't actually use the term "Revelle factors".

Line 143 and Extended Figure 6. The presence of a polynya is not very clear from the figures (an arrow in the time-series plot of sea ice concentration would suffice). The impact of the polynya in the spatial plots was also not clear to me. Note that some readers won't be familiar with the location of the Weddell Sea.

Lines 159-166. These statements need to be clearly denoted as either model results and conjecture. I worry they are too easily mixed at the moment.

Lines 167-172. This is a very "arm-wavy" paragraph. First the reference for the "Iberian Margin" is incorrect. The Martrat et al., 2014 reference is from the Alboran Sea. If the reference should be Martrat et al. 2007 for core MD01-2444 then I don't see evidence for the "two-phase cooling". As argued above, I view the ice core and speleothem data as a more direct link the ITCZ movement.

Line 201: A small question: how and why is global alkalinity held constant? Does this mean the LOVECLIM is incapable of CaCO₃ feedbacks? It is worth mentioning this even though they would have a relatively small effect over the timescale of interest. This would prevent spinning-up to an accurate LGM state.

Overall I thought the paper was very well written and it was a pleasure to read.

Figure 2:

It may just be me but I found the figure hard to follow. There is critical information presented in the d13C anomalies but also lots of generalized "arm waving" in the schematics. If the model is truly doing all this (which is the novelty of the study vs. data/idea paper) why couldn't this be conveyed more quantitatively? For example, could the wording and symbols about the d13C be replaced by subplots of the anomalies in the heat transport, SST, and sea ice over the Southern Ocean (with shading to indicate a zoom to the d13C plot)? Also, I wasn't sure what some of the arrows (like the red arrows) and symbols (like the ++) are referring to exactly.

Figure 3:

The meltwater and windstress forcing curves are bit hard to read. What does "calculated winstress" refer to? The DIC anomalies need units on the labels (umol/L?). "Taylor Dome" should be changed to "Taylor Glacier".

Figure 4: Unit labels appear missing on some graphs.

Extended Data:

Figure 1: Needs unit labels.

Figure 3: I think all the section plots in the paper are missing units. The tick marks could also be improved by making the bigger and higher fidelity.

Figure 5: The "tongue" of AAIW in the Atlantic looks pretty weak for DIC and ALK. Has this been discussed in the previous model work? Could it be problematic for your interpretation of the 16.3

ka CO₂ which relies on the interplay between DIC and ALK in these waters? If it hasn't been done before it would be useful to a cross-plot of DIC and ALK comparison of the model and data from the major water masses.

Thank you for considering my comments.

References:

- (1) Marcott, S. A.; Bauska, T. K.; Buizert, C.; Steig, E. J.; Rosen, J. L.; Cuffey, K. M.; Fudge, T. J.; Severinghaus, J. P.; Ahn, J.; Kalk, M. L.; McConnell, J. R.; Sowers, T.; Taylor, K. C.; White, J. W. C.; Brook, E. J. *Nature* 2014, 514 (7524), 616–619.
- (2) Rhodes, R. H.; Brook, E. J.; Chiang, J. C. H.; Blunier, T.; Maselli, O. J.; McConnell, J. R.; Romanini, D.; Severinghaus, J. P. *Science* 2015, 348 (6238), 1016–1019.
- (3) Bauska, T. K.; Baggenstos, D.; Brook, E. J.; Mix, A. C.; Marcott, S. A.; Petrenko, V. V.; Schaefer, H.; Severinghaus, J. P.; Lee, J. E. *Proceedings of the National Academy of Sciences* 2016, 113 (13), 3465–3470.
- (4) Tschumi, T.; Joos, F.; Gehlen, M.; Heinze, C. *Climate of the Past* 2011, 7 (3), 771–800.
- (5) Lee, S.-Y.; Chiang, J. C. H.; Matsumoto, K.; Tokos, K. S. *Paleoceanography* 2011, 26.
- (6) Schmittner, A.; Lund, D. C. *Clim. Past* 2015, 11 (2), 135–152.
- (7) Lauderdale, J. M.; Williams, R. G.; Munday, D. R.; Marshall, D. P. *Climate Dynamics* 2017, 48 (5), 1611–1631.
- (8) Hain, M. P.; Sigman, D. M.; Haug, G. H. *Earth and Planetary Science Letters* 2014, 394 (Supplement C), 198–208.
- (9) Shakun, J. D.; Clark, P. U.; He, F.; Marcott, S. A.; Mix, A. C.; Liu, Z.; Otto-Bliesner, B.; Schmittner, A.; Bard, E. *Nature* 2012, 484 (7392), 49–54.
- (10) Burke, A.; Robinson, L. F. *Science* 2012, 335 (6068), 557–561.
- (11) Chen, T.; Robinson, L. F.; Burke, A.; Southon, J.; Spooner, P.; Morris, P. J.; Ng, H. C. *Science* 2015, 349 (6255), 1537–1541.
- (12) Crichton, K. A.; Bouttes, N.; Roche, D. M.; Chappellaz, J.; Krinner, G. *Nature Geoscience* 2016, 9, 683.

Reviewer #3:

Remarks to the Author:

The paper provides some interesting insights into the characterisation of the last deglaciation, especially the role of southern hemisphere westerlies in enhancing CO₂ outgassing. The observed strengthening of SH westerlies in recent decades is predicted to increase under future climate scenarios and therefore the results are timely and likely to appeal to a wide audience.

Enhanced air-sea gas exchange and upwelling of old carbon-enriched deep waters in the southern ocean has been suggested as a driver of the rapid jumps in $\delta^{13}\text{C}$ observed in ice cores (eg Buaska et al., 2016; Schmitt et al., 2012). Modelling studies have demonstrated a step-wise strengthening of Southern ocean wind stress and produced approximately similar changes in ocean water $\delta^{13}\text{C}$ (Tschumi et al., 2011).

The layout of this study is convincing and provides testable theories for the changes observed in the proxy records. It highlights the importance of shifting westerly winds in driving changes in CO₂ and the findings support and enhance previous studies. However, the mechanisms for the CO₂ jumps proposed in this study are not necessarily new (eg proposed and modelled by Tschumi et al., 2011) and build on previous modelling studies about the role of westerly winds (eg Lauderdale et al., 2013 and others).

General comments:

Previous studies have also attributed the CO₂ increases to other sources. Such as precipitation changes in the SH and enhanced dust or changes to the biological pump. Can the authors

comment on these other hypothesis. Are they important in your scenarios?

Looking at the changes in ventilation age there is considerable mismatch between the simulated and observed (sediment cores) anomalies (fig 1 extended data). I know a perfect match cannot be expected but can this be explained? Are the values within the dating uncertainties of the sediment records?

Figures:

Fig 2. This figure is fundamental to explaining the sequence of events and needs to be expanded to make the processes clearer. What do the dots in circles signify?

Could the sea-air pCO₂ anomalies plot (extended data 4d) be included here? Replacing the central map to show the enhanced out gassing around the southern ocean under this scenario and how this relates to the ocean $\Delta\delta^{13}\text{C}$ depth profiles? It may make the plot too complicated but it could be helpful to tie the results spatially with the main conclusions of the paper.

Caption correct (r) and "shifts"

Fig 3. This is a really busy plot and needs to be separated or reformatted. The simulated DIC anomalies would work better side by side, eg LH1 Atlantic (left) and Pacific (right). To allow for easier comparison of changes in both oceans during the same scenario. As done in extended fig 1.

Plots a-g need to have more space to allow clearer separation of colours. I am not sure what plot h adds to the figure. Perhaps removing this could provide more space for the other timeseries?

Figure 4 – is it possible to plot the DIC anomalies on the same scale?

What is the relevance of 1320 m and 4346 m depths? It seems very specific and is quoted as 4300 m depth in the text.

Extended data 6 – just being picky but could you crop plot e to produce a circular plot?

Minor comments:

Line 42 (and others) – formatting error (h)? ‰

Line 77 – sentence starting via? Perhaps reword

Line 103 – move "weak" in front of AAIW

Line 186 – "ran"

Line 208 – "as follows"

Missing refs:

T. Tschumi, et al, Deep ocean ventilation, carbon isotopes, marine sedimentation and the deglacial CO₂ rise. *Clim. Past* 7, 771 (2011). doi:10.5194/cp-7-771-2011

Schmitt et al, Carbon Isotope Constraints on the Deglacial CO₂ Rise from Ice Cores *Science* (2012) DOI: 10.1126/science.1217161

Lauderdale, J.M., Garabato, A.C.N., Oliver, K.I.C. et al. *Clim Dyn* (2013) 41: 2145.

<https://doi.org/10.1007/s00382-012-1650-3>

Key=

Black= Reviewers' comments

Blue= Authors' responses

Green = Modified text in the manuscript

Reviewer #1 (Remarks to the Author):

We thank the Reviewer for their helpful comments, which contributed to improving our manuscript. We have expanded and amended our manuscript and we hope that the Reviewer will find the new finding clearer. In addition to the technological advances presented in this study, our work highlights the multi-decadal impact of SH westerlies on the global carbon cycle due to ventilation of intermediate waters. In the following we provide a point-by-point response to the Reviewers' comments in blue as well as excerpts from the manuscript showing the changes made in green.

This paper uses a simplified coupled climate model to study the response of the ocean, atmosphere and carbonate system to fresh water forcing. The objective is to simulate and understand the observed CO₂ increase in a couple of steps during the last deglaciation, 17-16,000 years ago. The subject is relevant and very interesting, and a coupled ocean-atmosphere model that incorporates an active carbonate system and tracks isotopic composition seems the right tool. The general idea that the Southern Ocean participated in causing glacial CO₂ changes, and that this was due to the response of the ocean biogeochemistry to changes in ocean circulation and mixing, is not new (the authors do only partial justice to previous works on the subject). This implies that the paper is to be judged to a significant degree based on its technical details, rather than as a representation of a completely new idea. Given this, I am afraid I did not find the case made by the paper to be strong due to several issues, making it difficult to recommend publication in Nature communications.

1) Adequacy of model formulation:

A) The general approach followed in the paper is to prescribe fresh water forcing in the northern hemisphere, presumably due to the deglaciation that started there for reasons that are beyond the scope of this paper, and observe the response. This response involves changes to the atmospheric inter-tropical convergence zone, leading to a response of the southern hemisphere winds, and then to a response of the Southern Ocean circulation and stratification, and finally to a response of the Southern Ocean biogeochemistry. However, the atmospheric model used here is a highly simplified 3-level quasi-geostrophic (QG) model. While simple models are extremely helpful in understanding the behavior of both the climate system and of more complex models, it seems that such a simplified model would not be able to realistically simulate the response of the atmospheric tropical convergence zones, which also puts in doubt the rest of the chain of arguments. Simulating these zones requires a reasonable representation of atmospheric convection, clouds and radiation feedbacks that a 3 level QG model seems incapable of including. In addition, the QG approximation breaks down at the equator, and it is therefore not clear how such a model can simulate the near-equatorial ITCZ atmospheric response. While the simple model

used here can probably be tuned more easily to fit the observed record, it is not obvious what are the robust lessons the resulting simulation can provide.

We understand the concerns of the reviewer with respect to the reduced complexity of the atmospheric model used. In response to the North Atlantic cooling obtained by weakening the AMOC, the model simulates a southward shift of the ITCZ, particularly over the Atlantic basin, that is in overall agreement with that simulated using full AOGCMs (see for example Kageyama et al., 2013 figures 4 and 5). We have now added a supplementary figure showing precipitation changes in three experiments as well as contours of annual mean precipitation changes in Figure 2.

Regarding the link between ITCZ changes and Southern Hemisphere westerlies, our reasoning also comes from other experiments performed with more complex atmospheric models. For example, Lee et al. (2010) use the CCM3, an atmosphere general circulation model with a resolution of T42, while Ceppi et al., (2013) use the GFDL AM2.1 and the ECHAM4.6 at T42. Results of these two studies show how a North Atlantic cooling shifts the ITCZ southwards, leading to a weakening of the Southern branch of the Hadley cell and leads to a stronger subtropical jet. This in turns shifts the eddy-driven jet poleward and intensifies the SH westerlies.

However, in our study, we are not testing the atmospheric teleconnection in itself because of the quasi-geostrophic approximation present in the atmospheric model used and its coarse resolution. Instead, we examine the impact on ocean biogeochemistry and thus $p\text{CO}_2$ of different changes in ocean circulation in response to carefully applied wind and freshwater anomalies. The goal is to understand the drivers of the atmospheric CO_2 increase during HS1. It is thus really the ocean dynamics and the biogeochemical component that will determine the changes in $p\text{CO}_2$ and its isotopes.

B) Similar issues, although if less extreme, seem to exist with the ocean model, with its coarse vertical resolution (20 levels), given the need to represent different water masses at different ocean levels, and vertical convection and mixing processes that are an important part of the proposed mechanism.

We understand the concerns of the Reviewer. However, the model we are using is currently the most complete model including carbon isotopes able to run long paleo simulations. And in fact, we are presenting the first transient simulation of Heinrich 1 performed with a carbon-isotopes enabled Ocean General Circulation model. As we are very much aware of potential issues with the coarse resolution, we also test our hypothesis with a very high-resolution, state of the art model: the MOM5-WOMBAT model with a horizontal resolution of $\frac{1}{4}$ degree and 50 vertical levels. This simulation confirms the impact of the Southern Hemisphere westerlies on AABW and AAIW as well as the impact on the carbon cycle.

C) Lines 70-71: "The LGM state is characterised by shallow NADW, relatively weak North Pacific Intermediate Water (NPIW) and very weak AABW, obtained by adding a melt water flux into the Southern Ocean and weakening the SH windstress by 20%." - This seems to imply artificial fresh water forcing and artificial wind stress modification, both imposed in the Southern Ocean, in order to overcome what the authors considered

model biases. But, of course, introducing such biases can lead to compensating model errors that can then influence the results of the forced simulation as well and thus influence the proposed mechanism. This seems reminiscent of the flux correction procedure that is long-gone in climate modelling.

In Menviel et al. (2017), a set of LGM experiments (24 experiments all up) was performed to identify the oceanic $\delta^{13}\text{C}$ distribution that was the closest to the proxy record. This previous study showed that the most plausible ocean state included weaker AABW. Weaker AABW leads to reduced deep ocean ventilation and increased remineralized carbon content in the deep ocean, in agreement with benthic $\delta^{13}\text{C}$ records and ventilation age estimates. In addition, weaker southern hemisphere westerlies further reduce the air-sea gas exchange and deep ocean ventilation. The impact of changes in NADW on the ocean carbon cycle was previously studied by the authors starting from different initial conditions: pre-industrial and LGM but without the artificial freshwater and wind forcing (e.g. Menviel et al., 2008, Menviel et al., 2014, Menviel et al., 2015). We note that while the initial state and boundary conditions can somewhat affect the magnitude of the atmospheric CO_2 changes, the direction of the change and the processes at play are robust.

D) I am confused whether the changes to the Southern Ocean winds are imposed by the authors (as suggested by Fig 3b), or are a response to other forcing (fresh water in north Atlantic) as the manuscript seemed to imply. Perhaps I simply misunderstood, but if the wind changes are imposed rather than simulated, this further weakens the case made by the paper.

The changes in Southern Ocean winds are directly imposed in the coupled ocean-ice-carbon experiments performed. This allows us to examine in detail the conundrum posed by the SH westerlies; namely, in LOVECLIM, a weakening of the AMOC weakens the annual mean SH westerly winds by 6%, despite a southward shift of the ITCZ and a weakening of the Southern branch of the Hadley cell. While the direction of the Hadley cell response in our model is correct, there is no significant expansion of the SH Hadley cell, contrary to what is suggested by other models (e.g. Ceppi et al., 2013, Drijfhout 2010). But as pointed out by the reviewer, since the weakness of LOVECLIM lies in its relatively simple and coarse resolution atmospheric model, it is hypothesised that the changes in the SH Hadley cell is underestimated thus not impacting the SH subtropical and eddy-driven jets correctly. The imposed wind change is added to the model in order to accurately evaluate its impact on the ocean and the carbon cycle.

In addition, while the response of the SH westerlies to future warming seems to be fairly well constrained by CMIP5 models (e.g. Chavaillaz et al., 2013), it is important to note that models diverge with respect to past changes in the SH westerlies. Our range of experiments presented in this manuscript show that the SH westerlies could have exerted a significant control on atmospheric CO_2 and climate during the early part of the deglaciation. Additional work is needed to further our understanding of drivers of SH westerlies.

To be clearer regarding our experimental design, the text has been modified as follows (L.113-123): "Cessation of NADW formation induces a southward shift of the Intertropical Convergence Zone (ITCZ) (Fig. 2) via reduced meridional heat transport to the North

Atlantic. Modelling studies (Ceppi et al., 2013, Lee et al., 2010) have shown that this ITCZ shift could weaken the SH Hadley cell and strengthen the subtropical jet, which in turn would shift the eddy-driven jet poleward and strengthen the SH westerlies by ~25%. To include this atmospheric teleconnection between the tropics and the high southern latitudes, which is not well represented in our coarse resolution atmospheric model, the SH westerlies are increased from their LGM state commencing at 17.2 ka. A further intensification of the SH westerlies and reduced Southern Ocean freshwater flux at 16.2 ka enhances Southern Ocean convection in our simulation, resulting in stronger AAIW and AABW formation during HS1 (Fig. 3,d,e), consistent with reduced ventilation ages in the South Atlantic (Skinner et al., 2010) and the Pacific (Siani et al., 2013) (Fig. 1g,h) as well as a peak in the Southern Ocean opal flux (Anderson et al., 2009)."

2) Interpretation of the results:

A) Lines 165+: "Through its impact on both atmospheric CO₂ and temperature, enhanced Southern Ocean ventilation could have thus initiated the last deglaciation."

- It is not clear precisely how ice sheets are updated during the model run (there is a reference in the methods section to Peltier's reconstruction, but no details). Assuming that they are modified in time, and given that fresh water forcing is changed by the authors in the northern hemisphere and the Southern Ocean merely responds to these forcings, I don't see how the SO could be considered the initiator of deglaciation.

This sentence was confusing so we modified it as follows (L. 292-293):

"Our simulations suggest that through its impact on both atmospheric CO₂ and temperature, enhanced Southern Ocean ventilation played a significant role during the last deglaciation."

We have also now modified the methods description to clearly indicate that the model is forced by time varying changes in Northern hemisphere ice-sheet extent, topography and albedo:

"A suite of transient experiments is performed starting from this LGM state by forcing the model with time-varying changes in orbital parameters (Berger, 1991) and Northern Hemispheric ice-sheet extent, topography and albedo (Peltier et al., 1994)."

B) Lines 146-147: "Open-ocean convection was likely the dominant source of AABW formation during glacial periods, in contrast to modern climate where the frequency of large-scale Weddell Sea polynya events is uncertain."

- The logic seems problematic here: how can we be confident about the state of Antarctic water mass formation during glacial periods if there is a large uncertainty even at present?

Today's main AABW formation regions are the Weddell and Ross Seas. Modelling work and paleo-data have shown that the Weddell and Ross Seas were most likely covered by grounded ice during the LGM, thus preventing AABW formation in these Seas. If AABW was formed it had to be through open ocean convection. We have rephrased this sentence to make this point more clearly (L.252-254):

“Open-ocean convection was likely the dominant source of AABW formation during glacial periods because grounded ice covered most of the Weddell and Ross Seas (Golledge et al. 2014), where AABW is primarily formed today.”

C) Reason for DIC changes: the carbonate system response is described as involving changes to the Dissolved Inorganic Carbon concentration, but the mechanism is not clearly explained. In addition, is ocean biology (productivity) playing a role?

Previous figure 3h, which is now figure 5, shows the processes leading to the atmospheric CO₂ increase, including changes in SST, SSS, DIC and Alkalinity. The oceanic DIC anomalies are now shown in Figure 2 as well as figure S5 in all the experiments. These figures and Table S1 show that the atmospheric CO₂ increase is due to a loss of carbon from the ocean, resulting from a DIC transfer from the deep and intermediate depth of the Ocean (particularly Indo-Pacific) into the surface mixed layer. This is mostly due to enhanced upwelling of carbon-rich waters in the Southern Ocean. Due to warmer conditions and globally enhanced oceanic circulation, the global export production increases significantly during HS1 thus providing a negative feedback to the atmospheric CO₂ increase. Changes in preformed and remineralized phosphate are shown in Figure S3 and are now discussed in the text.

We have added L. 180-186:

“A further weakening of the oceanic circulation from the LGM into HS1 as simulated in experiment LH1 only leads to small changes in the carbon reservoirs, with a slight increase in the global remineralized PO₄ content, thus implying a decrease in the preformed PO₄ content (Fig. S3). In contrast, enhanced Southern Ocean ventilation leads to an oceanic carbon loss associated with a decrease in remineralized PO₄ and increase in preformed PO₄, thus implying a reduced efficiency of the biological pumps in experiments LH1-SO, LH1-SHW and LH1-SO-SHW.”

The text L.201-203 now reads:

“The CO₂ rise in the wind forced simulation is due to an imbalance between DIC and alkalinity (ALK) increase (Fig. 5, blue diamond). As seen in Figure 2, stronger AABW and enhanced upwelling in the Southern Ocean lead to a DIC transfer from deep and intermediate depths of the Pacific and Southern Oceans up to the surface of the Southern Ocean.”

D) Lines 90-92: "Finally, due to the concurrent atmospheric CO₂ rise and enhanced meridional heat transport towards high southern latitudes, Southern Ocean Sea Surface Temperature (SST) and Antarctic air temperature increase by up to 3C and 4C, respectively..."

- It does not seem likely that such a small CO₂ change would play a dominant role in the significant 3 or 4C warming. This would indicate climate sensitivity beyond typically calculated values.

We agree with the reviewer that the pCO₂ change cannot explain the temperature increase on its own. This increase in high southern latitudes temperature is due to both changes in CO₂ and changes in oceanic circulation. Antarctic ice cores and marine sediment cores

suggest such a temperature increase during HS1. This has usually been explained by the concurrent effect of atmospheric CO₂ increase and “heat piracy” due to NADW cessation. Here we show that stronger AABW would enhance the meridional heat transport to high southern latitudes.

The text now reads (L.288-293):

“Both stronger SH westerlies and buoyancy loss near the Antarctic continent act to steepen the Southern Ocean isopycnals thus increasing the southward baroclinic flow. This results in a stronger southward meridional heat transport (Meniel et al., 2015), decreasing Southern Ocean sea-ice and further warming the mid to high southern latitudes. Our simulations thus suggest that through its impact on both atmospheric CO₂ and temperature, enhanced Southern Ocean ventilation played a significant role during the last deglaciation.”

Regarding climate sensitivity, the typically cited climate sensitivity values are derived from 2xCO₂ anthropogenic forcing experiments. Past climate states will be associated with substantially different sensitivities than the present climate. The climate’s sensitivity to external forcing will depend on the mean climate state, and its feedback mechanisms (IPCC AR4, 9.6).

Also,

3) Figure 2: the fit between the observations (stars) and model results (shading) does not seem very good. It needs to be quantified and some measure of quality of fit needs to be used to make a convincing case that this is, indeed, a good fit.

In the legend of this figure (now Figure 4), we note that the correlation coefficient between model and data is $R=0.77$ with $p=0.01$. This is a fairly good fit for paleo-data/model comparisons.

4) It seems that figure 3 would best be divided into 3 different and still fairly complex figures. This, with some other comments above suggest that a different venue that allows providing more details about the study and using more figures may be a more appropriate choice.

We thank the reviewer for this suggestion and we have now split figure 3 into 3 separate figures.

Reviewer #2 (Remarks to the Author):

We thank the reviewer for this thorough and very helpful review. As detailed below we have taken into account all of the comments, generally by adding significant information to the introduction and discussion, by modifying the figures of the main text, and/or by adding some additional figures to the Supplementary information. In the following we provide a point-by-point response to the Reviewers' comments in blue as well as excerpts from the manuscript showing the changes made in green.

Review of "Abrupt early deglacial atmospheric CO₂ increase driven by Southern Hemisphere westerlies"

Menviel et al., submitted to Nature Communications

The manuscript by Menviel et al. provides exciting modelling results that tackle important questions regarding the drivers of glacial-interglacial CO₂. The paper tests a widely discussed but somewhat untested hypothesis for glacial-interglacial CO₂ by demonstrating a clear mechanistic link between Southern Hemisphere westerlies and atmospheric CO₂ during the last glaciation. The paper also proposes a completely new and exciting mechanism for centennial-scale changes in atmospheric CO₂ that involves ventilation of Antarctic intermediate waters. The paper is particularly timely as it is one of the first modelling studies that incorporates new ice core records in a thought-provoking method. It also builds upon a solid foundation of previous carbon cycle modelling with LOVECLIM. I believe the paper meets the requirements of Nature Communications in terms of novelty and quality (after some major revisions).

I have a few comments and suggestions that are likely addressable but I feel amount to a major revision. Most are related to providing a more detailed and clear description of the model experiments and greater discussion of previous work in the main body of the text. Given that the paper seems very short (main body = 1900 words) compared to the word limit (5000 words), I suspect this will not be a problem. Overall, I thought the paper could have benefited from more in-depth discussion, so hopefully a more generous word count will help.

My comments will focus on the interpretation of the data as I am not qualified to discuss the technical details of the modelling.

Description of model forcing and the motivation behind the experiments

The biggest difficulty I had understanding the results was understanding why and exactly how the model had been forced. Just as an example, during my first casual reading of the paper I was stuck by the close between the modelled atmospheric CO₂ and data in terms of timing – most notably the 16.3 ka event. I was initially under the impression that this "event" was an unforced variation, but upon additional reading I became unsure.

In the first paragraphs (line 67 through line 97) the model description reads "By forcing the model with changes in orbital parameters, Northern Hemisphere ice-sheet extent and albedo as well as freshwater input in the North Atlantic (Methods, grey line Fig. 3a) during HS1....". The description then goes into model predictions that are a consequence of these

forcings. I took this to mean that these were the only forcings applied to the model and the following paragraphs were on simulated variables. The next paragraph continues “Via a southward shift of the Intertropical Convergence Zone (ITCZ), the North Atlantic cooling can strengthen and shift the SH westerlies poleward^{23, 24} (Fig. 2). Intensified SH westerlies and reduced Southern Ocean freshwater flux at 17.2 and 16.2 ka enhance Southern Ocean convection in our simulation, resulting in stronger AAIW and AABW formation during HS1...”

I found this language very ambiguous. Is a shift in the ITCZ observed in the model because the North Atlantic cooling or is this more general background information? Note that Figure 2 only shows some qualitative indicators of ITCZ and westerlies.

The model indeed simulates a southward shift of the ITCZ in response to the North Atlantic cooling. In the new Figure 2, we now include contours showing annual mean changes in precipitation in the tropics. We also now include a supplementary figure (S1) showing the simulated changes in precipitation during HS1. Finally, in Figure S2, simulated precipitation changes in Brazil and China as well as a comparison with the Paixao, Qingtian and Hulu speleothems have been added.

The text has also been changed as follows (L.113-123):

“Cessation of NADW formation induces a southward shift of the Intertropical Convergence Zone (ITCZ) (Fig. 2) via reduced meridional heat transport to the North Atlantic. Modelling studies (Ceppi et al., 2013, Lee et al., 2010) have shown that this ITCZ shift could weaken the SH Hadley cell and strengthen the subtropical jet, which in turn would shift the eddy-driven jet poleward and strengthen the SH westerlies by ~25%. To reflect this atmospheric teleconnection between the tropics and the high southern latitudes, which is not well represented in our coarse resolution atmospheric model, the SH westerlies are increased from their LGM state commencing at 17.2 ka. A further intensification of the SH westerlies and reduced Southern Ocean freshwater flux at 16.2 ka enhances Southern Ocean convection in our simulation, resulting in stronger AAIW and AABW formation during HS1 (Fig. 3,d,e), consistent with reduced ventilation ages in the South Atlantic (Skinner et al., 2010) and the Pacific (Siani et al., 2013) (Fig. 1g,h) as well as a peak in the Southern Ocean opal flux (Anderson et al., 2009).”

Furthermore, and this is vital to the whole paper, have the intensified westerlies and freshwater water fluxes in the SH forcings been chosen by the authors or did they emerge as a consequence of other forcings? Delving in the sensitivity experiments and Figure 3, it seems to me that the former is correct, which means that some of the agreement between the model and ice core data is artificial. If so, this needs to be stated explicitly. Also, the motivation for changing westerlies in the series of rapid steps should be outlined before the experiments are described. At the moment, there is some justification but it is only discussed very late in the paper (lines 167-172). I feel the paper would benefit from greater discussion of the motivation in the introduction (there is a narrative that describes NADW shutdown, the bipolar seasaw and ACC, but nothing about the westerlies). It would be a much stronger paper if the change in the westerlies could be derived in an objective way.

Are the authors justified in forcing the model in this way? I have one suggestion for

providing this motivation that would provide a more in-depth discussion of the previous hypotheses for the centennial-scale CO₂ variability during HS1, which I also think is necessary.

Marcott et al., 2014¹ hypothesized that the 16.3 ka event could ultimately be a consequence of Heinrich Event 1 “We propose that the abrupt rise in CO₂ at 16.3 kyr ago is directly related to an iceberg discharge event recorded in several ocean cores from the North Atlantic.... One potential cause of the CO₂ increase at this time could be a southward shift in the position of the Southern Hemisphere westerlies²⁴, leading to increased ocean upwelling and outgassing of respiration-derived CO₂”. By analyzing WAIS Divide CH₄ record Rhodes et al., 2015² found support for the idea of a southward shifted ITCZ during Heinrich Stadials (right at 16.3 ka during the deglaciation). Bauska et al., 2016³ used the carbon isotopes of CO₂ to propose that the 16.3 ka event could either be a terrestrial source (from a drying of the NH associated with the southward shifted ITCZ) or an increase in air-sea gas exchange in the Southern Ocean. “The southward migration of the ITCZ also led to drying in parts of the NH, possibly causing a reduction in organic land carbon, most notably around 16.3 ka. Alternatively, or additionally, the changing SH westerlies reached a threshold around 16.3 ka, in which wind speed over the SO increased, leading to enhanced air–sea gas exchange and possibly greater upwelling.”

At the moment, these studies are used only for the data and not their ideas but to me they provide the perfect motivation for these experiments. It is also important to acknowledge that this hypothesis is already out there in the literature (even though you will prove the Bauska air-sea gas exchange hypothesis wrong with your superior model and may be able to test the terrestrial hypothesis - see more below).

We thank the reviewer for the very helpful comments and suggestions. We have added some text in the introduction to briefly describe previous modelling work and expand the motivation of the study (L.65-88):

“A number of processes have been put forward to explain the early deglacial atmospheric CO₂ increase. Co-variation of iron flux and nutrient utilisation in the sub-Antarctic (Martinez Garcia et al., 2014) suggest that through its impact on the Southern Ocean biological pump, iron fertilization could exert a significant impact on atmospheric CO₂ during HS1. Modelling studies show that reduced iron fertilization would lead to a millennial atmospheric CO₂ increase of ~10 ppm coupled to a 0.1 permil δ¹³C_{CO₂} decrease (Bopp et al., 2003, Tagliabue et al., 2009, Menviel et al., 2012, Bauska et al., 2016), during the early part of the deglaciation. However, changes in iron fertilization would not lead to any changes in atmospheric Δ¹⁴C or ocean ventilation ages, and changes in Southern Ocean dust flux are ultimately controlled by SH hydrology and winds. Indeed, a deglacial decline in South Atlantic ventilation ages has been observed (Skinner et al., 2010), indicating a possible role of Southern Ocean ventilation in driving the deglacial CO₂ increase. This change in Southern Ocean ventilation could be modulated by SH westerlies (Toggweiler et al., 2006, Anderson et al., 2009, Marcott et al., 2014). Idealized modelling studies performed under constant pre-industrial boundary conditions have shown that stronger or poleward shifted SH westerlies could enhance deep ocean ventilation thus leading to an atmospheric CO₂ increase and δ¹³C_{CO₂} decrease (Tschumi et al., 2011, Lauderdale et al., 2013, Menviel et al., 2015). However, this hypothesis has been challenged by a recent modelling study, also performed under constant

pre-industrial boundary conditions, which concluded that changes in SH westerlies did not lead to significant changes in atmospheric CO₂ during HS1 (Schmittner and Lund, 2015). Instead, Schmittner and Lund (2015) suggested that the HS1 atmospheric CO₂ rise was solely due to a reduced efficiency of the biological pump resulting from a weaker Atlantic Meridional Overturning Circulation, even though no abrupt atmospheric CO₂ rise was simulated. In addition, while a strengthening and poleward shift of the SH westerlies seem to be a robust feature of future projections (Zheng et al., 2013), the behaviour of SH westerlies on glacial-interglacial and millennial timescales remain unclear (Chavaillaz et al., 2013, Kohfeld et al., 2013).

As the magnitude and rate associated with an oceanic carbon release to the atmosphere during the deglaciation have been questioned, a deglacial transfer of carbon from the terrestrial to atmospheric reservoir has been put forward, either as thawing of Northern Hemisphere permafrost (Crichton et al., 2016) or as a Northern Hemisphere terrestrial carbon release (Rhodes et al., 2015, Bauska et al., 2016) due to a southward shift of the Intertropical Convergence Zone (ITCZ). However, the terrestrial carbon reservoir globally increased over the deglaciation (Ciais et al., 2011) and the timing and magnitude of the permafrost carbon contribution remains poorly constrained. “

We also more explicitly state the goals of the study (L.95-101):

“Here, we explore the processes leading to the two-stage CO₂ increase during HS1, and their links to NADW weakening and Antarctic warming by performing a suite of transient experiments of HS1 with the carbon-isotope enabled Earth System Model LOVECLIM (Menviel et al., 2015). This suite of simulations assesses the impact of changes in Southern Ocean ventilation during HS1, including the potential role of buoyancy and dynamic forcing, such as meltwater and SH westerlies, in driving the rapid atmospheric CO₂ increase during HS1. The impact of SH westerlies on ocean carbon is further assessed in an eddy-permitting ocean model.”

We have also modified the text describing the experiment to make our experimental set-up and motivations clearer (L. 113-123).

“Cessation of NADW formation induces a southward shift of the ITCZ (Fig. 2) via reduced meridional heat transport to the North Atlantic. Modelling studies (Ceppi et al., 2013, Lee et al., 2010) have shown that this ITCZ shift could weaken the SH Hadley cell and strengthen the subtropical jet, which in turn would shift the eddy-driven jet poleward and strengthen the SH westerlies by ~25%. To reflect this atmospheric teleconnection between the tropics and the high southern latitudes, which is not well represented in our coarse resolution atmospheric model, the SH westerlies are increased from their LGM state commencing at 17.2 ka. A further intensification of the SH westerlies and reduced Southern Ocean freshwater flux at 16.2 ka enhances Southern Ocean convection in our simulation, resulting in stronger AAIW and AABW formation during HS1 (Fig. 3,d,e), consistent with reduced ventilation ages in the South Atlantic (Skinner et al., 2010) and the Pacific (Siani et al., 2013) (Fig. 1h,i) as well as a peak in the Southern Ocean opal flux (Anderson et al., 2009).”

While the imposed changes in SH westerlies are not derived in a quantitative way, a mechanism to explain these changes is proposed. In particular, we propose that a first

change in SH westerlies occurs during the early part of HS1, when the AMOC reaches its minimum. AMOC cessation leads to a North Atlantic cooling, a southern shift of the ITCZ in the Atlantic sector (simulated by the model) and enhanced SH westerlies (simulated by other studies: Ceppi et al., 2014; Lee et al., 2011). We further propose that the 16.2 ka event is due to changes in NPIW: melting of NH ice-sheets weakens NPIW formation, cooling the North Pacific and further shifting the ITCZ southward, particularly in the Pacific sector (also simulated by the model, Figures S1 and S2). This can in turn impact the SH westerlies. This mechanism is completely new.

Previous Modelling Studies

Some significant modelling work is currently ignored. How does this paper improve on the results of Tschumi et al., 2014 which tested the sensitivity of Bern3D model to westerly wind strength? Is the centennial-scale ventilation of AAIW present in Bern3D? Could this be a model dependent phenomenon or has it just been ignored or mischaracterized in previous work? There is also the work of Lee et al., 2015 with MEMSO that goes unmentioned, which may provide a stronger link between southward shift of the ITCZ and Southern Hemisphere westerlies (especially if the ITCZ shift is absent in the LOVECLIM model as discussed above).

As detailed in the previous comment, we have now expanded the introduction section to introduce previous modelling work, including the work by Tschumi et al. (2011). The experiments performed by Tschumi et al., (2011) were done under constant pre-industrial conditions and were done to study the equilibrium response to changes in SH westerlies (i.e. multi-millennial). The rate of change is not explored in any detail. Furthermore, while the Bern3D is a very useful model, it is of lower complexity than LOVECLIM. The ocean model is a coarse resolution, frictional geostrophic balance ocean model, and thus not an ocean general circulation model.

We do not have enough information from Tschumi et al (2011), nor from other work (including the first author's work with the Bern3D) on the impact of SH westerlies on AAIW in the Bern3D model. But in any case, given the lower complexity of the ocean physics resolved in the Bern3D, it is not the best model to test this mechanism. This is the reason we performed an additional experiment with the eddy-permitting full 3D ocean model MOM5-WOMBAT, which displays a strengthening of AAIW (as showed in Figure S9), as well as enhanced ventilation at intermediate depth (Figure 4c), on a very short time-scale (<50 years).

The work on atmospheric teleconnections of Lee et al., (2011), linking changes in the ITCZ to SH westerlies is mentioned several times in our work and is a very nice study linking North Atlantic cooling to changes in SH westerlies. However, MEMSO is a coarse resolution and low complexity ocean model. In addition, the experiment was performed under constant pre-industrial boundary conditions and without carbon isotopes. In our study, in contrast, we are testing different sets of changes in ocean circulation including cessation of NADW formation, changes in NPIW, changes in AABW due to buoyancy forcing and changes in the SH westerlies during the deglaciation and with a particular focus on carbon isotopes. We therefore cannot refer to all the sensitivity studies that have been done on the topic; we

thus focus on the most relevant previous studies, and pay less attention to the ones undertaken without any carbon isotopes.

It is also important to mention that an alternate hypothesis for the HS1 CO₂ was offered in Schmittner et al., 2015. These new results seem to contradict Schmittner despite using fairly similar model. For example the Schmittner abstract states “Modeled effects of AMOC-induced wind changes on the carbon and isotope cycles are small, suggesting that Southern Hemisphere westerly wind effects may have been less important for the global carbon cycle response during HS1 than previously thought.” What drives this model dependent response? Which model is better suited for simulating the paleodata? Schmittner didn’t spin up to a glacial-state (or at least a realistic one) so the Menviel et al., study is clearly a big improvement.

The introduction was modified and more background information was added, including detailing the work by Schmittner and Lund (2015) (L73-84):

“Indeed, a decline in South Atlantic ventilation ages through the last deglaciation has been observed (Skinner et al., 2010), indicating a possible role of Southern Ocean ventilation in driving the deglacial CO₂ increase. This change in Southern Ocean ventilation could be modulated by SH westerlies (Toggweiler et al., 2006, Anderson et al., 2009, Marcott et al., 2014). Idealized modelling studies performed under constant pre-industrial boundary conditions have shown that stronger or poleward shifted SH westerlies could enhance deep ocean ventilation thus leading to an atmospheric CO₂ increase and $\delta^{13}\text{CO}_2$ decrease (Tschumi et al., 2011, Lauderdale et al., 2013, Menviel et al., 2015). However, this hypothesis has been challenged by a recent modelling study, also performed under constant pre-industrial boundary conditions, which concluded that changes in SH westerlies did not lead to significant changes in atmospheric CO₂ during HS1 (Schmittner and Lund, 2015). Instead, Schmittner and Lund (2015) suggested that the HS1 atmospheric CO₂ rise was solely due to a reduced efficiency of the biological pump resulting from a weaker Atlantic Meridional Overturning Circulation, even though no abrupt atmospheric CO₂ rise was simulated.”

The Schmittner and Lund (2015) study was performed under constant pre-industrial boundary conditions using the UVic ESCM, and included two parts: one with constant wind forcing and the second part where the computed wind actually feeds back onto the ocean circulation. They concluded that changes in the wind had a small effect because there were small differences in simulated atmospheric CO₂ between the two parts. But this is most likely due to the small changes in simulated wind. In their simulation, the SH westerlies shift southward but their strength varies by less than 10%.

It is interesting to note that D’Orgeville et al. (2011) simulated a 36 ppm atmospheric CO₂ increase due to a doubling of the SH westerlies under constant pre-industrial boundary conditions with the UVic ESCM, showing that results from LOVECLIM and the Uvic ESCM are consistent in this regard.

It is also important to note that Schmittner and Lund (2015) simulate a relatively steady atmospheric CO₂ increase of 25 ppm over 2500 years, without any abrupt changes. Finally, based on figures 2 and 5 of Schmittner and Lund (2015), the loss of ocean carbon and

atmospheric CO₂ increase in their study is most likely due to enhanced formation of AABW and NPIW, in agreement with our results. This is also consistent with Menviel et al., (2014, Paleoceanography), who showed that the oceanic carbon responses of LOVECLIM and the UVic ESCM to changes in oceanic water-masses are consistent.

Eddy Permitting Model

The CO₂ prediction from the eddy-resolving method is really fascinating. Has the CO₂ increase been predicted with this type of model before? I am familiar with some work on CO₂ by Lauderdale et al., 2016 with MITgcm and an eddy-permitting sector model which may or may not be relevant. Also was the model forced only with windstress and no other 21st century changes (for example SST)? I think the language in the extended data is slightly ambiguous about the forcing.

As far as we know we are presenting a unique sensitivity experiment of changes in southern hemisphere westerlies in a coupled global ocean-carbon cycle eddy-permitting model. Via collaborations across the Centre of Excellence in Climate System Science in Australia, represented by the author group of this paper, a lot of work has gone into developing this model. The ocean-sea ice components of this model have been utilized in numerous (>10) published papers. This is the first study to incorporate the coupling with the biogeochemical model (WOMBAT). A significant effort has also gone into developing and assessing the biogeochemical model (see <http://cosima.org.au/index.php/models/mom-sis-025-wombat/>).

The control run and perturbed experiment differ only in the strength and position of the Southern Hemisphere westerlies. We have modified the sentence in the methods to clear out any confusion (L.386-388).

“The wind perturbation experiment includes a poleward intensifying wind forcing, namely a 4° southward shift and 15% increase in 10m wind speeds between 30S-65S (Fig. S7). This wind forcing is based on projected SH wind changes in CMIP5 business as usual scenarios (Zheng et al., 2013).”

See also Fig S3 of Spence et al. (2014, Geophysical Research Letters) for further justification of the wind perturbation used.

Critical Evaluation of the Paleodata

I am somewhat concerned that the model predicts a large drop in 14C at the 16.3 ka event that is not present in the data. Bauska suggests that the rapid increases could be terrestrial or enhanced air-sea gas exchange. The absence of a 14C drop in data might rule out westerly hypothesis for this event (a more contemporaneous carbon source would be a better fit if it was superimposed on a slower ventilation change). But maybe IntCal13 isn't high enough resolution or accurate enough to capture this event? Please include the IntCal13 underlying data with error bars or uncertainty on the reconstruction (<http://www.radiocarbon.org/IntCal13.htm>). The same goes for the ice core data, which are currently plotted as lines which implies continuous rather than discrete data.

Moreover, some of the radiocarbon record is related to production. Are you able to include variable production in your model? One could use the results of Hain et al., 20148.

The Intcal 2013 record is fairly smooth and even though the two-sigma uncertainty is not large, there might be significant uncertainty and smoothing included as it is a compilation of different records. In addition, given the short duration of the 16.2 ka event, it is quite possible that it is not recorded in the Intcal 2013 record. In Figure 1, we now include the Intcal 2013 +/-2 σ uncertainty. In a new figure S4, we are now showing all the different $\Delta^{14}\text{C}$ reconstructions available based on terrestrial and marine archives. In that figure, we have also added the simulated $\Delta^{14}\text{C}$ in three simulations similar to LH1-SHW-SO but with varying atmospheric ^{14}C production rates based on the study by Hain et al., (2015). Both simulations with a constant ^{14}C production rate and a “maximum” production rate following Hain et al., (2015) lead to $\Delta^{14}\text{C}$ variations broadly following the Intcal 2013 +/-2 σ uncertainty. It is true that the 16.2 ka event always falls outside of the 2 σ uncertainty, but as mentioned above, this could be due to the method applied in generating the Intcal 2013 record. It is also worth noting that permafrost release should also lead to a significant $\Delta^{14}\text{C}$ decrease as discussed in Koehler et al., (2015) for the Bolling/Allerod.

The model also fails to simulate the full increase in atmospheric CO₂ (missing about 20 ppm at 15ka) but seems capture the range in 13C and 14C. The missing CO₂ is significant at 50% of the total rise. The authors should discuss this shortcoming. The best source to explain CO₂ and 13C would be from rising SST or CaCO₃ feedbacks. Perhaps the authors should check their globally-averaged SST against Shakun et al., 20129? Changes in iron fertilization have been widely-discussed as important in interval but are completely ignored in this study. This effect wouldn't help the authors as it would cause d13C to decreases even further. Therefore the results may limit the role of iron fertilization. However, I feel it is important to at least mention that this is possible source that is not treated in the model.

We understand the concerns of the reviewer. Our simulations could underestimate changes in pCO₂ due to an overestimation of surface changes in alkalinity, as shown for example in Figure 5. An additional possibility for the underestimated pCO₂ increase is the change in global ocean alkalinity due to the deglacial sea-level rise and changes in carbonate compensation. As sea-level increases across the deglaciation, global ocean alkalinity should decrease, which is not taken into account in our standard experiment where total alkalinity is held constant. In addition, changes in oceanic circulation during HS1 impact deep ocean chemistry by decreasing DIC and thus increasing [CO₃²⁻]. This could lead to enhanced carbonate burial, thus leading to a global alkalinity decrease. To provide an estimate of this mechanism we now include an experiment where global alkalinity decreases linearly by 6 umol/L per 1000 years with a 2000 years lag from the initiation of the deglaciation (Figure 1d, orange line). Over the 16,000 years of the deglaciation this would amount to a global alkalinity decrease of ~96 umol/L, which corresponds to a first estimate of alkalinity changes scaled on global sea-level variations.

We have now added L. 218-226:

“While the simulated changes in atmospheric CO₂, $\delta^{13}\text{CO}_2$ and D14C are in very good agreement with paleo-records, the pCO₂ increase is slightly underestimated over the last 1000 years. This could be due to an overestimation of the surface alkalinity changes in these

simulations (Figure 5). In addition, it is important to note that in these transient simulations the global ocean alkalinity content was kept constant. However, a rising sea-level and an increase in deep ocean carbonate ion saturation during HS1 could have enhanced carbonate sedimentation, thus leading to a reduced global alkalinity content. Assuming a total glacial-interglacial alkalinity change of 96 $\mu\text{mol/L}$ (Methods), an ocean alkalinity decrease of 6 $\mu\text{mol/L}$ per 1000 years would induce a steady atmospheric CO_2 rise of ~ 5 ppm per 1000 years (orange line in Fig. 1d)."

We now also show simulated changes in hemispheric temperature, compared to the compilation of Shakun et al., (2012) in Figure S2. The simulated northern hemisphere temperature is about 0.5°C lower than the estimate of Shakun et al. (2012), while the southern hemisphere temperature is within the 2σ uncertainty.

We have also added some discussion about changes in iron fertilization (L.294-302):

"Changes in SH westerlies, Southern Ocean circulation and the resulting impact on the hydrological cycle and terrestrial biosphere could have reduced the iron input into the Southern Ocean, thus potentially contributing to the background atmospheric CO_2 increase during HS1 (Martinez Garcia et al, 2014). However, changes in iron fertilization alone cannot explain the observed variations in atmospheric $\Delta^{14}\text{C}$, ocean ventilation and high southern latitude warming. We therefore suggest that changes in iron fertilization mostly respond to changes in Southern Ocean ventilation, possibly providing a positive feedback. In addition, as marine export production decreased north of the polar front (Martinez Garcia et al., 2014), marine sediment cores south of the polar front display an increase in opal flux (Anderson et al., 2009), thus indicating a limited geographic extent of iron fertilization changes."

I am surprised to not see a close comparison with the ^{14}C coral data from Burke et al., 201210 and Chen et al. 201511. If the resolution is there, I would think these data provide a stronger test to the AAIW ventilation hypothesis than the deep sites currently used. At the moment the data model comparison only revisits an idea about deep ventilation in the comparison which is well-established. Also, I see that they are included in the Extended Data Table 3 and there is one data point at $\sim 56\text{S}$ and 2000 meters depth Extended Figure 1, but the Drake Passage Coral data are mostly from 1000 meters and as far south as 60S . Is some data missing?

The challenge with the coral data is that they come from different locations and depths adding to the difficulty of providing a quantitative framework. We have given serious consideration to including the Drake Passage record of Burke et al., (2012) in Figure 1. But given the uncertainties associated with these records, we have decided to include it in the Supplementary material instead. We are thus now showing the coral data from the Drake Passage from Burke et al. (2012) and Chen et al., (2015) in Figure S2.

The corals from the Drake Passage come from depths ranging from 800m to 1500m. This depth difference significantly impacts the ventilation ages and leads to significant uncertainties with respect to the timing of the decrease in ventilation ages in that region. The Drake Passage LGM data (20-18k) originates from $\sim 800\text{m}$ depth, while the data from the core of H1 (i.e., 16.176 ka, 15.76 ka and 15.22 ka) originate from significantly deeper

sites: 1196 and 1516m water depth. Their ventilation ages should thus be higher than the ones at ~800m. The reference of the Drake Passage coral data in our plot comes from an average of the ~800m data at 20.3, 19.6 and 18 ka (no data from deeper sites is available). To illustrate the uncertainties associated with the depth effect and based on the different ventilation ages shown in our simulations we have added an appropriate age shift to these 3 data points: 300 years for 1196m depth and 450 years for the data point at 1516m (grey line).

Regarding the extended figure, apologies, there was an error in our data file. We corrected it and the point referring to Burke et al., (2012) and Chen et al., (2015) from the Drake Passage is now at its correct place. Thank you for pointing this out.

Trigger for the deglaciation?

“Through its impact on both atmospheric CO₂ and temperature, enhanced Southern Ocean ventilation could have thus initiated the last deglaciation.”

This is minor conclusion that the authors have not oversold, but I am wonder if it is fully justified. To show that these changes are capable of sustaining atmospheric CO₂ and global temperature (or at the very least Antarctic temperature), I would have liked to see the model simulate the CO₂ plateau during the Bolling-Allerod and Antarctic Climate Reversal. At the moment the model run ends without a resumption of AMOC (stopping about 15ka). If the run was continued and CO₂ started to fall back to glacial levels it would fail to support this conclusion. The same critique could be made of Schittner et al., and many other results and (in my opinion) is a major problem for the models. If, on the other hand, the model was successful in sustaining CO₂ during the ACR, then it appears to have made a major breakthrough.

We understand this point raised by the Reviewer but we wanted to focus this study on the cold phase of HS1. Our study is already very comprehensive and including the Bolling-Allerod and ACR would make the study too long and complex. Instead, we want to focus on our findings that changes in Southern Ocean ventilation seem to be a necessary condition to increase atmospheric CO₂ and warm the high southern latitudes during HS1.

Terrestrial carbon

I could only see from one table in the Extended Data (Table 1) that the model was run with a terrestrial biosphere model. As far I as can tell this is not mentioned in the main text nor is the table referenced anywhere. Please convey this information to the reader and, if only just briefly, discuss how the terrestrial component acts as sink. How confident are the authors in the accuracy of the terrestrial model? A timeseries of the terrestrial and oceanic inventories would also be useful as seems standard for these type of modelling papers. The paper also opens with questions about the possibility of a terrestrial dominated source for the early deglacial CO₂ rise (Crichton et al.12) but then doesn't address this question directly. This study appears to say completely the opposite of Crichton. Why is this the case? It also doesn't support the terrestrial hypothesis for the 16.3ka event from Bauska et al., which should also be mentioned. Can the model rule out these hypotheses or are the available experiments incapable of testing this (e.g. is there

permafrost carbon in LOVECLIM to test Crichton or are there physical realistic changes in the precipitation during Heinrich Events to test Bauska?)

A permafrost module is not included in LOVECLIM, so the Crichton et al. (2016) hypothesis cannot be tested directly. However, it should be noted that the model used by Crichton et al (2016; namely, CLIMBER-2) is very simple: the atmospheric model is a statistical model, the land and atmospheric models have an extremely coarse resolution of 51° in longitude and 10° in latitude, and the ocean is reduced to a highly-idealised 3 basins, 2-dimensional model. This model suggests a permafrost carbon release of 200 GtC between 17 and 15ka, with a slight additional release of less than 50 GtC between 50 and 13 ka. This is in contrast with Kohler et al. (2015), who suggests a rapid permafrost carbon release during the Bolling-Allerod instead. It should also be noted that only a rough comparison of simulated vs proxy $\delta^{13}\text{C}$ changes were made due to the low resolution of their model: in particular, the surface North Atlantic (0-1500m, 10N-60N) and the deep South Atlantic (2500-5000m, 10S-40S) are compared to a stack of 3 cores in the North Atlantic and 1 core from the deep South Atlantic. Finally, it is also important to note in relation to the previous comment by the Reviewer, that a terrestrial carbon release originating from permafrost would also lead to an atmospheric $\Delta^{14}\text{C}$ decrease.

Nonetheless, we now evaluate and show in Figure S8, results of an additional simulation where the 16.2 ka event is simulated by a release of 50 GtC over 100 years (0.5 GtC/yr) with a $\delta^{13}\text{C}$ signature of -24 permil to mimic a terrestrial carbon release. This simulation leads to a relative agreement with atmospheric CO_2 and $\delta^{13}\text{CO}_2$ as also detailed L. 227-234:

“Recent studies (Rhodes et al., 2015, Bauska et al., 2016) raised the possibility that the abrupt CO_2 increase occurring at 16.2 ka could be due to a northern hemisphere terrestrial carbon release resulting from a southward shift of the ITCZ. Our model-data comparison suggests instead that Southern Ocean ventilation increased during HS1 and that changes in SH westerlies can cause an abrupt atmospheric CO_2 rise and $\delta^{13}\text{CO}_2$ decline at 16.2 ka in agreement with proxy data. In addition, our simulations suggest that the terrestrial carbon content increased during this period due to a higher CO_2 content and warmer conditions (Fig. 3). However, given the short duration of the 16.2 ka event, and the relative low resolution of the proxy data, a terrestrial carbon release cannot be ruled out (Fig. S8).”

Line-by-line comments

Title: I feel that the title may be too strong but I am willing to be convinced otherwise. It seems like the westerly wind hypothesis will remain a hypothesis (albeit a much stronger one after this study). Maybe something like “SH westerlies as a driver of early deglacial...”. Or something about the rapid ventilation of AAIW which is the more novel component of the study.

The title was changed to:

“Influence of Southern Hemisphere westerlies and Southern Ocean convection on the early deglacial atmospheric CO_2 rise”

Abstract: I would suggest using a term other than “jumps” to describe the rapid increases in atmospheric CO₂ (despite the fact that I see the term used once in Marcott et al. 2014). Also, lumping the CO₂ rise from ~17.5 to 16.5 with the truly rapid rise at 16.3 ka with the same term “jump” goes against the previous characterization.

“Jumps” has been changed to “phases”

Line 48. “So far, no 3-dimensional transient simulation...” The Schmittner et al., 2015 study requires this statement to be a bit more nuanced. If you provide a description and reference to that paper then you can easily make your case.

The Schmittner and Lund (2015) study was performed under constant pre-industrial boundary conditions. It was thus not a transient simulation of HS1. “Transient simulation” implies appropriate time-varying boundary conditions.

Lines 59-66. This reiterates some of the information in the abstract whilst also jumping to some conclusions. I am not sure it is totally necessary and might be a better place for some motivation for the study.

This paragraph has been removed.

Line 133. Revelle factors should be explained in a more generic form in the main body. Also, the section on Revelle factors in the methods doesn't actually use the term “Revelle factors”.

We changed this sentence to make it more explicit:

“DIC and ALK thus tend to compensate each other as their oceanic distributions are similar (Fig. S6) and their fractional impact on CO₂ have a similar magnitude, but an opposite sign (Methods).”

We have also added the reference to “Revelle factors” in the Methods:

“ γ_{DIC} , γ_{ALK} and γ_{SSS} are the Revelle factors and are equal to 10, -9.4 and 1 respectively (Sarmiento, 2006).”

Line 143 and Extended Figure 6. The presence of a polynya is not very clear from the figures (an arrow in the time-series plot of sea ice concentration would suffice). The impact of the polynya in the spatial plots was also not clear to me. Note that some readers won't be familiar with the location of the Weddell Sea.

We have now added an arrow to show the polynya and point to the Weddell Sea.

Lines 159-166. These statements need to be clearly denoted as either model results and conjecture. I worry they are too easily mixed at the moment.

This part of the text was significantly modified and expanded into two paragraphs.

Lines 167-172. This is a very “arm-wavy” paragraph. First the reference for the “Iberian Margin” is incorrect. The Martrat et al., 2014 reference is from the Alboran Sea. If the reference should be Martrat et al. 2007 for core MD01-2444 then I don't see evidence for the “two-phase cooling”. As argued above, I view the ice core and speleothem data as a more direct link the ITCZ movement.

This paragraph was significantly modified and the reference to the Alboran Sea record removed. Changes in the hydrological cycle as recorded in speleothems are now shown in Figure S2 and mentioned in that paragraph (Zhang et al., 2014).

Line 201: A small question: how and why is global alkalinity held constant? Does this mean the LOVECLIM is incapable of CaCO₃ feedbacks? It is worth mentioning this even though they would have a relatively small effect over the timescale of interest. This would prevent spinning-up to an accurate LGM state.

Most models are designed to conserve tracers to be able to perform long simulations. Therefore, in most biogeochemical models the total amount of carbon and alkalinity is kept constant through time. At present, LOVECLIM does not include a comprehensive sediment model, meaning that all quantities that are buried are “lost” from the model and are then artificially brought back through rivers to compensate. LOVECLIM is thus incapable of any CaCO₃ feedback in the sense that sedimented CaCO₃ cannot dissolve back into sea-water. However, as mentioned by the reviewer this CaCO₃ feedback has a relatively small direct impact on HS1 atmospheric CO₂ rise.

We agree with the reviewer that spinning-up to an appropriate LGM state is an important issue and that is why significant amount of time and effort was spent obtaining the LGM state. To obtain that state, the tracer concentration requirement was relaxed so as to obtain a higher DIC and ALK content in the LGM ocean than in the pre-industrial ocean. Careful validation and selection of the LGM state is discussed in Menviel et al. (2017). We should however point out that sensitivity experiments were performed from different LGM states and this did not affect the conclusions on the processes driving the HS1 atmospheric CO₂ increase.

Due to sea-level variations, the total ocean alkalinity most likely changed on glacial-interglacial timescales. Therefore, we also include an experiment in which the global ocean alkalinity decreases linearly to assess its impact on atmospheric CO₂.

Overall I thought the paper was very well written and it was a pleasure to read.

Figure 2:

It may just be me but I found the figure hard to follow. There is critical information presented in the d13C anomalies but also lots of generalized “arm waving” in the schematics. If the model is truly doing all this (which is the novelty of the study vs. data/idea paper) why couldn't this be conveyed more quantitatively? For example, could the wording and symbols about the d13C be replaced by subplots of the anomalies in the heat transport, SST, and sea ice over the Southern Ocean (with shading to indicate a zoom to the d13C plot)? Also, I wasn't sure what some of the arrows (like the red arrows) and symbols (like the ++) are referring to exactly.

Figure 2 was significantly modified based on the Reviewer's comments. A central map now shows the enhanced CO₂ outgassing, changes in sea-ice cover and precipitation anomalies. The zonally averaged plots now show DIC anomalies. δ¹³C anomalies are now shown in another figure.

Figure 3:

The meltwater and windstress forcing curves are bit hard to read. What does “calculated windstress” refer to? The DIC anomalies need units on the labels (umol/L?). “Taylor Dome” should be changed to “Taylor Glacier”.

Figure 3 has been split into three figures. In this version the forcing data should be easier to read. For the windstress, we now simply show the pre-industrial and LGM levels. The reference now correctly refers to “Taylor Glacier”.

Figure 4: Unit labels appear missing on some graphs.

Units have been added on all graphs.

Extended Data:

Figure 1: Needs unit labels.

Figure 3: I think all the section plots in the paper are missing units. The tick marks could also be improved by making the bigger and higher fidelity.

Figure 5: The “tongue” of AAIW in the Atlantic looks pretty weak for DIC and ALK. Has this been discussed in the previous model work? Could it be problematic for your interpretation of the 16.3 ka CO₂ which relies on the interplay between DIC and ALK in these waters? If it hasn't been done before it would be useful to a cross-plot of DIC and ALK comparison of the model and data from the major water masses.

Units have been added to all plots.

The reviewer is correct in saying that the tongues of both DIC and ALK in AAIW look weak in Figure S9. Model tuning often leads to compromises and when performing our simulations, this state of the model was the most appropriate one compared to observations.

As seen in Figure 6, the DIC anomalies at intermediate depth simulated by the global eddy permitting model do not extend much equatorward. So, the weak representation of AAIW in the global eddy-permitting model most likely leads to an underestimation of the potential oceanic carbon loss. However, while additional tuning of the model is underway, we cannot derive firm conclusions on the potential impacts of the weak AAIW tongue.

Thank you for considering my comments.

Thank you for your very constructive review.

References:

- (1) Marcott, S. A.; Bauska, T. K.; Buizert, C.; Steig, E. J.; Rosen, J. L.; Cuffey, K. M.; Fudge, T. J.; Severinghaus, J. P.; Ahn, J.; Kalk, M. L.; McConnell, J. R.; Sowers, T.; Taylor, K. C.; White, J. W. C.; Brook, E. J. *Nature* 2014, 514 (7524), 616–619.
- (2) Rhodes, R. H.; Brook, E. J.; Chiang, J. C. H.; Blunier, T.; Maselli, O. J.; McConnell, J. R.; Romanini, D.; Severinghaus, J. P. *Science* 2015, 348 (6238), 1016–1019.
- (3) Bauska, T. K.; Baggenstos, D.; Brook, E. J.; Mix, A. C.; Marcott, S. A.; Petrenko, V. V.; Schaefer, H.; Severinghaus, J. P.; Lee, J. E. *Proceedings of the National Academy of Sciences* 2016, 113 (13), 3465–3470.

- (4) Tschumi, T.; Joos, F.; Gehlen, M.; Heinze, C. *Climate of the Past* 2011, 7 (3), 771–800.
- (5) Lee, S.-Y.; Chiang, J. C. H.; Matsumoto, K.; Tokos, K. S. *Paleoceanography* 2011, 26.
- (6) Schmittner, A.; Lund, D. C. *Clim. Past* 2015, 11 (2), 135–152.
- (7) Lauderdale, J. M.; Williams, R. G.; Munday, D. R.; Marshall, D. P. *Climate Dynamics* 2017, 48 (5), 1611–1631.
- (8) Hain, M. P.; Sigman, D. M.; Haug, G. H. *Earth and Planetary Science Letters* 2014, 394 (Supplement C), 198–208.
- (9) Shakun, J. D.; Clark, P. U.; He, F.; Marcott, S. A.; Mix, A. C.; Liu, Z.; Otto-Bliesner, B.; Schmittner, A.; Bard, E. *Nature* 2012, 484 (7392), 49–54.
- (10) Burke, A.; Robinson, L. F. *Science* 2012, 335 (6068), 557–561.
- (11) Chen, T.; Robinson, L. F.; Burke, A.; Southon, J.; Spooner, P.; Morris, P. J.; Ng, H. C. *Science* 2015, 349 (6255), 1537–1541.
- (12) Crichton, K. A.; Bouttes, N.; Roche, D. M.; Chappellaz, J.; Krinner, G. *Nature Geoscience* 2016, 9, 683.

Reviewer #3 (Remarks to the Author):

We thank the Reviewer for their helpful comments, which contributed to improving our manuscript. We would like to emphasize that our work presents significant technological advances with the inclusion of transient simulations with a 3-dimensional carbon isotopes enabled model and a sensitivity study performed with a global eddy-permitting model. In addition our work highlights the multi-decadal impact of SH westerlies on the global carbon cycle due to ventilation of intermediate water. Additional information and discussion regarding the processes leading to atmospheric CO₂ increase during HS1 has now been added to the manuscript and Figure 2 was completely redone according to the Reviewer's comments. In the following we provide a point-by-point response to the Reviewers' comments in blue as well as excerpts from the manuscript showing the changes made in green.

The paper provides some interesting insights into the characterisation of the last deglaciation, especially the role of southern hemisphere westerlies in enhancing CO₂ outgassing. The observed strengthening of SH westerlies in recent decades is predicted to increase under future climate scenarios and therefore the results are timely and likely to appeal to a wide audience.

Enhanced air-sea gas exchange and upwelling of old carbon-enriched deep waters in the southern ocean has been suggested as a driver of the rapid jumps in $\delta^{13}\text{C}$ observed in ice cores (eg Buaska et al., 2016; Schmitt et al., 2012). Modelling studies have demonstrated a step-wise strengthening of Southern ocean wind stress and produced approximately similar changes in ocean water $\delta^{13}\text{C}$ (Tschumi et al., 2011).

The layout of this study is convincing and provides testable theories for the changes observed in the proxy records. It highlights the importance of shifting westerly winds in driving changes in CO₂ and the findings support and enhance previous studies. However, the mechanisms for the CO₂ jumps proposed in this study are not necessarily new (eg proposed and modelled by Tschumi et al., 2011) and build on previous modelling studies about the role of westerly winds (eg Lauderdale et al., 2013 and others).

General comments:

Previous studies have also attributed the CO₂ increases to other sources. Such as precipitation changes in the SH and enhanced dust or changes to the biological pump. Can the authors comment on these other hypothesis. Are they important in your scenarios?

We have now added more information to the Introduction about previous hypotheses on the early deglacial atmospheric CO₂ increase (L.65-88):

"A number of processes have been put forward to explain the early deglacial atmospheric CO₂ increase. Co-variation of iron flux and nutrient utilisation in the sub-Antarctic (Martinez Garcia et al., 2014) suggests that iron fertilization could exert a significant impact on atmospheric CO₂ during HS1 through its impact on the Southern Ocean biological pump. Modelling studies show that reduced iron fertilization would lead to a millennial atmospheric CO₂ increase of ~10 ppm coupled to a 0.1 permil $\delta^{13}\text{C}$ decrease (Bopp et al.,

2003, Tagliabue et al., 2009, Menviel et al., 2012, Bauska et al., 2016), during the early part of the deglaciation. However, changes in iron fertilization would not lead to any changes in atmospheric $\Delta^{14}\text{C}$ or ocean ventilation ages, and changes in Southern Ocean dust flux are ultimately controlled by SH hydrology and winds.

Indeed, a deglacial decline in South Atlantic ventilation ages has been observed (Skinner et al., 2010), indicating a possible role of Southern Ocean ventilation in driving the deglacial CO_2 increase. This change in Southern Ocean ventilation could be modulated by SH westerlies (Toggweiler et al., 2006, Anderson et al., 2009, Marcott et al., 2014). Idealized modelling studies, performed under constant pre-industrial boundary conditions have shown that stronger or poleward shifted SH westerlies could enhance deep ocean ventilation thus leading to an atmospheric CO_2 increase and $\delta^{13}\text{CO}_2$ decrease (Tschumi et al., 2011, Lauderdale et al., 2013, Menviel et al., 2015). However, this hypothesis has been challenged by a recent modelling study, also performed under constant pre-industrial boundary conditions, which concluded that changes in SH westerlies did not lead to significant changes in atmospheric CO_2 during HS1 (Schmittner and Lund, 2015). Instead, Schmittner and Lund (2015) suggested that the HS1 atmospheric CO_2 rise was solely due to a reduced efficiency of the biological pump resulting from a weaker Atlantic Meridional Overturning Circulation, even though no abrupt atmospheric CO_2 rise was simulated. In addition, while a strengthening and poleward shift of the SH westerlies seem to be a robust feature of future projections (Zheng et al., 2013), the behaviour of SH westerlies on glacial-interglacial and millennial timescales remain unclear (Chavillaz et al., 2013, Kohfeld et al., 2013).

As the magnitude and rate associated with an oceanic carbon release to the atmosphere during the deglaciation have been questioned, a deglacial transfer of carbon from the terrestrial to atmospheric reservoir has been put forward, either as thawing of Northern Hemisphere permafrost (Crichton et al., 2016) or as a Northern Hemisphere terrestrial carbon release (Rhodes et al., 2015, Bauska et al., 2016) due to a southward shift of the Intertropical Convergence Zone (ITCZ). However, the terrestrial carbon reservoir globally increased over the deglaciation (Ciais et al., 2011) and the timing and magnitude of the permafrost carbon contribution remain poorly constrained. “

We have also now added the simulated changes in precipitation and southward shift of the ITCZ (Figure 2, S1) as well as changes in terrestrial carbon content (Figures 3 and S1). Our results suggest an increase in terrestrial carbon across HS1 due to globally warmer conditions and higher pCO_2 (fertilization effect).

Changes in iron fertilization are not taken into account in our simulations, and on the contrary, globally-averaged marine export production increases across HS1.

We have however now added a paragraph on the other processes suggested to have played a role in the HS1 CO_2 increase:

L. 227-234:

“Recent studies (Rhodes et al., 2015, Bauska et al., 2016) raised the possibility that the abrupt CO_2 increase occurring at 16.2 ka could be due to a Northern Hemisphere terrestrial carbon release resulting from a southward shift of the ITCZ. Our model-data comparison

suggests instead that Southern Ocean ventilation increased during HS1 and that SH westerly winds change can cause an abrupt atmospheric CO₂ rise and $\delta^{13}\text{CO}_2$ decline at 16.2 ka in agreement with proxy data. In addition, our simulations suggest that the terrestrial carbon content increased during this period due to a higher CO₂ content and warmer conditions (Fig. 3). However, given the short duration of the 16.2 ka event and the relative low resolution of the proxy data a terrestrial carbon release cannot be ruled out (Fig. S8)."

And L.294-302:

"Changes in SH westerlies, Southern Ocean circulation and the resulting impact on the hydrological cycle and terrestrial biosphere could have reduced the iron input to the Southern Ocean, thus potentially contributing to the background atmospheric CO₂ rise during HS1 (Martinez Garcia et al, 2014). However, changes in iron fertilization alone cannot explain the observed variations in atmospheric $\Delta^{14}\text{C}$, ocean ventilation and high southern latitude warming. We therefore suggest that changes in iron fertilization mostly respond to changes in Southern Ocean ventilation, possibly providing a positive feedback. In addition, as marine export production decreased north of the polar front (Martinez Garcia et al., 2014), marine sediment cores south of the polar front display an increase in opal flux (Anderson et al., 2009), thus indicating a limited geographic extent of iron fertilization changes."

We have also added a sentence on changes in the efficiency of the biological pump in our simulations (L.180-186):

"A further weakening of the oceanic circulation from the LGM into HS1 as simulated in experiment LH1 only leads to small changes in the carbon reservoirs, with a slight increase in the global remineralized PO₄ content, and 2% decrease in the preformed PO₄ content (Fig. S3). In contrast, enhanced Southern Ocean ventilation leads to an oceanic carbon loss associated with a decrease in remineralized PO₄ and a 6% increase in preformed PO₄, thus implying a reduced efficiency of the biological pumps in experiments with enhanced AABW formation (LH1-SO, LH1-SHW and LH1-SO-SHW)."

Looking at the changes in ventilation age there is considerable mismatch between the simulated and observed (sediment cores) anomalies (fig 1 extended data). I know a perfect match cannot be expected but can this be explained? Are the values within the dating uncertainties of the sediment records?

The correlation value R between model and data is 0.52 for ventilation ages. This is not perfect but there are significant uncertainties associated with oceanic ventilation age estimates. In addition, it is worth keeping in mind that this figure shows simulated ventilation ages zonally averaged over the Atlantic and Pacific basins whereas the data represents a discrete point in space: i.e. longitudinal gradients could also play a role in the model-proxy mismatch. Nevertheless, both model and data display higher ventilation ages in the North Atlantic, reversing to younger ages in the Southern Ocean. In the Pacific, it is worth noting the negative age anomalies below 2000 m depth, in good agreement with the model.

Figures:

Fig 2. This figure is fundamental to explaining the sequence of events and needs to be

expanded to make the processes clearer. What do the dots in circles signify? Could the sea-air pCO₂ anomalies plot (extended data 4d) be included here? Replacing the central map to show the enhanced out gassing around the southern ocean under this scenario and how this relates to the ocean $\Delta\delta^{13}\text{C}$ depth profiles? It may make the plot too complicated but it could be helpful to tie the results spatially with the main conclusions of the paper.

Figure 2 was significantly modified based on the Reviewers' comments. A central map now shows the sea-air CO₂ anomalies indicating possible regions of CO₂ outgassing as well as changes in sea-ice cover and precipitation anomalies. The zonally averaged plots now show DIC anomalies. $\delta^{13}\text{C}$ anomalies are now shown in another figure.

Caption correct (r) and "shifts"
corrected

Fig 3. This is a really busy plot and needs to be separated or reformatted. The simulated DIC anomalies would work better side by side, eg LH1 Atlantic (left) and Pacific (right). To allow for easier comparison of changes in both oceans during the same scenario. As done in extended fig 1.

Plots a-g need to have more space to allow clearer separation of colours. I am not sure what plot h adds to the figure. Perhaps removing this could provide more space for the other timeseries?

Figure 3 has been split into three figures. The simulated DIC anomalies were moved to figure 2 and panel (h) now stands as a separate figure 5.

Figure 4 – is it possible to plot the DIC anomalies on the same scale?

What is the relevance of 1320 m and 4346 m depths? It seems very specific and is quoted as 4300 m depth in the text.

We kept DIC anomalies on different scales so that the anomalies are easier to see. We want to show the DIC changes at intermediate depth and in the deep ocean. We were showing the exact mid-depth of the vertical level we took in the model (i.e. 1320 m and 4346 m respectively). Now for clarity we are showing the range of depth for each level.

Extended data 6 – just being picky but could you crop plot e to produce a circular plot? Panel (e) was modified to produce a circular plot.

Minor comments:

Line 42 (and others) – formatting error (h)? ‰

This formatting error does not appear in our version... hopefully it will be fine in the new version. If not, we apologize.

Line 77 – sentence starting via? Perhaps reword
This has been rephrased.

Line 103 – move "weak" in front of AAIW → done

Line 186 – “ran” → modified

Line 208 – “as follows” → modified

Missing refs: → the following references have been added

T. Tschumi, et al, Deep ocean ventilation, carbon isotopes, marine sedimentation and the deglacial CO₂ rise. *Clim. Past* 7, 771 (2011). doi:10.5194/cp-7-771-2011

Schmitt et al, Carbon Isotope Constraints on the Deglacial CO₂ Rise from Ice Cores *Science* (2012) DOI: 10.1126/science.1217161

Lauderdale, J.M., Garabato, A.C.N., Oliver, K.I.C. et al. *Clim Dyn* (2013) 41: 2145. <https://doi.org/10.1007/s00382-012-1650-3>

Reviewers' Comments:

Reviewer #2:

Remarks to the Author:

Review "Influence of Southern Hemisphere Westerlies and Southern Ocean Convection on the early deglacial atmospheric CO₂ rise"

The authors have done a thorough job addressing my earlier comments. I believe the expanded introduction and discussion make for a much more impactful paper.

However, there is one issue that has arisen in this second round of review. In the first round, both reviewer 1 and myself were unclear about to what degree the model had been forced. It is now clear in this revision that the model results do rely on a significant westerly wind forcing that in some ways (mostly the timing) leads to the data model agreement. For the most part, the manuscript makes this clear and provides appropriate justification. However, it does change the novelty of the study. As a data focused person I find the results a compelling step-forward to explaining deglacial CO₂ (and of significant interest to the journal's audience). But I must mention that the results now only seem to be an incremental step-forward rather than a breakthrough (which would be a fully coupled model with only external forcing). The eddy-permitting model results seem to stand on their own as a novel result but my expertise doesn't allow me to evaluate this in detail.

My only major question is why are 17.2 and 16.2 ka chosen as the times to ramp up SH westerlies? I follow the logic of events starting from a shutdown of NADW to the SH westerly (lines 113-123), but the timing doesn't quite make sense to me. The model seems to simulate a near complete shutdown of NADW by 18 ka (Figure 1), so why impose a two-phase step whilst also delaying the enhanced westerlies for another few thousand years? It seems if another line of arguments is needed. Is there a large amount of internal variability during HS1? Could the forcing at 16 ka be the impact of a Heinrich Event? Note some authors have suggested multiple phases to Heinrich Stadials.

There are also a few issues with imprecise language that I raise below.

Page 2, Line 24: "We show that the early deglacial CO₂ increase is due to enhanced Southern Ocean upwelling..." This statement is too strong given the uncertainties and assumptions of this study. The authors have not fully ruled out other hypotheses as they rightfully acknowledge in the main text. I would replace "is due" with "can be driven by" or "is consistent with" or "can be fully explained by".

Page 3, line 42-44: I found this sentence quite vague. Moreover, the phrase "reducing the degrees of freedom" is typically used in a very strict sense whereas here it seems very general.

Page 3, line 46: In a similar vein I am not sure the term "bifurcation" is properly defined here.

Page 3, Line 55: The meaning of "opposite North Atlantic-Antarctic temperature change" is not clear

Page 4, line 62: Because changes in $\delta^{13}\text{C}$ -CO₂ require a net transfer of organic carbon in or out of the surface ocean, a more precise term for "oceanic nutrient utilization" might be "export productivity" or the "efficiency of the biological pump".

Page 4, line 72: Two other effects on dust flux would be the exposure of continental shelves and the extent of the Patagonian ice sheet.

Page 5, line 83-85: Drawing parallels to future predictions is important but I don't quite see the

link here. A more detailed argument about why we want to study the past to improve future predictions probably fits somewhere else in the paper.

Page 5, Line 91: A more appropriate reference for the regrowth of the terrestrial biosphere would be the coupled [CO₃] and ¹³C (Yu et al., 2010), the benthic ¹³C stacks (Peterson et al., 2014) or some of the modeling work by Kaplan (Kaplan et al., 2002). Note the Ciais reference currently used calls for the opposite in reference to permafrost. From the abstract: "We suggest that the disappearance of this carbon pool at the end of the Last Glacial Maximum may have contributed to the deglacial rise in atmospheric carbon dioxide concentrations."

Page 8, line 149: What specifically are the "early deglacial changes at mid and high southern latitudes"? Temperature, precipitation, or is it the actual outgassing of CO₂?

Page 17, line 320: It would be better if the "profound implications" could be quantified?

Page 32, Figure 2: This is a fantastic figure. However, the labelling that is projected into the page is hard to read. CO₂ needs subscripts.

Figure 3: "δ¹³C" needs superscript and Greek symbol

Figure 4: CO₂ subscripts; μmol needs Greek symbol

Overall, I noticed the term "proxy" used frequently to describe the ice core CO₂ and δ¹³C-CO₂ data. I feel that ice core data are fundamentally different than the other "proxies" we have for atmospheric CO₂ (boron isotopes, stomata, etc.) because they are direct samples of the ancient atmosphere. I thus suggest this term should not be used.

Supplemental Information

Page 3: Paixao is missing a tilde over the second a.

Figure S3: subscripts and greek symbols missing.

Figure S6: replace "and its isotopes" with something like "concentration and isotopic composition of atmospheric CO₂"

Figure S7: Austral summer sea ice extent looks extremely large - more like the old CLIMAP reconstruction than the Gersonde reconstruction (Gersonde et al., 2005). It's particularly bad in the Australian Sector. Has this been discussed in a previous study?

References

Gersonde, R., Crosta, X., Abelmann, A., & Armand, L. (2005). Sea-surface temperature and sea ice distribution of the Southern Ocean at the EPILOG Last Glacial Maximum - A circum-Antarctic view based on siliceous microfossil records. *Quaternary Science Reviews*, 24(7–9), 869–896. <https://doi.org/10.1016/j.quascirev.2004.07.015>

Kaplan, J. O., Prentice, I. C., Knorr, W., & Valdes, P. J. (2002). Modeling the dynamics of terrestrial carbon storage since the Last Glacial Maximum. *Geophysical Research Letters*, 29(22), 4.

Peterson Carlye D., Lisiecki Lorraine E., & Stern Joseph V. (2014). Deglacial whole-ocean δ¹³C change estimated from 480 benthic foraminiferal records. *Paleoceanography*, 29(6), 549–563. <https://doi.org/10.1002/2013PA002552>

Yu, J., Broecker, W. S., Elderfield, H., Jin, Z., McManus, J., & Zhang, F. (2010). Loss of Carbon from the Deep Sea Since the Last Glacial Maximum. *Science*, 330(6007), 1084–1087. <https://doi.org/10.1126/science.1193221>

Reviewer #3:

Remarks to the Author:

I am satisfied with the changes made to the paper and thank the authors for their thorough response. I like the revised figure 2 and feel the whole paper is much improved. I would be happy to see the paper published in nature communications.

Reviewer #4:

Remarks to the Author:

I am a new reviewer to this paper, who has not been involved in the first round of reviews. I was asked to see, if the concerns of reviewer 1 have been addressed in the revision. This is a difficult task, since I do not know the initial version of the paper, however, from what is written in the rebuttal, I think the authors have demonstrated that they took care about the issues raised and changed the draft accordingly. However, I myself have some points I like to raise, which should be addressed in another iteration (sorry). In summary, I think it a paper worth to be published in this journal.

1. In the introduction the idealized modelling experiments on changes in the SH westerlies and the carbon cycle are discussed, following the initial suggestion of Toggweiler et al (2006), ref 8. What is missing here is the to my knowledge most recent effort to this topics by Völker and Köhler (2013), which clearly showed that the background climate state is important to really say something meaningful. This needs to be discussed, or can be used to set some of the other papers in that direction (starting from preindustrial or modern, but not LGM conditions) into perspective. This finding of Völker and Köhler (2013) on state-dependency has also consequences for the usage and interpretation of the eddy-permitting model provided here at the end. If I understood it correctly the eddy-permitting simulations have been performed from modern background conditions, right? If so, this needs to be stated upfront in the main manuscript and asks how useful they can be here. I would even go that far to say that the part of the eddy-permitting results — if they are performed only for modern background conditions — can be completely left out of the draft, since I am not sure we learn a lot about LGM, or H1 ocean physics. But I leave this for the authors to decide. If left in, some careful note of caution on the restricted possibility to transfer the knowledge gained from them to the LGM is needed.

2. Based on this importance of background climate state for the carbon cycle I am not sure the final statement (lines 318-320) which transfers the importance of the westerlies during the analysed time frame (LGM, H1) to the future can be made as such. This at least needs to be stated more careful or left out completely.

3. I understand that 14C production rates have been varied according to some reviewer comments, but it is not written in the methods (line 344) at which rate the 14C production rates are kept constant. At modern levels or at LGM level which I believe would be 20% or so higher? If they are kept constant at modern levels, then the question arises how results would change with higher 14C production rates rate. Connected with that: Typically it takes quite some time (some 10-kyrs) to run 14C in the marine carbon cycle into equilibrium. Some details on the length of the spinup — or the equilibrium of the 14C cycle — or the simulated atm 14C at LGM should be included.

4. If I got the overall interpretation for 14C right your paper suggests, that the large and rather abrupt decline in atm 14C between 17.5 and 14.5 ka (also called Mystery Interval) by 190 permil is solely be explained by ocean overturning processes, and changes in 14C production rate can be ignored. If this is not correct, then I have missed something, what others might also miss and some rephrasing might be needed. If this is correct, then you suggest a substantially smaller contribution of 14C production rate than earlier box model studies, e.g. (Köhler et al., 2006; Hain

et al., 2014) which needs to be discussed.

5. The references list needs some careful finetuning, since a lot of the references are incomplete (missing paper numbers by Nature Communications or most AGU papers without page numbers (e.g ref 3, 8, 20, 23, 2, 33, 34, 36, 38, 42, 46, 70), wrong page numbers in ref 55.

6. Fig 5: If I understood the caption correctly, each of the 4 columns refers to one scenario, but not the scenario name is given in the x-labels, but a suggested strength of the AABW. I suggest to at least add the scenario names to the x axis. Further, I think the x position within one column has no meaning, right? If so, the results of the 7 analysed processes should be placed equidistant along x. If there is a meaning in the x-position, it needs to be explained.

7. Ice core data: It needs to be mentioned on which age model the EDC, WDC, Taylor Glacier data are plotted. If as published originally, please say so, but you should be aware that WDC has gained a revised WD2014 chronology, which shifted data points (Buizert et al., 2015; Sigl et al., 2016), check out <https://www.ncdc.noaa.gov/paleo-search/study/20246>. Please include dots in the ice core CO₂ time series (as done for d13CO₂) in Fig 1, and also include dots in CO₂ and d13CO₂ ice core data shown in Fig 3.

8. When referring to figures with multiple panels, please always mention to which subpanel you refer.

References

Buizert, C., Cuffey, K. M., Severinghaus, J. P., Baggenstos, D., Fudge, T. J., Steig, E. J., Markle, B. R., Winstrup, M., Rhodes, R. H., Brook, E. J., Sowers, T. A., Clow, G. D., Cheng, H., Edwards, R. L., Sigl, M., McConnell, J. R., and Taylor, K. C.: The WAIS Divide deep ice core WD2014 chronology — Part 1: Methane synchronization (68-31 ka BP) and the gas age-ice age difference, *Climate of the Past*, 11, 153–173, doi:10.5194/cp-11-153-2015, 2015.

Hain, M. P., Sigman, D. M., and Haug, G. H.: Distinct roles of the Southern Ocean and North Atlantic in the deglacial atmospheric radiocarbon decline, *Earth and Planetary Science Letters*, 394, 198 – 208, doi:<http://dx.doi.org/10.1016/j.epsl.2014.03.020>, 2014.

Köhler, P., Muscheler, R., and Fischer, H.: A model-based interpretation of low frequency changes in the carbon cycle during the last 120 000 years and its implications for the reconstruction of atmospheric $\Delta^{14}\text{C}$, *Geochemistry, Geophysics, Geosystems*, 7, Q11N06, doi:10.1029/2005GC001228, 2006.

Sigl, M., Fudge, T. J., Winstrup, M., Cole-Dai, J., Ferris, D., McConnell, J. R., Taylor, K. C., Welten, K. C., Woodruff, T. E., Adolphi, F., Bisiaux, M., Brook, E. J., Buizert, C., Caffee, M. W., Dunbar, N. W., Edwards, R., Geng, L., Iverson, N., Koffman, B., Layman, L., Maselli, O. J., McGwire, K., Muscheler, R., Nishiizumi, K., Pasteris, D. R., Rhodes, R. H., and Sowers, T. A.: The WAIS Divide deep ice core WD2014 chronology - Part 2: Annual-layer counting (0-31 ka BP), *Climate of the Past*, 12, 769–786, doi:10.5194/cp-12-769-2016, 2016.

Völker, C. and Köhler, P.: Responses of ocean circulation and carbon cycle to changes in the position of the Southern hemisphere westerlies at Last Glacial Maximum, *Paleoceanography*, 28, 726–739, doi: 10.1002/2013PA002556, 2013.

Key=

Black= Reviewers' comments

Blue= Authors' responses

Green = Modified text in the manuscript

Reviewer #2 (Remarks to the Author):

Review "Influence of Southern Hemisphere Westerlies and Southern Ocean Convection on the early deglacial atmospheric CO₂ rise"

The authors have done a thorough job addressing my earlier comments. I believe the expanded introduction and discussion make for a much more impactful paper.

However, there is one issue that has arisen in this second round of review. In the first round, both reviewer 1 and myself were unclear about to what degree the model had been forced. It is now clear in this revision that the model results do rely on a significant westerly wind forcing that in some ways (mostly the timing) leads to the data model agreement. For the most part, the manuscript makes this clear and provides appropriate justification. However, it does change the novelty of the study. As a data focused person I find the results a compelling step-forward to explaining deglacial CO₂ (and of significant interest to the journal's audience). But I must mention that the results now only seem to be an incremental step-forward rather than a breakthrough (which would be a fully coupled model with only external forcing). The eddy-permitting model results seem to stand on their own as a novel result but my expertise doesn't allow me to evaluate this in detail.

We thank the Reviewer for this second helpful review and the positive comments about the study. In the following we provide a point-by-point response to the Reviewers' comments in blue as well as excerpts from the manuscript showing the changes made in green.

My only major question is why are 17.2 and 16.2 ka chosen as the times to ramp up SH westerlies? I follow the logic of events starting from a shutdown of NADW to the SH westerly (lines 113-123), but the timing doesn't quite make sense to me. The model seems to simulate a near complete shutdown of NADW by 18 ka (Figure 1), so why impose a two-phase step whilst also delaying the enhanced westerlies for another few thousand years? It seems if another line of arguments is needed. Is there a large amount of internal variability during HS1? Could the forcing at 16 ka be the impact of a Heinrich Event? Note some authors have suggested multiple phases to Heinrich Stadials.

The timing of the SH westerly changes is based on changes in NADW as inferred from proxy records. The Pa/Th record from the Bermuda rise (McManus et al., 2004) suggests a significant weakening of NADW at ~17.2 ka (Figure 1a). However, Hodell et al., 2016 suggest that the main phase of the Laurentide ice-sheet discharge occurred at 16.2 ka. Interestingly, this timing is in agreement with a Pa/Th record from the Iberian margin (Gherardi et al., 2009), which displays a significant increase in Pa/Th, indicating a weakening of deep ocean circulation at ~16.5 ka.

What comes out of the proxy records is thus a first phase of NADW weakening at ~19-17.5 ka, probably linked to the disintegration of the Eurasian ice-sheet, very weak NADW between ~17.5 and 16.2 ka and disintegration of the Laurentide ice-sheet further weakening NADW transport at ~16.2 ka. There are however still uncertainties with the exact state of NADW formation during these phases, and only broad changes (such as the ones presented here) have been simulated.

It is thus hypothesised that the ITCZ changes at ~17.5 ka led to a first phase of SH westerly strengthening, while the proper Heinrich event at ~16.2 ka led to further changes in the ITCZ and SH westerly winds. It is important to note that across HS1 the concomitant atmospheric CO₂ increase and sea-ice decrease in the South would have also influenced the temperature gradient between the hemispheres and the ITCZ changes.

The text was modified as follows L. 121-124:

“This timing of 17.2 ka corresponds, within dating uncertainties, to the Bermuda Rise 231Pa/230Th record reaching maximum values (McManus et al., 2004), thus indicating very weak NADW transport. However, another phase of NADW weakening (Gherardi et al., 2009), probably associated with the disintegration of the Laurentide ice-sheet and Heinrich event 1, occurred at ~16.2 ka (Hodell et al., 2017).”

L. 299-302:

“During the first phase of weak NADW transport at ~17.2 ka (McManus et al., 2004), an intensification of the SH westerlies and associated enhanced deep ocean convection could have led to a rise in atmospheric CO₂ concentration and initiated the deglaciation at high southern latitudes.”

L. 321-324:

“Sediment records from the Iberian margin have shown that ~16.2 ka probably corresponds to the beginning of Heinrich event 1 *stricto sensu* (Hodell et al., 2017), and another phase of NADW weakening (Gherardi et al., 2009).”

There are also a few issues with imprecise language that I raise below.

Page 2, Line 24: “We show that the early deglacial CO₂ increase is due to enhanced Southern Ocean upwelling....” This statement is too strong given the uncertainties and assumptions of this study. The authors have not fully ruled out other hypotheses as they rightfully acknowledge in the main text. I would replace “is due” with “can be driven by” or “is consistent with” or “can be fully explained by”.

Following the Reviewer’s suggestion we changed the sentence to

“We show that the early deglacial CO₂ increase can be explained by...”

Page 3, line 42-44: I found this sentence quite vague. Moreover, the phrase “reducing the degrees of freedom” is typically used in a very strict sense whereas here it seems very general.

The sentence was amended as follows:

“A detailed study of the deglaciation would provide a more direct link between changes in the climate and carbon cycle and would allow a direct model-data comparison, thus further constraining the processes responsible for atmospheric CO₂ changes.”

Page 3, line 46: In a similar vein I am not sure the term “bifurcation” is properly defined here.

“bifurcation” was replaced by “transition”

Page 3, Line 55: The meaning of “opposite North Atlantic-Antarctic temperature change” is not clear

The sentence was changed to:

“However, the mechanisms leading to HS1 CO₂ rise are still poorly constrained and an overarching mechanism linking this CO₂ rise to the North Atlantic cooling and high southern latitudes warming is still missing.”

Page 4, line 62: Because changes in $\delta^{13}\text{C}$ -CO₂ require a net transfer of organic carbon in or out of the surface ocean, a more precise term for “oceanic nutrient utilization” might be “export productivity” or the “efficiency of the biological pump”.

“oceanic nutrient utilization” was replaced by “marine export production”.

Page 4, line 72: Two other effects on dust flux would be the exposure of continental shelves and the extent of the Patagonian ice sheet.

The sentence was amended as follows:

“and variations in atmospheric iron deposition to the Southern Ocean are ultimately controlled by the exposure of the continental shelves, SH hydrology and winds.”

Page 5, line 83-85: Drawing parallels to future predictions is important but I don’t quite see the link here. A more detailed argument about why we want to study the past to improve future predictions probably fits somewhere else in the paper.

The parallel to future projections was removed and the sentence was moved to another location. The sentence now reads:

“However, the latitudinal position of SH westerlies at the LGM remains unclear (Chavaillaz et al., 2013, Kohfeld et al., 2013).”

Page 5, Line 91: A more appropriate reference for the regrowth of the terrestrial biosphere would be the coupled [CO₂] and $\delta^{13}\text{C}$ (Yu et al., 2010), the benthic $\delta^{13}\text{C}$ stacks (Peterson et al., 2014) or some of the modeling work by Kaplan (Kaplan et al., 2002). Note the Ciais reference currently used calls for the opposite in reference to permafrost. From the abstract: “We suggest that the disappearance of this carbon pool at the end of the Last Glacial Maximum may have contributed to the deglacial rise in atmospheric carbon dioxide concentrations.”

As suggested by the reviewer, we have replaced the reference to Ciais et al. (2012), by the ones of Yu et al., (2010) and Peterson et al., (2014).

Page 8, line 149: What specifically are the “early deglacial changes at mid and high southern latitudes”? Temperature, precipitation, or is it the actual outgassing of CO₂?

Here we are mostly referring to temperature, even though this would also impact the hydrological cycle.

“early deglacial changes” was changed to “early deglacial warming”.

Page 17, line 320: It would be better if the “profound implications” could be quantified?

A quantification of the impact of SH westerly changes on future projections is out of the scope of the study. Nevertheless, we are now more specific:

“Given the projected poleward intensification of SH westerlies over the 21st Century, and the fact that the Southern Ocean has absorbed ~10% of anthropogenic CO₂ emissions (Sabine et al., 2004, Mikaloff-Fletcher et al., 2006), our results would suggest a reduced CO₂ sequestration in the Southern Ocean, with a significant impact on future atmospheric CO₂ and climate change.”

Page 32, Figure 2: This is a fantastic figure. However, the labelling that is projected into the page is hard to read. CO₂ needs subscripts.

Thank you for the positive comment. The figure labelling was slightly modified so as to be easier to read.

Figure 3: “d¹³C” needs superscript and Greek symbol

The text on figure 3 has been modified accordingly.

Figure 4: CO₂ subscripts; μmol needs Greek symbol

The text on figure 4 has been modified accordingly.

Overall, I noticed the term “proxy” used frequently to describe the ice core CO₂ and d¹³C-CO₂ data. I feel that ice core data are fundamentally different than the other “proxies” we have for atmospheric CO₂ (boron isotopes, stomata, etc.) because they are direct samples of the ancient atmosphere. I thus suggest this term should not be used.

We’ve gone through the document and changed the reference to the ice core records.

L. 224: “the atmospheric CO₂ increase in close agreement with ice core records.”

L. 243: “Our model-data comparison shows instead that Southern Ocean ventilation increased during HS1 and that changes in SH westerly winds can cause an abrupt atmospheric CO₂ rise and δ¹³CO₂ decline at 16.2 ka in agreement with ice core and marine sediment records.”

Legend of Figure 1: “Model-paleodata comparison across HS1. Time evolution of selected paleo records across HS1...”

Supplemental Information

Page 3: Paixao is missing a tilde over the second a.

A tilde has been added to Paixão.

Figure S3: subscripts and greek symbols missing.

Subscripts and Greek symbols have been added to Figure S3.

Figure S6: replace “and its isotopes” with something like “concentration and isotopic compositing of atmospheric CO₂”

The text has been changed to:

“Simulated atmospheric CO₂ concentration, $\delta^{13}\text{CO}_2$ and atmospheric $\Delta^{14}\text{C}$ ”

Figure S7: Austral summer sea ice extent looks extremely large - more like the old CLIMAP reconstruction than the Gersonde reconstruction (Gersonde et al., 2005). It’s particularly bad in the Australian Sector. Has this been discussed in a previous study?

LOVECLIM does simulate a fairly large sea-ice extent in the Southern Ocean in line with other models (e.g. Figure R1b, CCSM4, Sime et al., 2016), and globally consistent with the sea-ice data from Gersonde et al. (2005), except, as highlighted by the Reviewer, for 3 data points at ~120°W-100°W (Figure R1).

In the original Figure S7 we were showing the 0.1 m sea-ice contour. Sea-ice is quite a dynamic system, and the sea-ice concentration varies significantly in time and space. The 30% sea-ice content is usually used to define sea-ice extent, therefore we are now showing the 30% sea-ice contour. For reference we are also showing the 85% sea-ice contour.

Figure R1: a) Annual mean sea-ice concentration (%) as simulated by LOVECLIM in the initial LGM state. The 15% and 30% sea-ice contours are overlaid; b) Figure 4b from Sime et al., 2016 showing the annual mean LGM 15% sea-ice concentration contour in different PMIP3 models (lines) and compared to the Gersonde et al. (2005) dataset (blue and red dots).

Additional references:

J.M. Gherardi, L. Labeyrie, S. Nave, R. Francois, J.F. McManus, E. Cortijo, 2009, Glacial-interglacial circulation changes inferred from 231Pa/230Th sedimentary record in the North Atlantic region, *Paleoceanography*, 24, PA2204, doi:10.1029/2008PA001696.

D.A. Hodell, J.A. Nicholl, T.R.R. Bontognali, S. Danino, J. Dorador, J.A. Dowdeswell, J. Einsle, H. Kuhlmann, B. Martrat, M.J. Mlenek-Vautravers, F.J. Rodriguez-Tovar, U. Rohl, 2017, Anatomy of Heinrich Layer 1 and its role in the last deglaciation, *Paleoceanography*, 32, p 284-303, doi:10.1002/2016PA003028.

Mikaloff-Fletcher, S. et al. 2006, Inverse estimates of anthropogenic CO₂ uptake, transport, and storage by the ocean. *Global Biogeochemical Cycles* 20, doi:10.1029/2005GB002530.

Sabine, C. et al. 2004, The oceanic sink of anthropogenic CO₂. *Science* 305, 367-371.

Sime et al., (2016), Sea ice led to poleward-shifted winds at the Last Glacial Maximum: the influence of state dependency on CMIP5 and PMIP3 models. *Climate of the Past*, 12, 2241-2253, <https://doi.org/10.5194/cp-12-2241-2016>.

References

Gersonde, R., Crosta, X., Abelmann, A., & Armand, L. (2005). Sea-surface temperature and sea ice distribution of the Southern Ocean at the EPILOG Last Glacial Maximum - A circum-Antarctic view based on siliceous microfossil records. *Quaternary Science Reviews*, 24(7-9), 869-896. <https://doi.org/10.1016/j.quascirev.2004.07.015>

Kaplan, J. O., Prentice, I. C., Knorr, W., & Valdes, P. J. (2002). Modeling the dynamics of terrestrial carbon storage since the Last Glacial Maximum. *Geophysical Research Letters*, 29(22), 4.

Peterson Carlye D., Lisiecki Lorraine E., & Stern Joseph V. (2014). Deglacial whole-ocean $\delta^{13}\text{C}$ change estimated from 480 benthic foraminiferal records. *Paleoceanography*, 29(6), 549-563. <https://doi.org/10.1002/2013PA002552>

Yu, J., Broecker, W. S., Elderfield, H., Jin, Z., McManus, J., & Zhang, F. (2010). Loss of Carbon from the Deep Sea Since the Last Glacial Maximum. *Science*, 330(6007), 1084-1087. <https://doi.org/10.1126/science.1193221>

Reviewer #3 (Remarks to the Author):

I am satisfied with the changes made to the paper and thank the authors for their thorough response. I like the revised figure 2 and feel the whole paper is much improved. I would be happy to see the paper published in nature communications.

We thank the Reviewer for their constructive and positive comments, which improved the manuscript.

Reviewer #4 (Remarks to the Author):

I am a new reviewer to this paper, who has not been involved in the first round of reviews. I was asked to see, if the concerns of reviewer 1 have been addressed in the revision. This is a difficult task, since I do not know the initial version of the paper, however, from what is written in the rebuttal, I think the authors have demonstrated that they took care about the issues raised and changed the draft accordingly. However, I myself have some points I like to raise, which should be addressed in another iteration (sorry). In summary, I think it a paper worth to be published in this journal.

We thank the Reviewer for taking the time to review our manuscript, and for their constructive comments. In the following we provide a point-by-point response to the Reviewers' comments in blue as well as excerpts from the manuscript showing the changes made in green.

1. In the introduction the idealized modelling experiments on changes in the SH westerlies and the carbon cycle are discussed, following the initial suggestion of Toggweiler et al (2006), ref 8. What is missing here is the to my knowledge most recent effort to this topics by Völker and Köhler (2013), which clearly showed that the background climate state is important to really say something meaningful. This needs to be discussed, or can be used to set some of the other papers in that direction (starting from preindustrial or modern, but not LGM conditions) into perspective. This finding of Völker and Köhler (2013) on state-dependency has also consequences for the usage and interpretation of the eddy-permitting model provided here at the end. If I understood it correctly the eddy-permitting simulations have been performed from modern background conditions, right? If so, this needs to be stated upfront in the main manuscript and asks how useful they can be here. I would even go that far to say that the part of the eddy-permitting results — if they are performed only for modern background conditions — can be completely left out of the draft, since I am not sure we learn a lot about LGM, or H1 ocean physics. But I leave this for the authors to decide. If left in, some careful note of caution on the restricted possibility to transfer the knowledged gained from them to the LGM is needed.

We agree with the Reviewer that state dependency can be important. Völker and Köhler (2013) study the carbon cycle response to shifts in the SH westerlies (SHW) under LGM background conditions. They find that shifting the winds by +/- 10° leads to an atmospheric CO₂ increase of ~2 to 10 ppm, with the largest changes occurring for a poleward shift. As the Reviewer points out, this study highlights the importance of the initial position of the SHW when evaluating the impact of a SHW shift. Nevertheless, as far as we know changes in

the magnitude of the SHW are consistent across initial states, with stronger SHW leading to an increase in atmospheric pCO₂ and weaker SHW leading to an atmospheric pCO₂ decrease. Since the position of the LGM SH westerlies is poorly constrained and could well differ from the one simulated by the coupled model, we have focused on changes in the strength of the SHW instead for the transient simulations. This is also mostly reflected in the text, with for example the abstract focusing on the magnitude.

We have added in the introduction:

“While all numerical experiments performed show that stronger SH westerlies lead to an atmospheric CO₂ increase, the impact of changes in their latitudinal position is more ambiguous and could depend on their initial latitudinal position (Völker et al., 2013, D’Orgeville et al., 2010). However, the latitudinal position of SH westerlies at the LGM remains unclear (Chavaillaz et al., 2013, Kohfeld et al., 2013).”

It has become clear in the past few years that the representation of mesoscale processes is necessary to capture transient Southern Ocean response to changing climate. Similar to other coupled climate models used in paleo-studies, LOVECLIM cannot represent small-scale processes. Völker and Köhler (2013) also note the lack of resolved mesoscale dynamics in their simulations as a significant caveat. To address this short-coming, we are using the high-resolution ocean, sea ice model with coupled biogeochemistry. This novel high-resolution model more accurately simulates Southern Ocean dynamics and supports the fundamental ocean dynamics response to wind forcing suggested by LOVECLIM.

The higher resolution model is equilibrated under present day conditions and we have clarified this point to the reader. However, the computational requirements to equilibrate and then integrate the sensitivity experiment under LGM conditions are prohibitive. When designing the high-resolution model experiment, and given the uncertainties associated with past changes in the SH westerlies, it was decided to force the model with future scenarios estimates, that are also consistent with the changes observed over the last 20 years. The present-day state of the high-resolution model is therefore also useful for understanding the impact of 21st century Southern Ocean wind changes.

The boundary conditions used for the eddy-permitting experiments are now stated in the main text (L. 259):

“To test whether the Southern Ocean CO₂ outgassing response to a strengthening of SH westerlies shown above is a robust feature, we perform an experiment under fixed modern-day forcing with a global high-resolution ocean sea-ice carbon cycle model that resolves most of the ocean mesoscale energy (Methods).”

We have added L. 284-287:

“Even though this simulation was performed under fixed modern-day forcing and the carbon cycle response to a latitudinal shift of the SH westerlies could depend on their initial position (Völker et al., 2013), this high-resolution simulation supports the significant role played by intensified SH westerlies in driving abrupt Southern Ocean CO₂ outgassing.”

2. Based on this importance of background climate state for the carbon cycle I am not sure the final statement (lines 318-320) which transfers the importance of the westerlies during

the analysed time frame (LGM, H1) to the future can be made as such. This at least needs to be stated more carefully or left out completely.

The simulation performed with the global eddy-permitting model starts from modern-day conditions and is forced with projected changes in SH westerly winds. Therefore this experiment can inform on future changes.

We have changed the sentence as follows:

“Given the projected poleward intensification of SH westerlies over the 21st Century, and the fact that the Southern Ocean has absorbed ~10% of anthropogenic CO₂ emissions (Sabine et al., 2004, Mikaloff-Fletcher et al., 2006), our results suggest a future reduction in CO₂ sequestration in the Southern Ocean, with significant impacts on future atmospheric CO₂ and climate change.”

3. I understand that ¹⁴C production rates have been varied according to some reviewer comments, but it is not written in the methods (line 344) at which rate the ¹⁴C production rates are kept constant. At modern levels or at LGM level which I believe would be 20% or so higher? If they are kept constant at modern levels, then the question arises how results would change with higher ¹⁴C production rates rate. Connected with that: Typically it takes quite some time (some 10-kyrs) to run ¹⁴C in the marine carbon cycle into equilibrium. Some details on the length of the spinup — or the equilibrium of the ¹⁴C cycle — or the simulated atm ¹⁴C at LGM should be included.

The initial LGM state was the focus of a separate study and is fully described in Menviel et al., (2017). The Method section briefly describes the experimental set up, namely that this LGM state was obtained by equilibrating the model under 35 ka B.P. boundary conditions after which the model was run transiently until 20 ka with prognostic atmospheric CO₂, δ¹³CO₂ and atmospheric Δ¹⁴C. The model was thus run for 15 kyrs transiently in addition to a full spin-up.

Regarding the atmospheric ¹⁴C production rate, it was diagnosed after the equilibration phase (Δ¹⁴C = 393 permil), based on globally averaged atmosphere-ocean fluxes of ¹⁴C in the equilibrium state. This production rate was then applied as constant for all transient simulations of the LGM as well as for the deglaciation experiments presented here, except for three experiments presented in the supplementary information for which the production rate is varied according to Hain et al., (2014). The rate we are using is 2.05 atoms/cm²/s, which is 10 to 25% higher than present-day and Holocene estimates. We have now added more information into the Method section:

“The initial LGM state was selected amongst 28 LGM experiments based on its representation of oceanic δ¹³C and ventilation age distributions (Menviel et al., 2017). It was obtained by equilibrating LOVECLIM under 35 ka B.P. boundary conditions, namely appropriate orbital parameters (Berger, 1991), Northern Hemisphere ice-sheet extent, topography and albedo (Abe-Ouchi et al., 2007), an atmospheric CO₂ content of 190 ppmv, a δ¹³CO₂ of -6.46‰, and Δ¹⁴C of 393 permil. After this 10,000 years equilibration phase, an atmospheric ¹⁴C production rate of 2.05 atoms/cm²/s was diagnosed and then subsequently applied for the transient simulations. This rate is higher than Holocene and present-day ¹⁴C

production rate estimates of 1.64 and 1.88 atoms/cm²/s (Kovaltsov et al., 2012, Roth and Joos, 2013), consistent with a relatively high LGM $\Delta^{14}\text{C}$ (Reimer et al., 2013)."

4. If I got the overall interpretation for ^{14}C right your paper suggests, that the large and rather abrupt decline in atm ^{14}C between 17.5 and 14.5 ka (also called Mystery Interval) by 190 permil is solely be explained by ocean overturning processes, and changes in ^{14}C production rate can be ignored. If this is not correct, then I have missed something, what others might also miss and some rephrasing might be needed. If this is correct, then you suggest a substantially smaller contribution of ^{14}C production rate than earlier box model studies, e.g. (Köhler et al., 2006; Hain et al., 2014) which needs to be discussed.

The Reviewer is correct in saying that across the period 17.5 to 15 ka our simulations suggest that changes in ocean circulation, and air-sea gas exchange associated with reduced sea-ice and wind changes can explain most of the atm. $\Delta^{14}\text{C}$ change. A simulation in which the ^{14}C production rate is varied according to Hain et al., (2014) is also presented in Figure S4. The varying production rate does have an impact on the simulated $\Delta^{14}\text{C}$ on both centennial and millennial timescales, but it is of lower amplitude than the changes associated with changes in ocean circulation and air-sea gas exchange.

Our results are globally consistent with the ones shown in Hain et al., (2014) over the period 17.5-15ka, with a CO_2 release from the Southern Ocean leading to a larger $\Delta^{14}\text{C}$ decrease than the change in ^{14}C production ratio (their Figure 5A and B, ~75 permil vs ~20 permil). However, it is correct that the $\Delta^{14}\text{C}$ changes due to changes in oceanic circulation as simulated here are larger than the ones simulated in Hain et al., (2014).

Both Hain et al., (2014) and Köhler et al. (2006) use a carbon cycle box model. In Hain et al., (2014) the temperature of the ocean is kept constant while in Kohler et al., (2006) ocean temperature changes are estimated from proxy records. In addition, while Köhler et al., (2006) study the impact of NADW changes on $\Delta^{14}\text{C}$, the impact of changes in Southern Ocean circulation are not explicitly included, and the oceanic circulation changes were not associated with changes in temperature.

The relatively reduced role of changes in production rates found here compared to Hain et al., (2014) and Köhler et al. (2006) could thus be due to:

- i) Ocean physics: the model used in our study is an Ocean General Circulation Model with a dynamic/thermodynamic sea-ice model. Therefore, differences in the 3-dimensional representation of physical changes in the ocean system, changes in sea-ice and changes in air-sea gas exchange due to stronger SH westerlies will impact $\Delta^{14}\text{C}$.
- ii) Our initial state that was constrained by benthic $\delta^{13}\text{C}$ and $\Delta^{14}\text{C}$ records, thus leading to a fairly low $\delta^{13}\text{C}$ and $\Delta^{14}\text{C}$ in the deep ocean, and thus a large vertical gradient in the ocean.

To investigate ii), we analyse additional experiments that were performed from different initial conditions. A set of experiments with NADW cessation (Atl-LH1), NADW cessation and enhanced AABW (Atl-LH1-SO) and NADW cessation and enhanced SHW (Atl-LH1-SHW) were performed from a LGM state with weaker NADW but relatively strong AABW and strong SHW. The AABW and AAIW changes

imposed in this set of simulation are globally smaller than the ones presented in the manuscript (Figure R2 right). In this set of alternate simulations, both enhanced AABW and SHW lead to a significant atmospheric $\Delta^{14}\text{C}$ decrease, but this decrease is much smaller than in experiments LH1-SO and LH1-SHW (Figure R2, middle). It is hypothesised that this difference is primarily due to the initial state, and secondly to the magnitude of the changes imposed.

Figure R2: Impact of initial $\Delta^{14}\text{C}$ gradient in the Southern Ocean on simulated atmospheric $\Delta^{14}\text{C}$ change. From left to right: Southern Ocean $\Delta^{14}\text{C}$ (averaged over 40S-70S, permil) at the LGM initial state used in this study (blue) and for an alternate LGM state with relatively strong AABW and SHW (black); Change in atmospheric $\Delta^{14}\text{C}$ (permil); and change in AABW transport (Sv).

The atmospheric $\Delta^{14}\text{C}$ changes shown in Figure 1 might thus represent an upper estimate of the possible atmospheric $\Delta^{14}\text{C}$ changes.

We have added L. 225-231:

“The atmospheric $\Delta^{14}\text{C}$ changes shown in Figure 1f and S6 were obtained by keeping the atmospheric ^{14}C production rate constant at LGM levels (Methods). Our results thus suggest that most, if not all, of the atmospheric $\Delta^{14}\text{C}$ changes can be attributed to changes in ocean circulation and air-sea gas exchange with a smaller contribution from a varying atmospheric ^{14}C production rate (Fig. S4). The relatively large changes in atmospheric $\Delta^{14}\text{C}$ simulated here as compared to previous studies (Hain et al., 2014, Köhler et al., 2006) can be primarily attributed to the large oceanic $\Delta^{14}\text{C}$ gradient present in the initial state, itself resulting from the weak LGM oceanic circulation. “

5. The references list needs some careful finetuning, since a lot of the references are incomplete (missing paper numbers by Nature Communications or most AGU papers without page numbers (e.g ref 3, 8, 20, 23, 29, 33, 34, 36, 38, 42, 46, 70), wrong page numbers in ref 55.

We thank the Reviewer for carefully checking the references. The references have been amended and should now be correct and complete.

6. Fig 5: If I understood the caption correctly, each of the 4 columns refers to one scenario, but not the scenario name is given in the x-labels, but a suggested strength of the AABW. I suggest to at least add the scenario names to the x axis. Further, I think the x position within one column has no meaning, right? If so, the results of the 7 analysed processes should be placed equidistant along x. If there is a meaning in the x-position, it needs to be explained.

The Reviewer correctly understood the figure. To make it clearer, scenario names were added under each column and the legend was amended as follows to indicate the shift in the x axis (included for visibility).

“Deconvolution of $\Delta p\text{CO}_2$ into (from left to right in each column) its SST, SSS, DIC, and ALK ; solubility (SST+SSS) and combined DIC and alkalinity components (Methods); and total $p\text{CO}_2$ change, for all transient experiments: (LH1: weak AABW, magenta; LH1-SO: strong AABW through buoyancy forcing, green, LH1-SHW: strong AABW through SH westerlies, blue; and LH1-SO-SHW strong AABW through buoyancy forcing and wind, red).”

7. Ice core data: It needs to be mentioned on which age model the EDC, WDC, Taylor Glacier data are plotted. If as published originally, please say so, but you should be aware that WDC has gained a revised WD2014 chronology, which shifted data points (Buizert et al., 2015; Sigl et al., 2016), check out <https://www.ncdc.noaa.gov/paleo-search/study/20246>. Please include dots in the ice core CO_2 time series (as done for $\delta^{13}\text{CO}_2$) in Fig 1, and also include dots in CO_2 and $\delta^{13}\text{CO}_2$ ice core data shown in Fig 3.

The Antarctic air temperature chronology is as described in Parrenin et al., (2013): a modified version of EDC3 ice age scale. To avoid confusion, we have decided not to add additional information and simply keep the reference to Parrenin et al. (2013).

Both the $p\text{CO}_2$ and $\delta^{13}\text{CO}_2$ data are plotted on the WD2014 chronology.

Dots were added to the ice core $p\text{CO}_2$ and $\delta^{13}\text{CO}_2$ curves in figures 1 and 3. Dots on the $p\text{CO}_2$ are a bit smaller not to lose visibility.

The legend of Figure 1 was modified as follows:

“d, Atmospheric $p\text{CO}_2$ (Marcott et al., 2014) on the WD2014 chronology (Buizert et al., 2015); e, $\delta^{13}\text{CO}_2$ (Bauska et al., 2016) on the WD2014 chronology (Buizert et al., 2015)”

8. When referring to figures with multiple panels, please always mention to which subpanel you refer.

We have checked our reference to figures and added information about subpanels.

Additional references:

D'Orgeville, M., Sijp, W., England, M. & Meissner, K., 2010, On the control of glacial-interglacial atmospheric CO_2 variations by the Southern Hemisphere westerlies, *Geophys. Res. Lett.*, 37484 doi:10.1029/2010GL045261.

Kovaltsov, G., Mishev, A. & Usoskin, I., 2012, A new model of cosmogenic production of radiocarbon ^{14}C in the atmosphere., *Earth and Planetary Science Letters* 337-338, 114-120

Roth, R. & Joos, F., 2013, A reconstruction of radiocarbon production and total solar irradiance from the Holocene ^{14}C and CO_2 records: implications of data and model uncertainties, *Climate of the Past*, 9, 1879-1909.

References

Buizert, C., Cuffey, K. M., Severinghaus, J. P., Baggenstos, D., Fudge, T. J., Steig, E. J., Markle, B. R., Winstrup, M., Rhodes, R. H., Brook, E. J., Sowers, T. A., Clow, G. D., Cheng, H., Edwards, R. L., Sigl, M., McConnell, J. R., and Taylor, K. C.: The WAIS Divide deep ice core WD2014 chronology — Part 1: Methane synchronization (68-31 ka BP) and the gas age-ice age

difference, *Climate of the Past*, 11, 153–173, doi:10.5194/cp-11-153-2015, 2015.

Hain, M. P., Sigman, D. M., and Haug, G. H.: Distinct roles of the Southern Ocean and North Atlantic in the deglacial atmospheric radiocarbon decline, *Earth and Planetary Science Letters*, 394, 198 – 208, doi:<http://dx.doi.org/10.1016/j.epsl.2014.03.020>, 2014.

Köhler, P., Muscheler, R., and Fischer, H.: A model-based interpretation of low frequency changes in the carbon cycle during the last 120 000 years and its implications for the reconstruction of atmospheric $\Delta^{14}\text{C}$, *Geochemistry, Geophysics, Geosystems*, 7, Q11N06, doi:10.1029/2005GC001228, 2006.

Sigl, M., Fudge, T. J., Winstrup, M., Cole-Dai, J., Ferris, D., McConnell, J. R., Taylor, K. C., Welten, K. C., Woodruff, T. E., Adolphi, F., Bisiaux, M., Brook, E. J., Buizert, C., Caffee, M. W., Dunbar, N. W., Edwards, R., Geng, L., Iverson, N., Koffman, B., Layman, L., Maselli, O. J., McGwire, K., Muscheler, R., Nishiizumi, K., Pasteris, D. R., Rhodes, R. H., and Sowers, T. A.: The WAIS Divide deep ice core WD2014 chronology - Part 2: Annual-layer counting (0-31 ka BP), *Climate of the Past*, 12, 769–786, doi:10.5194/cp-12-769-2016, 2016.

Völker, C. and Köhler, P.: Responses of ocean circulation and carbon cycle to changes in the position of the Southern hemisphere westerlies at Last Glacial Maximum, *Paleoceanography*, 28, 726–739, doi: 10.1002/2013PA002556, 2013.

Reviewers' Comments:

Reviewer #2:

Remarks to the Author:

The authors have done an excellent job addressing my comments in this and the previous round of review. Throughout the review process that made earnest efforts to clarify their work. I very much look forward to the seeing the paper in published.

Reviewer #4:

Remarks to the Author:

I find that all comments by the reviewer have been well addressed and am satisfied with the current version of the draft.

I have only a few minor issues:

1. The paper of Mariotti et al (2016) should be cited and briefly addressed when discussing atm D14C. This study uses a slightly simpler model than applied here (2.5 D), but already showed the role of old ocean on atm D14C. This was already an improvement with respect to the previous box model studies, and should therefore be included here. I wanted to mention this paper already in the previous review, but I have somehow forgotten it.

Mariotti, V., D. Paillard, L. Bopp, D. M. Roche, and N. Bouttes (2016), A coupled model for carbon and radiocarbon evolution during the last deglaciation, *Geophys. Res. Lett.*, 43, 1306–1313, doi:10.1002/2015GL067489.

2. Please change around line 232:

"While the simulated changes in atmospheric CO₂, δ¹³C_{CO₂ and Δ¹⁴C are in very good agreement with paleo-records, the pCO₂ increase is slightly underestimated over the last 1000 years of HS1."}

to

"While the simulated changes in atmospheric δ¹³C_{CO₂ and Δ¹⁴C are in very good agreement with paleo-records, the pCO₂ increase is slightly underestimated over the last 1000 years of HS1."}

(delete CO₂ at the beginning), otherwise the sentences does not make sense.

3. The description of d¹³C_{CO₂ changes is a bit imprecise:}

At line 58: "... both phases of CO₂ rise associated with 0.2permil decrease in d¹³CCO₂"
Please clarify, that EACH of the phases is related to 0.2permil change. Correct?

Around line 140:

"In line with ice core records, both phases of atmospheric CO₂ rise are associated with 0.25 to 0.3permil d¹³C_{CO₂ decreases."}

I think this is not correct. When they says that the simulations show -0.25 or -0.3permil, but the data about -0.2permil each, I would argue it should not be stated "in line with ice core records", but more along "our simulations overestimate the change in d¹³C_{CO₂ with respect to ice core}

records, simulating -0.25 or -0.3permil, while reconstructions showing only -0.2permil each".

The authors might also go to more detail, e.g their simulation (red in Fig 1e) has an overshoot, and comes back to the reconstructed values, and might then use different numbers to quantify the simulated $\delta^{13}\text{C}_2$.

4. The paper cites about 80 references, and I suggest to cite another one. I believe this is necessary here, since the paper discusses quite a few of the previous modelling studies and reconstructions. I hope this long reference list will be possible within the framework of the journal.

Key=

Black= Reviewers' comments

Blue= Authors' responses

Green = Modified text in the manuscript

REVIEWERS' COMMENTS:

Reviewer #2 (Remarks to the Author):

The authors have done an excellent job addressing my comments in this and the previous round of review. Throughout the review process that made earnest efforts to clarify their work. I very much look forward to the seeing the paper in published.

We thank the Reviewer for the time spent reviewing our manuscript and for the constructive comments that improved our manuscript.

Reviewer #4 (Remarks to the Author):

I find that all comments by the reviewer have been well addressed and am satisfied with the current version of the draft.

We thank the Reviewer for this second helpful review and the positive comments about the study. In the following we provide a point-by-point response to the Reviewer's comments in blue as well as excerpts from the manuscript showing the changes made in green.

I have only a few minor issues:

1. The paper of Mariotti et al (2016) should be cited and briefly addressed when discussing atm D14C. This study uses a slightly simpler model than applied here (2.5 D), but already showed the role of old ocean on atm D14C. This was already an improvement with respect to the previous box model studies, and should therefore be included here. I wanted to mention this paper already in the previous review, but I have somehow forgotten it.

Mariotti, V., D. Paillard, L. Bopp, D. M. Roche, and N. Bouttes (2016), A coupled model for carbon and radiocarbon evolution during the last deglaciation, *Geophys. Res. Lett.*, 43, 1306–1313, doi:10.1002/2015GL067489.

We thank the reviewer for this suggestion. We have amended the paragraph on atm. $\Delta^{14}\text{C}$ as follows:

“The relatively large changes in atmospheric $\Delta^{14}\text{C}$ simulated here are in line with a deglacial experiment performed with CLIMBER-2 (Mariotti et al., 2016) showing the dominant role of reduced Southern Ocean stratification in decreasing atmospheric $\Delta^{14}\text{C}$ across HS1. However, the simulated atmospheric $\Delta^{14}\text{C}$ decrease is larger than the one simulated by global carbon cycle box models (Kohler et al., 2006 and Hain et al., 2014) possibly because of the large oceanic $\Delta^{14}\text{C}$ gradient present in the initial LGM state, itself resulting from the weak LGM oceanic circulation.”

2. Please change around line 232:

"While the simulated changes in atmospheric CO₂, $\delta^{13}\text{CO}_2$ and $\Delta^{14}\text{C}$ are in very good agreement with paleo-records, the pCO₂ increase is slightly underestimated over the last

1000 years of HS1."

to

"While the simulated changes in atmospheric $\delta^{13}\text{CO}_2$ and $\Delta^{14}\text{C}$ are in very good agreement with paleo-records, the $p\text{CO}_2$ increase is slightly underestimated over the last 1000 years of HS1."

(delete CO₂ at the beginning), otherwise the sentences does not make sense.

Thank you for this suggestion, we have amended the sentence as suggested.

3. The description of $d^{13}\text{CO}_2$ changes is a bit imprecise:

At line 58: "... both phases of CO₂ rise associated with 0.2permil decrease in $d^{13}\text{CO}_2$ "

Please clarify, that EACH of the phases is related to 0.2permil change. Correct?

Around line 140:

"In line with ice core records, both phases of atmospheric CO₂ rise are associated with 0.25 to 0.3permil $d^{13}\text{CO}_2$ decreases."

I think this is not correct. When they says that the simulations show -0.25 or -0.3permil, but the data about -0.2permil each, I would argue it should not be stated "in line with ice core records", but more along "our simulations overestimate the change in $d^{13}\text{CO}_2$ with respect to ice core records, simulating -0.25 or -0.3permil, while reconstructions showing only -0.2permil each".

The authors might also go to more detail, e.g their simulation (red in Fig 1e) has an overshoot, and comes back to the reconstructed values, and might then use different numbers to quantify the simulated $Dd^{13}\text{CO}_2$.

L. 58 does state that each phase of atmospheric CO₂ increase is associated with a $\delta^{13}\text{CO}_2$ decrease of ~0.2 ‰.

"each associated with a ~0.2‰ decrease in the atmospheric carbon isotopic "

We have modified L. 140 as follows:

"In line with ice core records (Bauska et al., 2016), each phase of atmospheric CO₂ rise is associated with ~0.25‰ $\delta^{13}\text{CO}_2$ decrease (Fig. 1e). Each of these drops in $\delta^{13}\text{CO}_2$ is however ~0.05‰ higher than the ones recorded in ice-core records, thus leading to a significant overshoot at ~16 ka."

4. The paper cites about 80 references, and I suggest to cite another one. I believe this is necessary here, since the paper discusses quite a few of the previous modelling studies and reconstructions. I hope this long reference list will be possible within the framework of the journal.

We thank the Reviewer for their concern, but there are now 57 references in the main text, which is less than the maximum of 70.